# Natural Policy Gradient for Average Reward Non-Stationary RL

## Abstract

We consider the problem of non-stationary reinforcement learning (RL) in the infinite-horizon average-reward setting. We model it by a Markov Decision Process with time-varying rewards and transition probabilities, with a variation budget of $\Delta_T$. Existing non-stationary RL algorithms focus on model-based and model-free value-based methods. Policy-based methods, however, despite their flexibility in practice, are not theoretically well understood in non-stationary RL. We propose and analyze the first model-free policy-based algorithm, Non-Stationary Natural Actor-Critic (NS-NAC), a policy gradient method with efficient exploration for change and a novel interpretation of learning rates as adapting factors. We present a dynamic regret of $\tilde{\mathcal{O}}(|\mathcal{S}|^{\frac{1}{2}}|\mathcal{A}|^{\frac{1}{2}}\Delta_T^{\frac{1}{9}}T^{\frac{8}{9}})$, where $T$ is the time horizon, and $|\mathcal{S}|$, $|\mathcal{A}|$ are the sizes of the state and action spaces. The regret analysis relies on adapting the Lyapunov function based analysis to dynamic environments and characterizing the effects of simultaneous updates in policy, value function estimate and environment.

## 1 Introduction

Reinforcement Learning is a sequential decision-making framework where an agent learns optimal behavior by iteratively interacting with its environment. At each timestep, the agent observes the current state of the environment, takes an action, receives a reward, and transitions to the next state. While RL has traditionally been studied in stationary environments with time-invariant rewards and state-transition dynamics, this may not always be the case. Consider the examples of a carbon-aware datacenter job scheduler that tracks the dynamic electricity prices and local weather patterns (Yeh et al., 2024) and recommendation systems with evolving user preferences (Chen et al., 2018). Time-varying environments are also observed in inventory control (Mao et al., 2024), healthcare (Chandak et al., 2020), ride-sharing (Kanoria & Qian, 2024), multi-agent systems (Zhang et al., 2021a).

Motivated by these applications, we consider the problem of non-stationary reinforcement learning, modeled by a Markov Decision Process with time-varying rewards and transition probabilities, in the infinite horizon average reward setting. While many works consider discounted rewards (Chandak et al., 2020; Igl et al., 2020; Lecarpentier & Rachelson, 2019), the more challenging average-reward setting is vital in representing problems where the importance of rewards does not decay with time, such as in robotics (Mahadevan, 1996; Peters et al., 2003) or scheduling workloads in cloud computing systems (Jali et al., 2024; Liu et al., 2022). The key challenges for an agent operating in a dynamic environment are learning an optimal behavior policy that varies with the environment, devising an efficient exploration strategy, and effectively incorporating the acquired information into its behavior.

Current algorithms designed for non-stationary MDPs in the average reward setting can be classified broadly into model-based and model-free value-based methods. Model-based solutions incorporate sliding windows, forgetting factors, and confidence interval management mechanisms into UCRL (Cheung et al., 2020; Ortner et al., 2020; Gajane et al., 2018; Jaksch et al., 2010). Model-free value-based methods assimilate restarts and optimism into Q-Learning (Mao et al., 2024; Feng et al., 2023) and LSVI (Zhou et al., 2020; Touati & Vincent, 2020). A significant gap in the literature is the absence of model-free policy-based techniques for time-varying environments. The inherent flexibility of policy-based algorithms makes them suitable for continuous state-action spaces, facilitates efficient parameterization in high-dimensional state-action spaces, and enables effective exploration through stochastic policy learning (Sutton & Barto, 2018).

**Our Contributions.** We make the following contributions to tackle the problem of non-stationary reinforcement learning in the challenging infinite-horizon average reward setting.

1. We propose and analyze Non-Stationary Natural Actor-Critic (NS-NAC), a policy gradient algorithm with efficient exploration for change and a novel interpretation of learning rates as adapting factors. To the best of our knowledge, this is the first model-free policy-based method for time-varying environments.

2. We present a dynamic regret bound of $\tilde{\mathcal{O}}\left(|\mathcal{S}|^{\frac{1}{2}}|\mathcal{A}|^{\frac{1}{2}}\Delta_T^{\frac{1}{9}}T^{\frac{8}{9}}\right)$ under standard assumptions where $T$ is the time horizon, $\Delta_T$ represents the variation budget of rewards and transition probabilities, $|\mathcal{S}||\mathcal{A}|$ is the size of the state-action space and $\tilde{\mathcal{O}}(\cdot)$ hides logarithmic factors.

3. We address theoretical challenges presented by non-stationarity by adapting the Lyapunov function-based analysis of NPG methods in the stationary case to dynamic environments; characterizing the effects of simultaneously evolving actor policy and changing environment on the value function estimates and average reward; and amending the martingale analysis that characterizes the distribution of state-action observations to incorporate time-varying transitions.

## 2 RELATED WORK

**Non-Stationary RL.** Solutions to the non-stationary RL problem can be categorized into passive and active methods. Active algorithms are designed to actively detect changes in the environment in contrast to passive ones which implicitly adapt to new environments without distinct recognition of the change. While we focus our attention on passive techniques with dynamic regret as the performance metric in this work, a comprehensive survey can be found in Padakandla (2021) and Khetarpal et al. (2022). Model-based solutions in the infinite horizon average reward setting incorporate into UCRL a sliding window or a forgetting factor for piecewise stationary MDPs (Gajane et al., 2018), variation aware restarts (Ortner et al., 2020) and a bandit based tuning of sliding window and confidence intervals (Cheung et al., 2020) for gradual or abrupt changes constrained by a variation budget and pessimistic tree search for Lipschitz continuous changes (Lecarpentier & Rachelson, 2019).

In the episodic setting, model-free value based methods assimilate restarts and optimism into Q-Learning (Mao et al., 2024), LSVI (Zhou et al., 2020; Touati & Vincent, 2020) and sliding window and optimistic confidence set based exploration into a value function approximated learning (Feng et al., 2023). Further, in the episodic setting, Lee et al. (2024) proposes strategically pausing learning as an effective solution to non-stationarity with forecasts of the future. Wei & Luo (2021) proposed an algorithm agnostic black-box approach that finds a non-stationary equivalent to optimal regret stationary MDP algorithms. Further, Mao et al. (2024) presents an information theoretic lower bound on the dynamic regret for both the episodic and infinite horizon settings and Peng & Papadimitriou (2024) captures the complexity of updating value functions with any change. We note the distinction between the scope of this work and the body of research on adversarial MDPs which often allow for only changes in rewards, study the static regret and work with full information feedback instead of bandit feedback. We direct readers to Appendix J for a table of comparison of regret bounds across the above mentioned algorithms which we omit here due to a paucity of space.

**Non-Stationary Bandits.** A precursor to non-stationary RL, the multi-armed bandit problem with time-varying rewards was first proposed in Garivier & Moulines (2008). Solutions include UCB with a sliding window or a discounting factor (Garivier & Moulines, 2008), UCB with adaptive blocks of exploration and exploitation (Besbes et al., 2014), Restart-Exp3 (Besbes et al., 2014), Thompson Sampling with a discounting factor (Raj & Kalyani, 2017) and bandit based sliding window tuning (Cheung et al., 2019). Further, while most existing works assume arbitrarily (constrained by variation budget) changing reward distributions and achieve $\Theta(T^{2/3})$ regret, (Jia et al., 2023) achieves an improved $\mathcal{O}(T^{3/5})$ regret when the reward distributions change smoothly. Recent work by Liu et al. (2023a) points out ambiguities in the definition of non-stationary bandits and how the dynamic regret performance metric causes over-exploration, and Liu et al. (2023b) proposes, predictive sampling, an algorithm that deprioritizes acquiring information that loses usefulness quickly.

**Policy Gradient Algorithms for Stationary RL.** Wu et al. (2020) was the first to provide a finite time analysis of the two timescale Advantage Actor-Critic (A2C) with function approximation to a

stationary point in the average reward, Markovian sampling setting and Chen & Zhao (2023) further improved its rate by leveraging a single timescale algorithm. Convergence to global optima of A2C was analyzed in Bai et al. (2024); Murthy et al. (2023) which use a two loop structure with the inner loop estimating the state-action value function and outer loop learning the policy. Further, Lazic et al. (2021) characterized global convergence and regret for the Politex algorithm which combines a mirror descent update with experience replay. Natural Policy Gradient (NPG) was analyzed in the discounted reward case in Agarwal et al. (2021); Khodadadian et al. (2021) and with entropy regularization in Cen et al. (2022). NPG in the average reward setting was characterized in Even-Dar et al. (2009); Murthy & Srikant (2023) which assume access to information of the exact value functions. The most relevant to our work is the Natural Actor Critic (NAC) algorithm where the actor learns the policy by natural gradient ascent and critic estimate the value function. Khodadadian et al. (2022) establishes an $\tilde{\mathcal{O}}(1/T^{1/6})$ rate convergence of the discounted reward tabular NAC to the global optima. Average reward setting with (compatible) function approximation is considered in Wang et al. (2024) and an $\tilde{\mathcal{O}}(1/T^{1/3})$ rate convergence to the global optima is derived. A more detailed review of the Actor-Critic algorithm literature can be found in Section 1.2, Wang et al. (2024).

## 3 PROBLEM SETTING

In this section, we first present preliminaries of a Markov Decision Process and the Natural Actor Critic algorithm in a stationary environment. We then introduce the problem of non-stationary reinforcement learning, where the MDP has time-varying rewards and transition probabilities, and define dynamic regret as a performance metric.

**Notation.** Standard typeface (e.g., $s$) denote scalars and bold typeface (e.g., $\mathbf{r}, \mathbf{A}$) denote vectors and matrices. $\|\cdot\|_\infty$ denotes the infinity norm and $\|\cdot\|_2$ denotes the 2-norm of vectors and matrices. Given two probability measures $P$ and $Q$, $d_{TV}(P, Q) = \frac{1}{2} \int_{\mathcal{X}} |P(dx) - Q(dx)|$ is the total variation distance between $P$ and $Q$, while $D_{\mathrm{KL}}(P\|Q) = \int_{\mathcal{X}} P(dx) \log \frac{P(dx)}{Q(dx)}$ is the KL-divergence. For two sequences $\{a_n\}$ and $\{b_n\}$, $a_n = \mathcal{O}(b_n)$ represents the existence of an absolute constant $C$ such that $a_n \leq C b_n$. Further $\tilde{\mathcal{O}}$ is used to hide logarithmic factors. $|\mathcal{S}|$ denotes the cardinality of a set $\mathcal{S}$. Given a positive integer $T$, $[T]$ denotes the set $\{0, 1, 2, \cdots, T-1\}$.

### 3.1 PRELIMINARIES: STATIONARY RL

**Markov Decision Process.** Reinforcement learning tasks can be modeled as discrete-time Markov Decision Processes (MDPs). An MDP is represented as $\mathcal{M} = (\mathcal{S}, \mathcal{A}, \mathbf{P}, \mathbf{r})$ where $\mathcal{S}$ and $\mathcal{A}$ are, respectively, finite sets of states and actions, $\mathbf{P} \in \mathbb{R}^{|\mathcal{S}||\mathcal{A}| \times |\mathcal{S}|}$ is the transition probability matrix, with $P(s'|s, a) \in [0, 1]$, for $s, s' \in \mathcal{S}, a \in \mathcal{A}$, and $\mathbf{r} \in \mathbb{R}^{|\mathcal{S}||\mathcal{A}|}$ is the reward vector with individual entries $\{r(s, a)\}$ bounded in magnitude by constant $U_R > 0$. An agent in state $s$ takes an action $a \sim \pi(\cdot|s)$ according to a policy $\boldsymbol{\pi}$, where for each state $s$, $\pi(\cdot|s)$ is a probability distribution over the action space. The agent then receives a reward $r(s, a)$ and transitions to the next state $s' \sim P(\cdot|s, a)$. We denote the *policy* by $\boldsymbol{\pi} \in \mathbb{R}^{|\mathcal{S}||\mathcal{A}|}$, which concatenates $\{\pi(\cdot|s)\}_s$. In a stationary MDP, the transition probabilities $\mathbf{P}$ and the rewards $\mathbf{r}$ are *time-invariant*.

**Average Reward and Value Functions.** In this work, we consider the average reward setting (instead of discounted rewards), which is essential to model problems where the importance of rewards does not decay with time (Peters et al., 2003; Liu et al., 2022). If the Markov chain is ergodic, the average reward received by an agent over time following policy $\boldsymbol{\pi}$ converges to

$$J^{\boldsymbol{\pi}} := \lim_{T \to \infty} \frac{\sum_{t=0}^{T-1} r(s_t, a_t)}{T} = \mathbb{E}_{s \sim d^{\boldsymbol{\pi}, \mathbf{P}}(\cdot), a \sim \pi(\cdot|s)} [r(s, a)], \qquad (1)$$

where $d^{\boldsymbol{\pi}, \mathbf{P}}$ is the stationary distribution over states induced by policy $\boldsymbol{\pi}$ and transition probabilities $\mathbf{P}$. The *relative* state-value function defines the overall reward (relative to the average reward) accumulated by policy $\boldsymbol{\pi}$ when starting from state $s$ as

$$V^{\boldsymbol{\pi}}(s) := \mathbb{E}\left[\sum_{t=0}^{\infty} (r(s_t, a_t) - J^{\boldsymbol{\pi}}) \,\Big|\, s_0 = s\right],$$

where the expectation is over the trajectory rolled out by $a_t \sim \pi(\cdot|s_t)$ and $s_{t+1} \sim P(\cdot|s_t, a_t)$. Similarly, the relative state-action value function defines the overall reward (relative to the average reward) accumulated by policy $\boldsymbol{\pi}$ when starting from state $s$ and action $a$ as

$$Q^{\boldsymbol{\pi}}(s, a) := \mathbb{E}\left[\sum_{t=0}^{\infty} (r(s_t, a_t) - J^{\boldsymbol{\pi}}) | s_0 = s, a_0 = a\right].$$

**Natural Actor-Critic.** The goal of an agent is to find a policy that maximizes the average reward

$$\boldsymbol{\pi}^{\star} = \max_{\boldsymbol{\pi}} J^{\boldsymbol{\pi}} = \max_{\boldsymbol{\pi}} \mathbb{E}_{s \sim d^{\boldsymbol{\pi}}, \mathbf{P}(\cdot), a \sim \pi(\cdot|s)} \left[r(s, a)\right].$$

Here, we consider the actor-critic class of policy-based algorithms. While actor-only methods are at a disadvantage due to inefficient use of samples and high variance and critic-only methods are at a risk of the divergence from the optimal policy, actor-critic methods provide the best of both worlds (Wu et al., 2020). An actor-critic algorithm learns the policy and the value function simultaneously by gradient methods. Further, the natural actor-critic leverages the second-order method of natural gradient to establish guarantees of global optimality (Bhatnagar et al., 2009; Khodadadian et al., 2022). The *actor* updates the policy by performing a natural gradient ascent (Martens, 2020) step

$$\boldsymbol{\pi} \leftarrow \boldsymbol{\pi} + \beta F_{\boldsymbol{\pi}}^{-1} \nabla J^{\boldsymbol{\pi}}, \quad \text{where} \quad F_{\boldsymbol{\pi}} := \mathbb{E}_{s \sim d^{\boldsymbol{\pi}}, \mathbf{P}(\cdot), a \sim \pi(\cdot|s)} \left[\nabla \log \pi(a|s) \left(\nabla \log \pi(a|s)\right)^{\top}\right]. \quad (2)$$

$F_{\boldsymbol{\pi}}$ is called the Fisher Information matrix. The gradient of the average reward is given by the Policy Gradient Theorem (Sutton & Barto, 2018, Section 13.2) as

$$\nabla J^{\boldsymbol{\pi}} = \mathbb{E}_{s \sim d^{\boldsymbol{\pi}}, \mathbf{P}(\cdot), a \sim \pi(\cdot|s)} \left[Q^{\boldsymbol{\pi}}(s, a) \nabla \log \pi(a|s)\right].$$

The *critic* enables an approximate policy gradient computation by estimating the Q-Value function $Q^{\boldsymbol{\pi}}(s, a)$ using TD-learning as

$$Q(s, a) \leftarrow Q(s, a) + \alpha \left[r(s, a) - \eta + Q(s', a') - Q(s, a)\right],$$

where $s' \sim P(\cdot|s, a)$, $a' \sim \pi(\cdot|s')$, and $\eta$ is an estimate of the average reward $J^{\boldsymbol{\pi}}$. While we consider the tabular setting in this work, actor-critic algorithms can be extended to the function approximation case by parameterizing the policy and/or value function.

## 3.2 NON-STATIONARY RL

In this work, we study reinforcement learning with *time-varying environments*. The MDP is modeled by a sequence of environments $\mathcal{M} = \{\mathcal{M}_t = (\mathcal{S}, \mathcal{A}, \mathbf{P}_t, \mathbf{r}_t)\}_{t=0}^{T-1}$, with time-varying rewards $\{\mathbf{r}_t\}$ and transition probabilities $\{\mathbf{P}_t\}$. At each time $t$, the agent in state $s_t$ takes action $a_t$, receives a reward $r_t(s_t, a_t)$, and transitions to the next state $s_{t+1} \sim P_t(\cdot|s_t, a_t)$. The cumulative change in the reward and transition probabilities is quantified in terms of *variation budgets* $\Delta_{R,T}$ and $\Delta_{P,T}$ as

$$\Delta_{R,T} = \sum_{t=0}^{T-1} \|\mathbf{r}_{t+1} - \mathbf{r}_t\|_{\infty}, \quad \Delta_{P,T} = \sum_{t=0}^{T-1} \|\mathbf{P}_{t+1} - \mathbf{P}_t\|_{\infty}, \quad \Delta_T = \Delta_{R,T} + \Delta_{P,T}. \quad (3)$$

Note that while the overall budgets $\Delta_{R,T}, \Delta_{P,T}$ may be used as inputs by the agent, the variations at a given time $t$, $\|\mathbf{r}_{t+1} - \mathbf{r}_t\|_{\infty}$ and $\|\mathbf{P}_{t+1} - \mathbf{P}_t\|_{\infty}$, are unknown.

We denote the long-term average reward obtained by following policy $\boldsymbol{\pi}_t$ in the environment $\mathcal{M}_t$ by

$$J_t^{\boldsymbol{\pi}_t} = \mathbb{E}_{s \sim d^{\boldsymbol{\pi}_t}, \mathbf{P}_t(\cdot), a \sim \pi(\cdot|s)} \left[r_t(s, a)\right].$$

Further, the state and state-action value functions at time $t$ are solutions to the Bellman equations

$$V_t^{\boldsymbol{\pi}_t}(s) = \sum_{a \in \mathcal{A}} \pi(a|s) Q_t^{\boldsymbol{\pi}_t}(s, a) \quad \text{and} \quad Q_t^{\boldsymbol{\pi}_t}(s, a) = r_t(s, a) - J_t^{\boldsymbol{\pi}_t} + \sum_{s' \in \mathcal{S}} P_t(s'|s, a) V_t^{\boldsymbol{\pi}_t}(s').$$

The solutions to the Bellman equations $\mathbf{Q}_t^{\boldsymbol{\pi}_t} = \{\mathbf{Q}_{t,E}^{\boldsymbol{\pi}_t} + c\mathbf{1} | \mathbf{Q}_{t,E}^{\boldsymbol{\pi}_t} \in E, c \in \mathbb{R}\}$ where E is the subspace orthogonal to the all ones vector and $\mathbf{Q}_{t,E}^{\boldsymbol{\pi}_t}$ is the unique solution in $E$ (Zhang et al., 2021b).

The goal of the agent is to maximize the time-averaged reward $\sum_{t=0}^{T-1} r_t(s_t, a_t)/T$. We measure the performance using an equivalent metric called the *dynamic regret* defined as

$$\text{Dyn-Reg}(\mathcal{M}, T) := \mathbb{E}\left[\sum_{t=0}^{T-1} J_t^{\boldsymbol{\pi}_t^{\star}} - r_t(s_t, a_t)\right], \quad (4)$$

where $\boldsymbol{\pi}_t^\star = \arg\max_{\boldsymbol{\pi}} J_t^{\boldsymbol{\pi}}$ is the optimal policy in the environment $\mathcal{M}_t = (\mathcal{S}, \mathcal{A}, \mathbf{P}_t, \mathbf{r}_t)$ at time $t$. The optimal average reward $J_t^{\boldsymbol{\pi}_t^\star}$ associated with $\boldsymbol{\pi}_t^\star$ can be computed by solving the linear program (28) described in Appendix G. The notion of dynamic regret has also been used in several recent works (Cheung et al., 2020; Fei et al., 2020; Zhou et al., 2020; Mao et al., 2024; Feng et al., 2023). It is more challenging to analyze than static regret, which compares the cumulative reward collected by an agent against that of a single stationary optimal policy (Even-Dar et al., 2009; Touati & Vincent, 2020). Further, in applications such as robotics or network routing, where the underlying environment evolves over time, a single best action/policy in hindsight might not be a realistic benchmark. On the other hand, dynamic regret provides a more useful performance measure.

**Challenges due to Non-Stationarity.** When running policy-gradient methods in stationary RL, the policy evolves to efficiently learn a fixed environment $(\mathbf{P}, \mathbf{r})$. However, in non-stationary case, the environment $(\mathbf{P}_t, \mathbf{r}_t)$ also changes over time. Therefore, the agent chases a moving target, namely, the *time-varying optimal policy* $\boldsymbol{\pi}_t^\star$, resulting in the following unique challenges.

- *Explore-for-Change vs Exploit:* The agent needs to explore more aggressively than in the stationary setting to adapt to the changing dynamics. As an example, a sub-optimal action at the current timestep may become optimal at a later timestep, necessitating re-exploration. This is in sharp contrast to stationary RL, where sub optimal actions are picked less often as time progresses.
- *Forgetting Old Environments:* The policy and value function estimates must evolve quickly lest they might become irrelevant once the environment changes significantly. However, observations are noisy and the agent needs to collect multiple samples to obtain confident estimates. Hence, the agent has to carefully balance the rate of forgetting the old environment versus learning a new one.

## 4 ALGORITHM

In this section, we present Non-Stationary Natural Actor-Critic (NS-NAC), a two-timescale natural policy gradient method with an entropy based exploration for change and step-sizes designed to carefully balance the rate of forgetting the old environment and adapting to a new one.

---

**Algorithm 1** Non-Stationary Natural Actor-Critic (NS-NAC)

---

1: **Input** time horizon $T$, variation budgets $\Delta_{R,T}$, $\Delta_{P,T}$, projection radius $R_Q$
2: **Set** actor step-size $\beta$, critic step-size $\alpha$, average reward step-size $\gamma$ and exploration parameter $\epsilon$
3: **Initialize** policy $\pi_0(a|s) = \frac{1}{|\mathcal{A}|}$, value function $Q_0(s, a) = 0$, for all $s \in \mathcal{S}, a \in \mathcal{A}$, and average reward estimate $\eta_0 = 0$
4: Sample $s_0 \sim \text{Unif}\{0, 1, \ldots, |\mathcal{S}| - 1\}$, take action $a_0 \sim \pi_0(\cdot|s_0)$
5: **for** $t = 0, 1, 2, \ldots, T - 1$ **do**
6:     Observe reward $r_t(s_t, a_t)$, next state $s_{t+1} \sim P_t(\cdot|s_t, a_t)$, and take action $a_{t+1} \sim \pi_t(\cdot|s_{t+1})$
7:     $\eta_{t+1} \leftarrow \eta_t + \gamma\left(r_t(s_t, a_t) - \eta_t\right)$         ▷ Average Reward estimate
8:     $Q_{t+1}(s_t, a_t) \leftarrow Q_t(s_t, a_t) + \alpha\left[r_t(s_t, a_t) - \eta_t + Q_t(s_{t+1}, a_{t+1}) - Q_t(s_t, a_t)\right]$
9:     $\mathbf{Q}_{t+1} \leftarrow \Pi_{R_Q}\left[\mathbf{Q}_{t+1}\right]$         ▷ Critic update
10:     $\pi_{t+1}(a|s) = \frac{(\pi_t(a|s))^{1-\beta\epsilon} \exp(\beta Q_t(s,a))}{\sum_{a' \in \mathcal{A}} (\pi_t(a'|s))^{1-\beta\epsilon} \exp(\beta Q_t(s,a'))}$, for all $s, a$     ▷ Actor update
11: **end for**

---

The NS-NAC algorithm seeks to maximize the total reward received over the time horizon $T$, given the variation budgets $\Delta_{R,T}$ and $\Delta_{P,T}$. The pseudocode is presented in Algorithm 1. At timestep $t$, $\boldsymbol{\pi}_t$ denotes the tabular policy with $\pi(\cdot|s)$, such that $\pi(a|s) \geq 0$, for all $a \in \mathcal{A}$, and $\sum_a \pi(a|s) = 1$, for all $s \in \mathcal{S}$. $\boldsymbol{\pi}_t^\star = \arg\max_{\boldsymbol{\pi}} J_t^{\boldsymbol{\pi}}$ is the optimal policy in the environment $\mathcal{M}_t$. The estimate of the tabular state-action value function $\mathbf{Q}_t^{\boldsymbol{\pi}_t}$ is denoted by $\mathbf{Q}_t \in \mathbb{R}^{|\mathcal{S}||\mathcal{A}|}$. $\eta_t$ denotes the estimate of the average reward $J_t^{\boldsymbol{\pi}_t}$. We consider the class of tabular softmax policies, parameterized by $\boldsymbol{\theta} \in \mathbb{R}^{|\mathcal{S}||\mathcal{A}|}$, for all $s, a$, with

$$\pi(a|s) = \frac{\exp \theta_{s,a}}{\sum_{a' \in \mathcal{A}} \exp \theta_{s,a'}}.$$

Note that while we consider the tabular formulation in this work for the ease of presentation of regret analysis, the NS-NAC can also be extended to the function approximation setting.

To ensure the algorithm sufficiently *explores for change*, we consider an entropy based incentive for the actor choosing diverse actions as

$$\max_{\boldsymbol{\pi}} J_t^{\boldsymbol{\pi}} + \mathbb{E}_{s \sim d^{\boldsymbol{\pi}, \mathbf{P}_t}(\cdot)} \left[ \epsilon \mathcal{H}(\boldsymbol{\pi}(\cdot|s)) \right] = \max_{\boldsymbol{\pi}} \mathbb{E}_{s \sim d^{\boldsymbol{\pi}, \mathbf{P}_t}, a \sim \boldsymbol{\pi}(\cdot|s)} \left[ r_t(s, a) - \epsilon \log \boldsymbol{\pi}(a|s) \right]$$

where $\epsilon$ controls the weight of exploration and $\mathcal{H}(\boldsymbol{\pi}(\cdot|s)) = \mathbb{E}_{a \sim \boldsymbol{\pi}(\cdot|s)}[- \log \boldsymbol{\pi}(a|s)]$ is the entropy of policy $\boldsymbol{\pi}(\cdot|s)$. At time $t$, the *actor* (slower timescale) takes a natural gradient ascent step towards the optimal policy in environment $\mathcal{M}_t$ by the Policy Gradient Theorem (Sutton & Barto, 2018) as

$$\boldsymbol{\theta}_{t+1} = \boldsymbol{\theta}_t + \beta F_{\boldsymbol{\pi}_t}^{-1} \mathbb{E}_{s,a} \left[ (\mathbf{Q}_t^{\boldsymbol{\pi}_t}(s, a) - \epsilon) \nabla \log \pi_t(a|s) \right] = (1 - \beta\epsilon)\boldsymbol{\theta}_t + \beta \mathbf{Q}_t^{\boldsymbol{\pi}_t}$$

where $\beta$ is the actor step-size, and $F_{\boldsymbol{\pi}}$ is the Fisher Information matrix (2). In the absence of knowledge of the exact natural gradient, the actor uses an estimate to update the policy as

$$\boldsymbol{\theta}_{t+1} \leftarrow (1 - \beta\epsilon)\boldsymbol{\theta}_t + \beta \mathbf{Q}_t. \tag{5}$$

With the softmax parameterization, this is equivalent to the update equation in line 10.

The *critic* (faster timescale) estimates the tabular state-action value function of the current policy $\boldsymbol{\pi}_t$ as $\mathbf{Q}_t$ using TD-Learning (line 8) with step-size $\alpha$. The projection step in line 9 is defined as $\Pi_{R_Q}[\mathbf{x}] := \arg\min_{\|\mathbf{y}\|_2 \leq R_Q} \|\mathbf{x} - \mathbf{y}\|_2$. [1]. The average reward estimate $\eta_t$ is updated (line 7) with step-size $\gamma$. Using a two timescale technique with $\alpha \gg \beta$, NS-NAC thus enables the actor to chase the moving target $\boldsymbol{\pi}_t^\star$ facilitated by the critic updates of the value function estimates which adapt to the changed data distribution. In the stationary RL case, this change in data distribution is induced solely by the evolving actor policy, while in non-stationary RL, the time-varying environment $(\mathbf{P}_t, \mathbf{r}_t)$ further exacerbates it. Further, as Theorem 1 suggests, a careful selection of the step-sizes enables NS-NAC to balance the rate of *forgetting the old environment* versus learning a new one.

# 5 REGRET ANALYSIS

In this section, we set up notation and assumptions, state our main result establishing an upper bound on the dynamic regret and present a sketch of the proof.

## 5.1 ASSUMPTIONS

**Notation.** We denote an observation $O_t = (s_t, a_t, s_{t+1}, a_{t+1})$. If $d^{\boldsymbol{\pi}_t, \mathbf{P}_t}(\cdot)$ is the stationary distribution induced over the states, we define the matrices $\mathbf{A}(O_t), \bar{\mathbf{A}}^{\boldsymbol{\pi}_t, \mathbf{P}_t} \in \mathbb{R}^{|\mathcal{S}||\mathcal{A}| \times |\mathcal{S}||\mathcal{A}|}$ as

$$\mathbf{A}(O_t)_{i,j} = \begin{cases} -1, & \text{if } (s_t, a_t) \neq (s_{t+1}, a_{t+1}), i = j = (s_t, a_t) \\ 1, & \text{if } (s_t, a_t) \neq (s_{t+1}, a_{t+1}), i = (s_t, a_t), j = (s_{t+1}, a_{t+1}) \\ 0, & \text{else} \end{cases}$$

$$\bar{\mathbf{A}}^{\boldsymbol{\pi}_t, \mathbf{P}_t} = \mathbb{E}_{s \sim d^{\boldsymbol{\pi}_t, \mathbf{P}_t}(\cdot), a \sim \boldsymbol{\pi}_t(\cdot|s), s' \sim \mathbf{P}_t(\cdot|s,a), a' \sim \boldsymbol{\pi}_t(\cdot|s')} \left[ \mathbf{A}(s, a, s', a') \right].$$

If $D^{\boldsymbol{\pi}_t, \mathbf{P}_t} = diag\left( d^{\boldsymbol{\pi}_t, \mathbf{P}_t}(s)\pi_t(a|s) \right)$ and $\mathbf{1}$ is the all ones vector, then the TD limiting point satisfies

$$\mathbf{D}^{\boldsymbol{\pi}_t, \mathbf{P}_t} \left( \mathbf{r}_t - J_t^{\boldsymbol{\pi}_t}\mathbf{1} \right) + \bar{\mathbf{A}}^{\boldsymbol{\pi}_t, \mathbf{P}_t} \mathbf{Q}_t^{\boldsymbol{\pi}_t} = 0. \tag{6}$$

**Assumption 1** (Uniform Ergodicity). *A Markov chain generated by implementing policy $\boldsymbol{\pi}$ and transition probabilities $\mathbf{P}$ is called uniformly ergodic, if there exists $m > 0$ and $\rho \in (0, 1)$ such that*

$$d_{TV}\left( P(s_\tau \in \cdot | s_0 = s), d^{\boldsymbol{\pi}, \mathbf{P}} \right) \leq m\rho^\tau \; \forall \tau \geq 0, s \in \mathcal{S},$$

*where $d^{\boldsymbol{\pi}, \mathbf{P}}$ is the stationary distribution induced over the states. We assume Markov chains induced by all potential policies $\boldsymbol{\pi}_t$ in all environments $\mathbf{P}_t, t \in [T]$, are uniformly ergodic. Further, if $\boldsymbol{\pi}_t^\star$ denotes the optimal policy for the environment $\mathcal{M}_t = (\mathcal{S}, \mathcal{A}, \mathbf{P}_t, \mathbf{r}_t)$, there exists $C > 0$ such that*

$$C = \inf_{s,t,t',\boldsymbol{\pi}} \frac{d^{\boldsymbol{\pi}, \mathbf{P}_{t'}}(s)}{d^{\boldsymbol{\pi}_t^\star, \mathbf{P}_t}(s)} > 0.$$

---

[1]See Lemma 5.1 and the following discussion on how to choose $R_Q$.

**Lemma 5.1** (Zhang et al. (2021b), Lemma 2). *Under Assumption 1, for all potential policies $\pi_t$ in all environments $\mathbf{P}_t$, $t \in [T]$, the matrix $\bar{\mathbf{A}}^{\pi_t, \mathbf{P}_t}$ is negative semi-definite. Further, define its maximum non-zero eigenvalue as $-\lambda$.*

Assumption 1 is standard in literature (Murthy & Srikant, 2023; Wu et al., 2020; Zou et al., 2019). Also note that we set the projection radius $R_Q = 2U_R\lambda^{-1}$ in line 9 of Algorithm 1 because $\| (\bar{\mathbf{A}}^{\pi_t, \mathbf{P}_t})^\dagger \|_2 \leq \lambda^{-1}$ where $\dagger$ represents the pseudo-inverse.

## 5.2 BOUNDS ON REGRET

The dynamic regret achieved by Algorithm 1 can be upper bounded as follows.

**Theorem 1.** *If Assumption 1 is satisfied and the step-sizes and exploration parameter are chosen as $0 < \alpha, \beta, \gamma, \epsilon < 1/2$ in Algorithm 1, then we have*

$$Dyn\text{-}Reg(\mathcal{M}, T) = \mathbb{E}\left[\sum_{t=0}^{T-1} J_t^{\pi_t^\star} - r_t(s_t, a_t)\right]$$

$$\leq \underbrace{\tilde{\mathcal{O}}\left(\frac{N}{\beta\epsilon}\right) + \tilde{\mathcal{O}}\left(\epsilon T\right) + \tilde{\mathcal{O}}\left(\sqrt{\frac{T}{\alpha}}\right)}_{\text{Effect of initialization and exploration}} + \underbrace{\tilde{\mathcal{O}}\left(\frac{\beta T}{\alpha}\right) + \tilde{\mathcal{O}}\left(T\sqrt{\beta}\right)}_{\substack{\text{Bounds cumulative change} \\ \text{in policy over horizon } T}} + \underbrace{\tilde{\mathcal{O}}\left(\frac{\beta T}{\gamma}\right) + \tilde{\mathcal{O}}\left(T\sqrt{\gamma}\right) + \tilde{\mathcal{O}}\left(\sqrt{\frac{T}{\gamma}}\right)}_{\text{Error in Average Reward Estimate } (\eta_t) \text{ at Critic}}$$

$$+ \underbrace{\tilde{\mathcal{O}}\left(T\sqrt{\alpha}\right)}_{\substack{\text{Bounds cumulative} \\ \text{change in critic estimates}}} + \underbrace{\tilde{\mathcal{O}}\left(\frac{\Delta_T T}{N}\right) + \tilde{\mathcal{O}}\left(\Delta_T\right) + \tilde{\mathcal{O}}\left(\sqrt{\Delta_T T}\right) + \tilde{\mathcal{O}}\left(\Delta_T^{1/3} T^{2/3}\left(\frac{1}{\alpha} + \frac{1}{\gamma}\right)\right)}_{\text{Error due to Non-Stationarity}},$$

$$(7)$$

*where $\Delta_T = \Delta_{R,T} + \Delta_{P,T}$, $\tilde{\mathcal{O}}(\cdot)$ hides the constants and logarithmic dependence on the time horizon $T$, and $N$ is a parameter in the analysis which divides the total horizon $T$ into $N$ segments of equal length. Our results hold for any $1 \leq N \leq T$ and when $N$, together with $\alpha, \beta, \gamma, \epsilon$, are optimized, we get $\alpha^\star = \gamma^\star = \left(\frac{\Delta_T}{T}\right)^{2/9}$, $\beta^\star = \left(\frac{\Delta_T}{T}\right)^{3/9}$, $N^\star = \Delta_T^{8/9} T^{1/9}$, $\epsilon^\star = \left(\frac{\Delta_T}{T}\right)^{4/9}$. The resulting regret (with explicit dependence on the size of the state-action space $|\mathcal{S}|, |\mathcal{A}|$) is*

$$Dyn\text{-}Reg(\mathcal{M}, T) \leq \tilde{\mathcal{O}}\left(|\mathcal{S}|^{\frac{1}{2}}|\mathcal{A}|^{\frac{1}{2}}\Delta_T^{\frac{1}{9}} T^{\frac{8}{9}}\right). \quad (8)$$

We provide a sketch of the proof in Section 5.3 and the full proof in Appendix C.

**Effect of Non-Stationarity.** The variation budget $\Delta_T$ (3) represents the extent of non-stationarity of the environment. In Theorem 1, as the variation budget increases, so do the optimal step-sizes and exploration parameter and the regret incurred (8). This observation is consistent with the intuition that in a rapidly changing environment, the algorithm must adapt quickly and explore more (hence, larger step-sizes and exploration parameter). However, as a result, the algorithm cannot exploit its current policy and value-function estimates, which soon become outdated (hence, higher regret). Also, in environments with larger state/action spaces, the agent requires proportionately more samples to detect changes and learn a good policy.

Next, we compare the upper bound in Theorem 1 with the following lower bound on dynamic regret.

**Theorem 2** ((Mao et al., 2024), Proposition 1). *For any learning algorithm, there exists a non-stationary MDP such that the dynamic regret of the algorithm is at least $\Omega(|\mathcal{S}|^{1/3}|\mathcal{A}|^{1/3}\Delta_T^{1/3} T^{2/3})$.*

**Gap between Bounds.** To the best of our knowledge, this is the first bound on dynamic regret for model-free policy-based algorithm in the infinite horizon average reward setting. We conjecture that the gap between the bounds results from a slack in the analysis of the underlying Natural Actor-Critic (NAC) algorithm. The best-known regret bounds for NAC for an infinite horizon *stationary* MDP in the (compatible) function approximation setting with a single timescale algorithm is $\tilde{\mathcal{O}}(T^{2/3})$ (Wang et al., 2024), and tabular setting with a two timescale algorithm is $\tilde{\mathcal{O}}(T^{5/6})$ (Khodadadian et al., 2022). A single timescale analysis is considerably more intricate, and we opt for a two-timescale algorithm in our work to effectively characterize non-stationarity, which is the focus of this work. The analysis of the actor involves the norm of the critic estimation error $\|\mathbf{Q}_t - \mathbf{Q}_t^{\pi_t}\|$ (Proposition 1) whereas

guarantees for critic establish a bound on norm-squared of the error $\|\mathbf{Q}_t - \mathbf{Q}_t^{\boldsymbol{\pi}_t}\|^2$ (Proposition 2). This mismatch, which underlies the sub-optimality of the current best stationary infinite horizon NAC analysis, becomes even more pronounced in non-stationary environments. [2]

Moreover, the infinite horizon setting (only one sample per environment is available) is harder than the episodic setting (environment remains stationary during the episode). Also, note that model-free policy-based methods for non-stationary RL are more challenging in the infinite horizon setting due to the absence of high probability bounds for policy gradient algorithms which the model-based methods leverage. Further, the $|\mathcal{S}|^{\frac{1}{2}}|\mathcal{A}|^{\frac{1}{2}}$ dependence occurs due to the use of a tabular policy and state-action value function. These terms can be improved using a low-dimensional function approximation, a technique amenable to integration into the actor-critic framework (Chen & Zhao, 2023; Wang et al., 2024; Wu et al., 2020).

## 5.3 PROOF SKETCH

We now present a sketch of the proof where we address the following theoretical challenges that non-stationarity presents in NS-NAC. (a) Stationary environment NAC analyses use the KL-divergence to the optimal policy as a Lyapunov function. What is an appropriate function for dynamic environments? (b) How do the simultaneously varying environment and evolving policy affect the average reward and state-action value function? (c) How do the time-varying transition probabilities affect the martingale-based argument used to analyze the Markovian noise ?

**Regret Decomposition.** We start by decomposing the dynamic regret as

$$\mathbb{E}\left[\sum_{t=0}^{T-1} J_t^{\boldsymbol{\pi}_t^\star} - r_t(s_t, a_t)\right] = \underbrace{\mathbb{E}\left[\sum_{t=0}^{T-1} J_t^{\boldsymbol{\pi}_t^\star} - J_t^{\boldsymbol{\pi}_t}\right]}_{I_1:\ \substack{\text{Difference of optimal versus} \\ \text{actual average reward}}} + \underbrace{\mathbb{E}\left[\sum_{t=0}^{T-1} J_t^{\boldsymbol{\pi}_t} - r_t(s_t, a_t)\right]}_{I_2:\ \substack{\text{Difference of actual versus} \\ \text{instantaneous reward}}}, \qquad (9)$$

where $I_1$ characterizes the difference between the average reward of the actual policy $\boldsymbol{\pi}_t$ at time $t$ relative to the optimal policy $\boldsymbol{\pi}_t^\star$. The second term $I_2$ analyzes the gap between the average reward and the actual rewards received due to the stochasticity of the Markovian sampling process.

**Actor (Proposition 1).** We first bound $I_1$ in (9) by adapting the Natural Policy Gradient analysis for average-reward stationary MDPs in Murthy & Srikant (2023) to non-stationary environments. NPG in the stationary case is analyzed by characterizing the drift of the policy towards the optimal policy using an appropriate Lyapunov function. In non-stationary case we innovatively decompose and analyze the change in the environment from the drift of the policy as follows. We start by dividing the total horizon $T$ into $N$ segments of length $T_0$ each and break down $I_1$ as

$$I_1 = \mathbb{E}\left[\sum_{n=0}^{N-1} \sum_{j=nT_0}^{(n+1)T_0-1} \underbrace{\left(J_j^{\boldsymbol{\pi}_j^\star} - J_{nT_0}^{\boldsymbol{\pi}_{nT_0}^\star}\right)}_{I_3:\ \substack{\text{optimal avg. reward} \\ \text{across two environments}}} + \underbrace{\left(J_{nT_0}^{\boldsymbol{\pi}_{nT_0}^\star} - J_{nT_0}^{\boldsymbol{\pi}_j}\right)}_{I_4:\ \substack{\text{avg. reward} \\ \text{sub-optimality}}} + \underbrace{\left(J_{nT_0}^{\boldsymbol{\pi}_j} - J_j^{\boldsymbol{\pi}_j}\right)}_{I_5:\ \substack{\text{avg. reward with same} \\ \text{policy in two environments}}}\right].$$

To analyze the drift in the policy, we consider the beginning of each segment as a pseudo-restart and the environment to be pseudo-stationary. We benchmark the policies learned in each segment $n \in [N]$ against the optimal average reward at the initial time step $nT_0$ i.e. $J_{nT_0}^{\boldsymbol{\pi}_{nT_0}^\star}$. [3] We bound $I_4$ by a mirror descent style analysis for each segment $n$ with the Lyapunov function adapted to non-stationarity as

$$W(\boldsymbol{\pi}_j) = \sum_s d^{\boldsymbol{\pi}_{nT_0}^\star, \mathbf{P}_{nT_0}}(s) D_{\mathrm{KL}}(\boldsymbol{\pi}_{nT_0}^\star(\cdot|s)\|\boldsymbol{\pi}_j(\cdot|s)).$$

---

[2] The term characterizing the difference in value functions at consecutive timesteps $\|\mathbf{Q}_{t+1}^{\boldsymbol{\pi}_{t+1}} - \mathbf{Q}_t^{\boldsymbol{\pi}_t}\|$ is the cause for the $\tilde{\mathcal{O}}\left(\Delta_T^{1/3} T^{2/3}\left(\frac{1}{\alpha} + \frac{1}{\gamma}\right)\right)$ term (see $I_4$, $I_5$ in Proposition 2).

[3] Note that we say pseudo-stationary because we characterize the effect of change in the environment separately.

In addition, since NS-NAC does not have access to the exact value functions $\mathbf{Q}_j^{\boldsymbol{\pi}_j}$, $I_4$ also depends on the critic estimation error $\|\mathbf{Q}_j^{\boldsymbol{\pi}_j} - \mathbf{Q}_j\|_\infty$.

We analyze the change in the environment next. We bound $I_3$, the difference in the optimal average rewards in two different environments, in terms of the corresponding changes in the environment $\|\mathbf{r}_j - \mathbf{r}_{nT_0}\|_\infty$ and $\|\mathbf{P}_j - \mathbf{P}_{nT_0}\|_\infty$ (Lemma G.4) by a clever use of the linear programming formulation of an MDP. Similarly, we deftly bound $I_5$, the difference in average rewards when following the same policy $\boldsymbol{\pi}_j$ in two different environments, in terms of the change in the environment (Lemma G.5). Note that $N$ is a parameter that balances the accounting of the effects of the changing environment and the drifting policy and we optimize it in Theorem 1 to minimize regret.

**Critic (Proposition 2).** We bound the critic estimation error $\boldsymbol{\psi}_t = \Pi_E\left[\mathbf{Q}_t - \mathbf{Q}_t^{\boldsymbol{\pi}_t}\right]$ [4] by adapting the critic analysis used in stationary MDPs (Wu et al., 2020; Khodadadian et al., 2022; Zhang et al., 2021b) to non-stationary environments. We decompose the error as

$$\|\boldsymbol{\psi}_{t+1}\|_2^2 \lesssim (1-\alpha)\|\boldsymbol{\psi}_t\|_2^2 + \alpha \underbrace{\boldsymbol{\psi}_t^\top\left[(\mathbf{r}_t(O_t) - \mathbf{J}_t^{\boldsymbol{\pi}_t}(O_t) + \mathbf{A}(O_t)\mathbf{Q}_t^{\boldsymbol{\pi}_t}) + \left(\mathbf{A}(O_t) - \bar{\mathbf{A}}^{\boldsymbol{\pi}_t, \mathbf{P}_t}\right)\boldsymbol{\psi}_t\right]}_{I_6:\text{Error due to Markov noise}}$$

$$+ \alpha \underbrace{(\mathbf{J}_t^{\boldsymbol{\pi}_t}(O_t) - \boldsymbol{\eta}_t(O_t))^2}_{I_7:\text{Avg. reward estimation error}} + \frac{1}{\alpha}\underbrace{\|\Pi_E\left[\mathbf{Q}_t^{\boldsymbol{\pi}_t} - \mathbf{Q}_{t+1}^{\boldsymbol{\pi}_{t+1}}\right]\|_2^2}_{I_8:\text{Value function drift}} + \underbrace{\alpha^2\|\mathbf{r}_t(O_t) - \boldsymbol{\eta}_t(O_t) + \mathbf{A}(O_t)\mathbf{Q}_t\|_2^2}_{I_9:\text{Variance term}}.$$

$$(10)$$

$I_6$ is the error induced by the Markovian noise which is analyzed leveraging the auxiliary Markov chain described below. $I_7$ describes the error due to an inaccurate estimation of the average reward which is bounded below. $I_8$, the change in the true value function is caused by drifting policies and environments, and can be neatly bounded in terms of the change in policy, rewards and transition probabilities (Lemma G.8). Finally, $I_{10}$ is the variance term.

**Bound on Markovian Noise.** Given time indices $t > \tau > 0$, consider the *auxiliary Markov chain* starting from $s_{t-\tau}$ constructed by conditioning on $\mathcal{F}_{t-\tau} = \{s_{t-\tau}, \boldsymbol{\pi}_{t-\tau-1}, \mathbf{P}_{t-\tau}\}$ and rolling out by applying $\boldsymbol{\pi}_{t-\tau-1}, \mathbf{P}_{t-\tau}$ as

$$s_{t-\tau} \xrightarrow{\boldsymbol{\pi}_{t-\tau-1}} a_{t-\tau} \xrightarrow{\mathbf{P}_{t-\tau}} \tilde{s}_{t-\tau+1} \xrightarrow{\boldsymbol{\pi}_{t-\tau-1}} \tilde{a}_{t-\tau+1} \xrightarrow{\cdots} \tilde{s}_t \xrightarrow{\boldsymbol{\pi}_{t-\tau-1}} \tilde{a}_t \xrightarrow{\mathbf{P}_{t-\tau}} \tilde{s}_{t+1} \xrightarrow{\boldsymbol{\pi}_{t-\tau-1}} \tilde{a}_{t+1}.$$

Recall that the *original Markov chain* is

$$s_{t-\tau} \xrightarrow{\boldsymbol{\pi}_{t-\tau-1}} a_{t-\tau} \xrightarrow{\mathbf{P}_{t-\tau}} s_{t-\tau+1} \xrightarrow{\boldsymbol{\pi}_{t-\tau}} a_{t-\tau+1} \xrightarrow{\cdots} s_t \xrightarrow{\boldsymbol{\pi}_{t-1}} a_t \xrightarrow{\mathbf{P}_t} s_{t+1} \xrightarrow{\boldsymbol{\pi}_t} a_{t+1}.$$

This method enables us to characterize properties of the original Markov chain in comparison to the auxiliary chain as $d_{TV}(P(O_t \in \cdot|\mathcal{F}_{t-\tau}), P(\tilde{O}_t \in \cdot|\mathcal{F}_{t-\tau}))$. We do this by bounding the effects of drifting policies and transition probabilities in the original chain and leveraging uniform ergodicity in the auxiliary chain. While prior works use auxiliary Markov chains for stationary environments (Zou et al., 2019; Wu et al., 2020; Wang et al., 2024), ours is the first adaptation to a non-stationary environment. Observe that the time-varying transition probabilities $\mathbf{P}_t$ add an extra layer of complexity, unlike the stationary case where only the policy changes over time.

**Average Reward Estimation Error (Proposition 4).** To bound $I_7$ in (10), i.e., the error in the average reward estimate $\phi_t = \eta_t - J_t^{\boldsymbol{\pi}_t}$, we can decompose the error as

$$\phi_{t+1}^2 \lesssim (1-\gamma)\phi_t^2 + \underbrace{\gamma(r_t(O_t) - J_t^{\boldsymbol{\pi}_t})^2}_{I_{10}:\text{Error due to Markov noise}} + \frac{1}{\gamma}\underbrace{(J_t^{\boldsymbol{\pi}_t} - J_{t+1}^{\boldsymbol{\pi}_{t+1}})^2}_{I_{11}:\text{Avg reward at consecutive time steps}} + \underbrace{\gamma^2(r_t(O_t) - \eta_t)^2}_{I_{12}:\text{Variance term}},$$

where $\gamma$ is the step-size (line 7, Algorithm 1). $I_{10}$ is the error induced by Markovian noise and is analyzed using the auxiliary Markov chain construction. $I_{11}$ quantifies the difference in average rewards at consecutive timesteps, and is neatly bounded in Lemma G.5 in terms of the corresponding changes in policies, rewards, and transition probabilities. $I_{12}$ is again the variance term.

Finally, $I_2$ in (9) characterizes the difference between the average reward and the instantaneous reward at any time, and is analyzed in Proposition 3 using the auxiliary Markov chain to bound the bias occurring due to Markovian sampling. This concludes the proof sketch.

---

[4] $\Pi_E[\mathbf{x}] = \arg\min_{\mathbf{y} \in E}\|\mathbf{x} - \mathbf{y}\|_2$ is the projection to $E$, the subspace orthogonal to the all ones vector $\mathbf{1}$.

# 6 SIMULATIONS

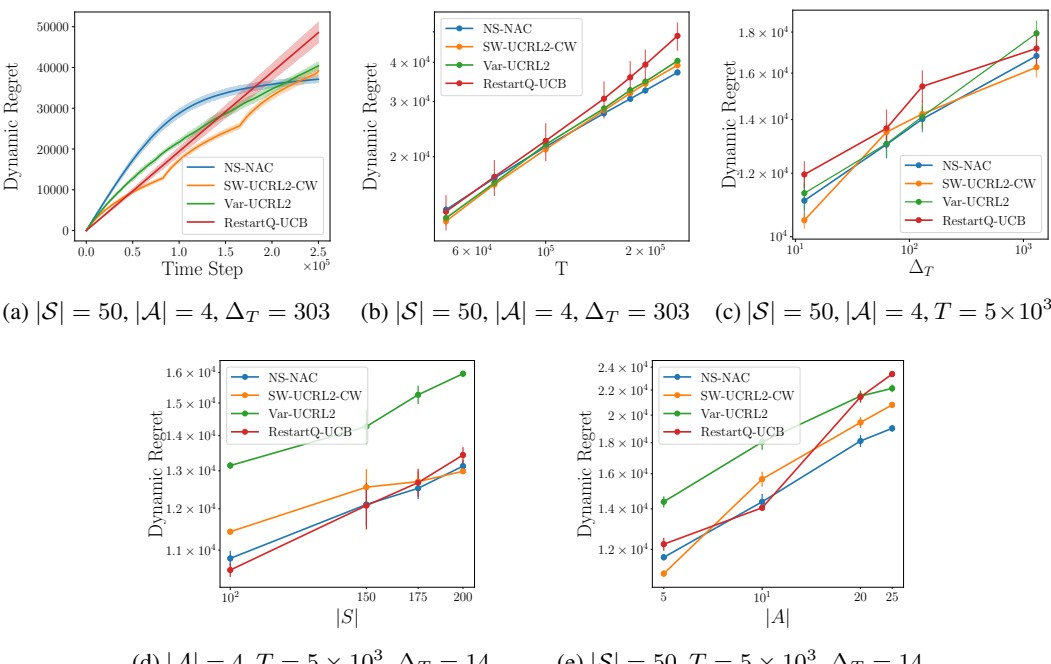

(a) $|\mathcal{S}| = 50, |\mathcal{A}| = 4, \Delta_T = 303$    (b) $|\mathcal{S}| = 50, |\mathcal{A}| = 4, \Delta_T = 303$    (c) $|\mathcal{S}| = 50, |\mathcal{A}| = 4, T = 5 \times 10^3$

(d) $|\mathcal{A}| = 4, T = 5 \times 10^3, \Delta_T = 14$    (e) $|\mathcal{S}| = 50, T = 5 \times 10^3, \Delta_T = 14$

Figure 1: Performance of NS-NAC and baseline algorithms across various settings. (a) Dynamic regret for a single instance with $T = 25 \times 10^4$ steps. Log-log plots showing the effect of varying: (b) time horizon $T$, (c) variation budget $\Delta_T$, (d) number of states $|\mathcal{S}|$, and (e) number of actions $|\mathcal{A}|$.

We empirically evaluate the performance of our algorithm, NS-NAC, on a synthetic non-stationary MDP (see Appendix K), comparing it with three baseline algorithms: SW-UCRL2-CW (Cheung et al. (2023)), Var-UCRL2 (Ortner et al. (2020)), and RestartQ-UCB (Mao et al. (2024)). SW-UCRL2-CW is a model-based algorithm that adapts to non-stationarity by maintaining a sliding window of recent observations, applying extended value iteration, and adjusting confidence intervals to track changing dynamics. Var-UCRL2, also model-based, adjusts its confidence intervals dynamically based on the observed variations in rewards and transitions. RestartQ-UCB, a model-free approach, periodically restarts Q-learning and resets its upper confidence bounds to adapt to non-stationarity. While there is a gap between our theoretical analysis of regret and those of the baseline methods, we observe in simulations that NS-NAC strongly matches their performance. We observe a consistent sub-linear dynamic regret across all experimental settings: varying time horizon $T$ (fig. 1b), variation budget $\Delta_T$ (fig. 1c), number of states $|\mathcal{S}|$ (fig. 1d), and number of actions $|\mathcal{A}|$ (fig. 1e).

# 7 CONCLUSION

We consider the problem of non-stationary reinforcement learning in the infinite-horizon average-reward setting and model it as a Markov decision process with time-varying rewards and transition probabilities. We analyze the first model-free policy-based algorithm, Non-Stationary Natural Actor-Critic (NS-NAC). It is a two-timescale natural policy gradient based method that utilizes learning rates as adapting factors and entropy based exploration to learn the time-varying optimal policy. NS-NAC achieves a dynamic regret that is sublinear in the time horizon thus theoretically validating policy gradient methods often used in practice in continual non-stationary RL. Directions for future work include designing parameter-free algorithms that do not require prior knowledge of the variation budget. Further, we believe a tighter regret bound can be obtained by a more refined analysis of the norm of the policy gradient using by leveraging the Fisher Information preconditioner.

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

CONTENTS

# A   NOTATION

**Variation Budgets**

$$\Delta_{R,T} = \sum_{t=0}^{T-1} \|\mathbf{r}_{t+1} - \mathbf{r}_t\|_\infty; \Delta_{R,t-\tau+1,t} = \sum_{i=t-\tau+1}^{t} \|\mathbf{r}_i - \mathbf{r}_{i-1}\|_\infty,$$

$$\Delta_{P,T} = \sum_{t=0}^{T-1} \|\mathbf{P}_{t+1} - \mathbf{P}_t\|_\infty; \Delta_{P,t-\tau+1,t} = \sum_{i=t-\tau+1}^{t} \|\mathbf{P}_i - \mathbf{P}_{i-1}\|_\infty,$$

$$\Delta_T = \Delta_{R,T} + \Delta_{P,T}.$$

The critic update (line 8 in Algorithm 1) can be defined in vector form using the following notation. Note that we use a one-to-one mapping $\sigma : \mathcal{S} \times \mathcal{A} \to \{1, 2, \dots, |\mathcal{S}||\mathcal{A}|\}$, to map state-action pairs $(s, a) \in \mathcal{S} \times \mathcal{A}$ to vector/matrix entries. However, for ease of notation, we denote the index of each entry by $(s, a)$, instead of the more accurate $\sigma(s, a)$.

$$O_t = (s_t, a_t, s_{t+1}, a_{t+1})$$

$$\mathbf{r}_t(O_t) = [0; \cdots; 0; r_t(s_t, a_t); 0; \cdots; 0]^\top \in \mathbb{R}^{|\mathcal{S}||\mathcal{A}|}$$

$$\eta_t(O_t) = [0; \cdots; 0; \eta_t; 0; \cdots; 0]^\top \in \mathbb{R}^{|\mathcal{S}||\mathcal{A}|}$$

$$\mathbf{J}_t^{\boldsymbol{\pi}}(O_t) = [0; \cdots; 0; J_t^{\boldsymbol{\pi}}; 0; \cdots; 0]^\top \in \mathbb{R}^{|\mathcal{S}||\mathcal{A}|}$$

$$\mathbf{A}(O) \in \mathbb{R}^{|\mathcal{S}||\mathcal{A}| \times |\mathcal{S}||\mathcal{A}|} \quad \text{such that}$$

$$\mathbf{A}(O)_{i,j} = \mathbf{A}(s, a, s', a')_{i,j} = \begin{cases} -1 & \text{if } (s,a) \neq (s',a'), i = j = (s,a) \\ 1 & \text{if } (s,a) \neq (s',a'), i = (s,a), j = (s',a') \\ 0 & \text{else} \end{cases}$$

As a result, we get the critic update

$$\mathbf{Q}_{t+1} = \prod_{R_Q} \left[ \mathbf{Q}_t + \alpha \left( \mathbf{r}_t(O_t) - \eta_t(O_t) + \mathbf{A}(O_t)\mathbf{Q}_t \right) \right].$$

For the purpose of analysis, we define the following quantities.

$$\bar{\mathbf{A}}^{\boldsymbol{\pi},\mathbf{P}} = \mathbb{E}_{s \sim d^{\boldsymbol{\pi},\mathbf{P}}(\cdot), a \sim \boldsymbol{\pi}(\cdot|s), s' \sim \mathbf{P}(\cdot|s,a), a' \sim \boldsymbol{\pi}(\cdot|s')} \left[ \mathbf{A}(s, a, s', a') \right]$$

$$\mathbf{Q}^{\boldsymbol{\pi},\mathbf{P},\mathbf{r}} = \mathbf{Q} \text{ associated with } \boldsymbol{\pi}, \mathbf{P}, \mathbf{r}$$

$$J^{\boldsymbol{\pi},\mathbf{P},\mathbf{r}} = \sum_s d^{\boldsymbol{\pi},\mathbf{P}}(s) \sum_a \boldsymbol{\pi}(a|s) r(s, a)$$

$$\Pi_E[\mathbf{x}] = \arg\min_{\mathbf{y} \in E} \|\mathbf{x} - \mathbf{y}\|_2 \text{ where } E \text{ is the subspace orthogonal to the all ones vector } \mathbf{1}$$

$$\boldsymbol{\psi}_t = \Pi_E \left[ \mathbf{Q}_t - \mathbf{Q}_t^{\boldsymbol{\pi}_t} \right] \qquad \text{(Error in the value-function estimate)}$$

$$\Gamma(\boldsymbol{\pi}, \mathbf{P}, \mathbf{r}, \boldsymbol{\psi}, O) = \boldsymbol{\psi}^\top \left( \mathbf{r}(O) - \mathbf{J}^{\boldsymbol{\pi},\mathbf{P},\mathbf{r}}(O) + \mathbf{A}(O)\mathbf{Q}^{\boldsymbol{\pi},\mathbf{P},\mathbf{r}} \right) + \boldsymbol{\psi}^\top \left( \mathbf{A}(O) - \bar{\mathbf{A}}^{\boldsymbol{\pi},\mathbf{P}} \right) \boldsymbol{\psi}$$

$$\phi_t = \eta_t - J_t^{\boldsymbol{\pi}_t} \qquad \text{(Error in the average reward estimate)}$$

$$\Lambda(\boldsymbol{\pi}, \mathbf{P}, \mathbf{r}, \eta, O) = (\eta - J^{\boldsymbol{\pi},\mathbf{P},\mathbf{r}})(r(s, a) - J^{\boldsymbol{\pi},\mathbf{P},\mathbf{r}})$$

Given time indices $t > \tau > 0$, consider the *auxiliary Markov chain* starting from $s_{t-\tau}$ constructed by conditioning on $s_{t-\tau}, \boldsymbol{\pi}_{t-\tau-1}, \mathbf{P}_{t-\tau}$ and rolling out by applying $\boldsymbol{\pi}_{t-\tau-1}, \mathbf{P}_{t-\tau}$ as

$$s_{t-\tau} \xrightarrow{\boldsymbol{\pi}_{t-\tau-1}} a_{t-\tau} \xrightarrow{\mathbf{P}_{t-\tau}} \tilde{s}_{t-\tau+1} \xrightarrow{\boldsymbol{\pi}_{t-\tau-1}} \tilde{a}_{t-\tau+1} \xrightarrow{\mathbf{P}_{t-\tau}} \dots \tilde{s}_t \xrightarrow{\boldsymbol{\pi}_{t-\tau-1}} \tilde{a}_t \xrightarrow{\mathbf{P}_{t-\tau}} \tilde{s}_{t+1} \xrightarrow{\boldsymbol{\pi}_{t-\tau-1}} \tilde{a}_{t+1}.$$

Recall that the *original Markov chain* is

$$s_{t-\tau} \xrightarrow{\boldsymbol{\pi}_{t-\tau-1}} a_{t-\tau} \xrightarrow{\mathbf{P}_{t-\tau}} s_{t-\tau+1} \xrightarrow{\boldsymbol{\pi}_{t-\tau}} a_{t-\tau+1} \xrightarrow{\mathbf{P}_{t-\tau+1}} \dots s_t \xrightarrow{\boldsymbol{\pi}_{t-1}} a_t \xrightarrow{\mathbf{P}_t} s_{t+1} \xrightarrow{\boldsymbol{\pi}_t} a_{t+1}.$$

## B  SYMBOL REFERENCE

| Constant | First Appearance |
|---|---|
| $U_R$ | Section 3.1 |
| $U_Q$ | Lemma I.3 |
| $E$ = subspace orthogonal to all ones vector $\mathbf{1}$ | Algorithm 1 |
| $C = \inf_{s,t,t',\pi} \frac{d^{\pi,\mathbf{P}_{t'}}(s)}{d^{\pi_t^\star,\mathbf{P}_t}(s)}$ | Assumption 1 |
| $m, \rho$ | Assumption 1 |
| $M = \lceil \log_\rho m^{-1} \rceil + \frac{1}{1-\rho}$ | Lemma I.1 |
| $\lambda$ | Lemma 5.1 |
| $\xi = \max_{t \in [T]} D_{\mathrm{KL}}\left(\pi_t^\star(\cdot|s) \| \pi_t(\cdot|s)\right)$ | Proposition 1 |
| $W_1 = (3G_R^2)^{1/3}(4U_Q^2)^{2/3}$ | Proposition 2 |
| $W_2 = (3G_P^2)^{1/3}(4U_Q^2)^{2/3}$ | Proposition 2 |
| $D_1 = L_\pi B_2 + 4U_R\sqrt{|\mathcal{S}||\mathcal{A}|}B_2 + 4U_R$ | Proposition 3 |
| $D_2 = 4U_R + L_P$ | Proposition 3 |
| $D_3 = 4U_R F_6 + 8U_R^2$ | Proposition 4 |
| $D_4 = 9L_\pi^2 B_2^2$ | Proposition 4 |
| $W_3 = (3)^{1/3}(4U_R^2)^{2/3}$ | Proposition 4 |
| $W_4 = (3L_P^2)^{1/3}(4U_R^2)^{2/3}$ | Proposition 4 |
| $B_1 = 4\sqrt{|\mathcal{A}|}U_Q^2$ | Lemma G.1 |
| $B_2 = 2U_Q$ | Lemma G.2 |
| $B_3 = (F_{1\pi} + G_\pi + F_3\sqrt{|\mathcal{S}||\mathcal{A}|} + F_4)B_2$ | Lemma G.9 |
| $B_4 = F_2(2U_R + 2U_Q)$ | Lemma G.9 |
| $B_5 = F_2 G_R$ | Lemma G.9 |
| $B_6 = F_{1\mathbf{P}} + F_2 G_P + F_3$ | Lemma G.9 |
| $B_7 = (F_5 L_\pi + F_7\sqrt{|\mathcal{S}||\mathcal{A}|} + F_8)B_2$ | Lemma G.10 |
| $B_8 = F_7 + F_5 L_P$ | Lemma G.10 |
| $L_\pi = 4U_R(M+1)\sqrt{|\mathcal{S}||\mathcal{A}|}$ | Lemma G.5 |
| $L_P = 4U_R M$ | Lemma G.5 |
| $G_\pi = 2U_Q\sqrt{|\mathcal{S}||\mathcal{A}|}$ | Lemma G.6 |
| $G_R = 2\lambda^{-1}\sqrt{|\mathcal{S}||\mathcal{A}|}$ | Lemma G.7 |
| $G_P = (\lambda^{-1}L_P + 4U_R\lambda^{-1}M + 4U_R\lambda^{-2}(M+1))\sqrt{|\mathcal{S}||\mathcal{A}|}$ | Lemma G.7 |
| $F_{1\pi} = 2U_Q L_\pi + 4U_Q G_\pi + 8U_Q^2(M+2)|\mathcal{S}||\mathcal{A}|$ | Lemma H.3 |
| $F_{1P} = 2U_Q L_P + 4U_Q G_P + 8U_Q^2(M+1)\sqrt{|\mathcal{S}||\mathcal{A}|}$ | Lemma H.3 |
| $F_2 = 2U_R + 18U_Q$ | Lemma H.4 |
| $F_3 = 16U_R U_Q + 24U_Q^2\sqrt{|\mathcal{S}||\mathcal{A}|}$ | Lemma H.5 |
| $F_4 = 8U_R U_Q + 24U_Q^2\sqrt{|\mathcal{S}||\mathcal{A}|}$ | Lemma H.6 |
| $F_5 = 4U_R$ | Lemma H.8 |
| $F_6 = 2U_R$ | Lemma H.9 |
| $F_7 = 8U_R^2$ | Lemma H.10 |
| $F_8 = 8U_R^2$ | Lemma H.11 |

## C    REGRET ANALYSIS

**Theorem 1.** *If Assumption 1 is satisfied and the step-sizes and exploration are chosen as $0 < \alpha, \beta, \gamma, \epsilon < 1/2$ in Algorithm 1, then we have*

$$Dyn\text{-}Reg(\mathcal{M}, T) = \mathbb{E}\left[\sum_{t=0}^{T-1} J_t^{\boldsymbol{\pi}_t^\star} - r_t(s_t, a_t)\right]$$

$$\leq \underbrace{\tilde{\mathcal{O}}\left(\frac{N}{\beta\epsilon}\right) + \tilde{\mathcal{O}}\left(\epsilon T\right) + \tilde{\mathcal{O}}\left(\sqrt{\frac{T}{\alpha}}\right)}_{\text{Effect of initialization and exploration}} + \underbrace{\tilde{\mathcal{O}}\left(\frac{\beta T}{\alpha}\right) + \tilde{\mathcal{O}}\left(T\sqrt{\beta}\right)}_{\substack{\text{Bounds cumulative change} \\ \text{in policy over horizon } T}} + \underbrace{\tilde{\mathcal{O}}\left(\frac{\beta T}{\gamma}\right) + \tilde{\mathcal{O}}\left(T\sqrt{\gamma}\right) + \tilde{\mathcal{O}}\left(\sqrt{\frac{T}{\gamma}}\right)}_{\text{Error in Average Reward Estimate } (\eta_t) \text{ at Critic}}$$

$$+ \underbrace{\tilde{\mathcal{O}}\left(T\sqrt{\alpha}\right)}_{\substack{\text{Bounds cumulative} \\ \text{change in critic estimates}}} + \underbrace{\tilde{\mathcal{O}}\left(\frac{\Delta_T T}{N}\right) + \tilde{\mathcal{O}}\left(\Delta_T\right) + \tilde{\mathcal{O}}\left(\sqrt{\Delta_T T}\right) + \tilde{\mathcal{O}}\left(\Delta_T^{1/3} T^{2/3}\left(\frac{1}{\alpha} + \frac{1}{\gamma}\right)\right)}_{\text{Error due to Non-Stationarity}},$$

*where $\Delta_T = \Delta_{R,T} + \Delta_{P,T}$, $\tilde{\mathcal{O}}(\cdot)$ hides the constants and logarithmic dependence on the time horizon $T$, and $N$ is a parameter in the analysis which divides the total horizon $T$ into $N$ segments of equal length. Our results hold for any $1 \leq N \leq T$ and when $N$, together with $\alpha, \beta, \gamma, \epsilon$, are optimized, we get $\alpha^\star = \gamma^\star = \left(\frac{\Delta_T}{T}\right)^{2/9}$, $\beta^\star = \left(\frac{\Delta_T}{T}\right)^{3/9}$, $N^\star = \Delta_T^{8/9} T^{1/9}$, $\epsilon^\star = \left(\frac{\Delta_T}{T}\right)^{4/9}$. The resulting regret (with explicit dependence on the size of the state-action space $|\mathcal{S}|, |\mathcal{A}|$) is*

$$Dyn\text{-}Reg(\mathcal{M}, T) \leq \tilde{\mathcal{O}}\left(|\mathcal{S}|^{\frac{1}{2}} |\mathcal{A}|^{\frac{1}{2}} \Delta_T^{\frac{1}{9}} T^{\frac{8}{9}}\right).$$

*Proof.* We start by dividing the total horizon $T$ into $N$ segments of length $T_0$ each i.e. $N = \lceil\frac{T}{T_0}\rceil$, which enables analysis in the non-stationary environment. For simplicity, we assume $T = kT_0$, for some positive integer $k$. Decomposing the regret as follows, we have

$$\mathbb{E}\left[\sum_{t=0}^{T-1} J_t^{\boldsymbol{\pi}_t^\star} - r_t(s_t, a_t)\right]$$

$$\leq 2U_R \tau_T + \mathbb{E}\left[\sum_{t=\tau_T}^{T-1} J_t^{\boldsymbol{\pi}_t^\star} - J_t^{\boldsymbol{\pi}_t}\right] + \mathbb{E}\left[\sum_{t=\tau_T}^{T-1} J_t^{\boldsymbol{\pi}_t} - r_t(s_t, a_t)\right]$$

$$\overset{(a)}{\leq} \tilde{\mathcal{O}}\left(1\right) + \tilde{\mathcal{O}}\left(\frac{\Delta_T T}{N}\right) + \tilde{\mathcal{O}}\left(\frac{N}{\beta\epsilon}\right) + \tilde{\mathcal{O}}\left(\epsilon T\right) + \tilde{\mathcal{O}}\left(\beta T\right)$$

$$+ 2\sum_{t=\tau_T}^{T-1} \mathbb{E}\left[\|\mathbf{Q}_t^{\boldsymbol{\pi}_t} - \mathbf{Q}_t\|_\infty\right] + \mathbb{E}\left[\sum_{t=\tau_T}^{T-1} J_t^{\boldsymbol{\pi}_t} - r_t(s_t, a_t)\right]$$

$$\overset{(b)}{\leq} \tilde{\mathcal{O}}\left(1\right) + \tilde{\mathcal{O}}\left(\frac{\Delta_T T}{N}\right) + \tilde{\mathcal{O}}\left(\frac{N}{\beta\epsilon}\right) + \tilde{\mathcal{O}}\left(\epsilon T\right) + \tilde{\mathcal{O}}\left(\beta T\right)$$

$$+ \tilde{\mathcal{O}}\left(\sqrt{\frac{T}{\alpha}}\right) + \tilde{\mathcal{O}}\left(\sqrt{\alpha}T\right) + \tilde{\mathcal{O}}\left(\frac{\beta T}{\alpha}\right) + \tilde{\mathcal{O}}\left(\sqrt{\beta}T\right) + \tilde{\mathcal{O}}\left(\frac{\beta T}{\gamma}\right) + \tilde{\mathcal{O}}\left(\sqrt{\gamma}T\right) + \tilde{\mathcal{O}}\left(\sqrt{\frac{T}{\gamma}}\right)$$

$$+ \tilde{\mathcal{O}}\left(\sqrt{\Delta_T T}\right) + \tilde{\mathcal{O}}\left(\frac{\Delta_T^{1/3} T^{2/3}}{\alpha}\right) + \tilde{\mathcal{O}}\left(\frac{\Delta_T^{1/3} T^{2/3}}{\gamma}\right) + \mathbb{E}\left[\sum_{t=\tau_T}^{T-1} J_t^{\boldsymbol{\pi}_t} - r_t(s_t, a_t)\right]$$

$$\overset{(c)}{\leq} \tilde{\mathcal{O}}\left(1\right) + \tilde{\mathcal{O}}\left(\frac{\Delta_T T}{N}\right) + \tilde{\mathcal{O}}\left(\frac{N}{\beta\epsilon}\right) + \tilde{\mathcal{O}}\left(\epsilon T\right) + \tilde{\mathcal{O}}\left(\beta T\right)$$

$$+ \tilde{\mathcal{O}}\left(\sqrt{\frac{T}{\alpha}}\right) + \tilde{\mathcal{O}}\left(\sqrt{\alpha}T\right) + \tilde{\mathcal{O}}\left(\frac{\beta T}{\alpha}\right) + \tilde{\mathcal{O}}\left(\sqrt{\beta}T\right) + \tilde{\mathcal{O}}\left(\frac{\beta T}{\gamma}\right) + \tilde{\mathcal{O}}\left(\sqrt{\gamma}T\right) + \tilde{\mathcal{O}}\left(\sqrt{\frac{T}{\gamma}}\right)$$

$$+ \tilde{\mathcal{O}}\left(\sqrt{\Delta_T T}\right) + \tilde{\mathcal{O}}\left(\frac{\Delta_T^{1/3} T^{2/3}}{\alpha}\right) + \tilde{\mathcal{O}}\left(\frac{\Delta_T^{1/3} T^{2/3}}{\gamma}\right) + \tilde{\mathcal{O}}\left(\beta T\right) + \tilde{\mathcal{O}}\left(\Delta_{P,T}\right)$$

$$\leq \tilde{\mathcal{O}}\left(T\sqrt{\alpha}\right) + \tilde{\mathcal{O}}\left(\sqrt{\frac{T}{\alpha}}\right) + \tilde{\mathcal{O}}\left(\frac{\beta T}{\alpha}\right) + \tilde{\mathcal{O}}\left(T\sqrt{\beta}\right) + \tilde{\mathcal{O}}\left(\frac{\beta T}{\gamma}\right) + \tilde{\mathcal{O}}\left(T\sqrt{\gamma}\right) + \tilde{\mathcal{O}}\left(\sqrt{\frac{T}{\gamma}}\right)$$

$$+ \tilde{\mathcal{O}}\left(\frac{\Delta_T T}{N}\right) + \tilde{\mathcal{O}}\left(\frac{N}{\beta\epsilon}\right) + \tilde{\mathcal{O}}(\epsilon T) + \tilde{\mathcal{O}}\left(\Delta_T\right) + \tilde{\mathcal{O}}\left(\sqrt{\Delta_T T}\right) + \tilde{\mathcal{O}}\left(\frac{\Delta_T^{1/3} T^{2/3}}{\alpha}\right) + \tilde{\mathcal{O}}\left(\frac{\Delta_T^{1/3} T^{2/3}}{\gamma}\right),$$

where $(a)$ is due to Proposition 1, $(b)$ is because $\sum \mathbb{E}[\|\mathbf{Q}_t^{\boldsymbol{\pi}_t} - \mathbf{Q}_t\|_\infty] \leq \sum \mathbb{E}[\|\mathbf{Q}_t^{\boldsymbol{\pi}_t} - \mathbf{Q}_t\|_2] \leq T^{1/2}\left(\sum \mathbb{E}[\|\mathbf{Q}_t^{\boldsymbol{\pi}_t} - \mathbf{Q}_t\|_2^2]\right)^{1/2}$ and Proposition 2, $(c)$ is by Proposition 3 and $\Delta_T = \Delta_{R,T} + \Delta_{P,T}$. We further have $\tau_T = \mathcal{O}(\log T)$. Note that $\tilde{\mathcal{O}}(\cdot)$ hides constants and logarithmic terms. $\square$

# D ACTOR

The next result bounds the cumulative difference between the average rewards of the optimal policy $\boldsymbol{\pi}_t^\star$ and that of the current policy $\boldsymbol{\pi}_t$, in the environment $\mathcal{M}_t$.

**Proposition 1.** *If Asumption 1 holds, we have*

$$\mathbb{E}\left[\sum_{t=0}^{T-1} J_t^{\boldsymbol{\pi}_t^\star} - J_t^{\boldsymbol{\pi}_t}\right] \leq \underbrace{\left(2 + 2G_R + \frac{1}{C}\right)\frac{T\Delta_{R,T}}{N} + \left(U_Q + L_P + 2G_P + \frac{L_P}{C}\right)\frac{T\Delta_{P,T}}{N}}_{\text{Error due to Non-Stationarity}}$$

$$+ \underbrace{2\sum_{t=0}^{T-1}\mathbb{E}\left[\|\Pi_E\left[\mathbf{Q}_t^{\boldsymbol{\pi}_t} - \mathbf{Q}_t\right]\|_\infty\right]}_{\text{Critic Estimation Error}} + \underbrace{N\cdot\frac{\log|\mathcal{A}|}{\beta}}_{\substack{N\times\text{Bias of}\\\text{Initialization}}} + \underbrace{\frac{2U_Q N}{\beta\epsilon} + \epsilon T\log|\mathcal{A}| +}_{\text{Effect of exploration}} \underbrace{\frac{B_1\beta T}{C}}_{\substack{\text{Bounds cumulative}\\\text{change in policy}}} + \underbrace{\frac{U_R}{C}}_{\text{constant}},$$

*where* $\Delta_{R,T} = \sum_{t=0}^{T-1}\|\mathbf{r}_{r+1} - \mathbf{r}_t\|_\infty$, $\Delta_{P,T} = \sum_{t=0}^{T-1}\|\mathbf{P}_{t+1} - \mathbf{P}_t\|_\infty$, $C$ *is defined in Assumption 1, and the total horizon $T$ is divided into $N$ segments of equal length. The remaining constants are defined in Appendix B.*

**Remark.** *Note that $N$ is an artifact of the proof and does not affect the algorithm. In Theorem 1, we choose $N$ to optimize the regret upper bound.*

*Proof.* We start by dividing the total horizon $T$ into $N$ segments of length $T_0$ each i.e. $N = \lceil\frac{T}{T_0}\rceil$. For simplicity, we assume $T = kT_0$, for some positive integer $k$. In each segment (indexed by $n \in [N]$), we use $J_{nT_0}^{\boldsymbol{\pi}_{nT_0}^\star}$ as an anchor against which to compare the performance of the learned policies.

$$\mathbb{E}\left[\sum_{t=0}^{T-1} J_t^{\boldsymbol{\pi}_t^\star} - J_t^{\boldsymbol{\pi}_t}\right] \leq \mathbb{E}\left[\sum_{n=0}^{N-1}\sum_{j=nT_0}^{(n+1)T_0-1}\left(J_j^{\boldsymbol{\pi}_j^\star} - J_{nT_0}^{\boldsymbol{\pi}_{nT_0}^\star}\right) + \left(J_{nT_0}^{\boldsymbol{\pi}_{nT_0}^\star} - J_{nT_0}^{\boldsymbol{\pi}_j}\right) + \left(J_{nT_0}^{\boldsymbol{\pi}_j} - J_j^{\boldsymbol{\pi}_j}\right)\right]$$

$$\overset{(a)}{\leq} \mathbb{E}\left[\sum_{n=0}^{N-1}\sum_{j=nT_0}^{(n+1)T_0-1}\left(2\|\mathbf{r}_j - \mathbf{r}_{nT_0}\|_\infty + (U_Q + L_P)\|\mathbf{P}_j - \mathbf{P}_{nT_0}\|_\infty\right) + \left(J_{nT_0}^{\boldsymbol{\pi}_{nT_0}^\star} - J_{nT_0}^{\boldsymbol{\pi}_j}\right)\right]$$

$$\overset{(b)}{\leq} \sum_{n=0}^{N-1}\sum_{j=nT_0+1}^{(n+1)T_0-1}(T_0 - 1)\left(2\|\mathbf{r}_j - \mathbf{r}_{j-1}\|_\infty + (U_Q + L_P)\|\mathbf{P}_j - \mathbf{P}_{j-1}\|_\infty\right)$$

$$+ \mathbb{E}\left[\sum_n\sum_j\left(J_{nT_0}^{\boldsymbol{\pi}_{nT_0}^\star} - J_{nT_0}^{\boldsymbol{\pi}_j}\right)\right]$$

$$\leq (T_0 - 1)\left(2\Delta_{R,T} + (U_Q + L_P)\Delta_{P,T}\right) + \mathbb{E}\left[\sum_n\sum_j J_{nT_0}^{\boldsymbol{\pi}_{nT_0}^\star} - J_{nT_0}^{\boldsymbol{\pi}_j}\right], \tag{11}$$

where $(a)$ is by Lemma G.4 and Lemma G.5 and $(b)$ is by triangle inequality. We now bound the last term as

$$\sum_{n=0}^{N-1} \sum_{j=nT_0}^{(n+1)T_0-1} J_{nT_0}^{\boldsymbol{\pi}_{nT_0}^\star} - J_{nT_0}^{\boldsymbol{\pi}_j}$$

$$\overset{(c)}{=} \sum_{n=0}^{N-1} \sum_{j=nT_0}^{(n+1)T_0-1} \frac{1}{\beta} \sum_s \sum_a d^{\boldsymbol{\pi}_{nT_0}^\star, \mathbf{P}_{nT_0}}(s) \boldsymbol{\pi}_{nT_0}^\star(a|s) \left[\beta Q_{nT_0}^{\boldsymbol{\pi}_j}(s,a) - \beta V_{nT_0}^{\boldsymbol{\pi}_j}(s)\right]$$

$$= \sum_n \sum_j \frac{1}{\beta} \sum_s \sum_a d^{\boldsymbol{\pi}_{nT_0}^\star, \mathbf{P}_{nT_0}}(s) \boldsymbol{\pi}_{nT_0}^\star(a|s) \left[\beta Q_j^{\boldsymbol{\pi}_j}(s,a) - \beta V_j^{\boldsymbol{\pi}_j}(s) + \beta Q_j(s,a) - \beta Q_j(s,a)\right]$$

$$+ \sum_n \sum_j \frac{1}{\beta} \sum_s \sum_a d^{\boldsymbol{\pi}_{nT_0}^\star, \mathbf{P}_{nT_0}}(s) \boldsymbol{\pi}_{nT_0}^\star(a|s) \left[\beta Q_{nT_0}^{\boldsymbol{\pi}_j}(s,a) - \beta Q_j^{\boldsymbol{\pi}_j}(s,a) + \beta V_j^{\boldsymbol{\pi}_j}(s) - \beta V_{nT_0}^{\boldsymbol{\pi}_j}(s)\right]$$

$$= \sum_n \sum_j \frac{1}{\beta} \sum_s \sum_a d^{\boldsymbol{\pi}_{nT_0}^\star, \mathbf{P}_{nT_0}}(s) \boldsymbol{\pi}_{nT_0}^\star(a|s) \left[\beta Q_j^{\boldsymbol{\pi}_j}(s,a) - \beta V_j^{\boldsymbol{\pi}_j}(s) + \beta Q_j(s,a) - \beta Q_j(s,a)\right]$$

$$+ \sum_n \sum_j 2\|\mathbf{Q}_{nT_0}^{\boldsymbol{\pi}_j} - \mathbf{Q}_j^{\boldsymbol{\pi}_j}\|_\infty$$

$$\overset{(d)}{\leq} \sum_n \sum_j \frac{1}{\beta} \sum_s \sum_a d^{\boldsymbol{\pi}_{nT_0}^\star, \mathbf{P}_{nT_0}}(s) \boldsymbol{\pi}_{nT_0}^\star(a|s) \left[\beta Q_j^{\boldsymbol{\pi}_j}(s,a) - \beta V_j^{\boldsymbol{\pi}_j}(s) + \beta Q_j(s,a) - \beta Q_j(s,a)\right]$$

$$+ \sum_{n=0}^{N-1} \sum_{j=nT_0}^{(n+1)T_0-1} 2G_R\|\mathbf{r}_{nT_0} - \mathbf{r}_j\|_\infty + 2G_P\|\mathbf{P}_{nT_0} - \mathbf{P}_j\|_\infty$$

$$\overset{(e)}{=} \sum_n \sum_j \frac{1}{\beta} \sum_s \sum_a d^{\boldsymbol{\pi}_{nT_0}^\star, \mathbf{P}_{nT_0}}(s) \boldsymbol{\pi}_{nT_0}^\star(a|s) \left[\underbrace{\log Z_j(s) - \beta V_j^{\boldsymbol{\pi}_j}(s)}_{I_1}\right]$$

$$+ \sum_n \sum_j \frac{1}{\beta} \sum_s \sum_a d^{\boldsymbol{\pi}_{nT_0}^\star, \mathbf{P}_{nT_0}}(s) \boldsymbol{\pi}_{nT_0}^\star(a|s) \left[\underbrace{\log \frac{\pi_{j+1}(a|s)}{(\pi_j(a|s))^{1-\beta\epsilon}}}_{I_2} + \underbrace{\beta Q_j^{\boldsymbol{\pi}_j}(s,a) - \beta Q_j(s,a)}_{I_3}\right]$$

$$+ (T_0-1)(2G_R\Delta_{R,T} + 2G_P\Delta_{P,T}) \tag{12}$$

where $(c)$ follows from the Performance Difference Lemma G.3, $(d)$ follows from Lemma G.7 and $(e)$ from the actor update equation (line 10 in Algorithm 1) and $Z_t(s) = \sum_{a'\in\mathcal{A}}(\pi_t(a'|s))^{1-\beta\epsilon}\exp(\beta Q_t(s,a'))$. Next, we bound each of $I_1, I_2, I_3$. Using Lemma G.1, we have

$$I_1 = \sum_n \sum_j \sum_s d^{\boldsymbol{\pi}_{nT_0}^\star, \mathbf{P}_{nT_0}}(s) \left[\frac{\log Z_j(s)}{\beta} - V_j^{\boldsymbol{\pi}_j}(s)\right] \underbrace{\sum_a \pi_{nT_0}^\star(a|s)}_{=1}$$

$$\leq \sum_n \sum_j \left[\frac{J_{j+1}^{\boldsymbol{\pi}_{j+1}} - J_j^{\boldsymbol{\pi}_j}}{C} + \epsilon \log|\mathcal{A}| + \|\mathbf{Q}_j^{\boldsymbol{\pi}_j} - \mathbf{Q}_j\|_\infty + \frac{B_1\beta}{C} + \frac{\|\mathbf{r}_{j+1} - \mathbf{r}_j\|_\infty}{C} + \frac{L_P\|\mathbf{P}_{j+1} - \mathbf{P}_j\|_\infty}{C}\right]. \tag{13}$$

Next, we establish a bound on $I_2$ as

$$I_2 = \frac{1}{\beta} \sum_n \sum_j \sum_s \sum_a d^{\boldsymbol{\pi}_{nT_0}^\star, \mathbf{P}_{nT_0}}(s) \boldsymbol{\pi}_{nT_0}^\star(a|s) \log \frac{\pi_{j+1}(a|s)}{(\pi_j(a|s))^{1-\beta\epsilon}}$$

$$\leq \frac{1}{\beta} \sum_{n=0}^{N-1} \sum_{j=nT_0}^{(n+1)T_0-1} \sum_s d^{\boldsymbol{\pi}_{nT_0}^\star, \mathbf{P}_{nT_0}}(s) \left[D_{\mathrm{KL}}\left(\boldsymbol{\pi}_{nT_0}^\star(\cdot|s)\|\boldsymbol{\pi}_j(\cdot|s)\right) - D_{\mathrm{KL}}\left(\boldsymbol{\pi}_{nT_0}^\star(\cdot|s)\|\boldsymbol{\pi}_{j+1}(\cdot|s)\right)\right]$$

$$= \frac{1}{\beta} \sum_n \sum_s d^{\boldsymbol{\pi}_{nT_0}^\star, \mathbf{P}_{nT_0}}(s) \left[D_{\mathrm{KL}}\left(\boldsymbol{\pi}_{nT_0}^\star(\cdot|s)\|\boldsymbol{\pi}_{nT_0}(\cdot|s)\right) - D_{\mathrm{KL}}\left(\boldsymbol{\pi}_{nT_0}^\star(\cdot|s)\|\boldsymbol{\pi}_{(n+1)T_0}(\cdot|s)\right)\right]$$

$$\overset{(f)}{\leq} \frac{1}{\beta} \sum_n \sum_s d^{\boldsymbol{\pi}^\star_{nT_0}, \mathbf{P}_{nT_0}}(s) D_{\mathrm{KL}}\left(\boldsymbol{\pi}^\star_{nT_0}(\cdot|s) \| \boldsymbol{\pi}_{nT_0}(\cdot|s)\right)$$

$$\overset{(g)}{\leq} \frac{1}{\beta} \sum_n \sum_s d^{\boldsymbol{\pi}^\star_{nT_0}, \mathbf{P}_{nT_0}}(s) \log \frac{|\mathcal{A}|}{e^{-2U_Q/\epsilon}} \leq \frac{N \log |\mathcal{A}|}{\beta} + \frac{2U_Q N}{\beta \epsilon} \tag{14}$$

where $(f)$ is because $\sum_{n,j,s,a} d^{\boldsymbol{\pi}^\star_{nT_0}, \mathbf{P}_{nT_0}} \boldsymbol{\pi}^\star_{nT_0}(a|s) \beta \epsilon \log \pi_j(a|s) < 0$ and non-negativity of KL-divergence and $(g)$ is due to Lemma H.1. Lastly, $I_3$ can be bounded as

$$I_3 = \sum_n \sum_j \sum_s \sum_a d^{\boldsymbol{\pi}^\star_{nT_0}, \mathbf{P}_{nT_0}}(s) \boldsymbol{\pi}^\star_{nT_0}(a|s) \left[Q_j^{\boldsymbol{\pi}_j}(s,a) - Q_j(s,a)\right]$$

$$\leq \sum_n \sum_j \|\mathbf{Q}_j^{\boldsymbol{\pi}_j} - \mathbf{Q}_j\|_\infty. \tag{15}$$

We substitute the bounds on $I_1, I_2, I_3$ from (13)-(15) in (12) and then combine with (11). Recall that the set of solutions to the Bellman equations is $\mathbf{Q}_t^{\boldsymbol{\pi}_t} = \{\mathbf{Q}_{t,E}^{\boldsymbol{\pi}_t} + c\mathbf{1} | \mathbf{Q}_{t,E}^{\boldsymbol{\pi}_t} \in E, c \in \mathbb{R}\}$ where E is the subspace orthogonal to the all ones vector and $\mathbf{Q}_{t,E}^{\boldsymbol{\pi}_t}$ is the unique solution in $E$ (Zhang et al., 2021b). Finally, we use the equivalence $\|\mathbf{Q}_t^{\boldsymbol{\pi}_t} - \mathbf{Q}_t\|_\infty = \|\Pi_E\left[\mathbf{Q}_t^{\boldsymbol{\pi}_t} - \mathbf{Q}_t\right]\|_\infty$ to get the result. $\square$

## E   CRITIC

**Proposition 2.** *If Assumption 1 is satisfied and $0 < \gamma < 1/2$, then we have*

$$\sum_{t=\tau_T}^{T-1} \mathbb{E}\left[\|\Pi_E\left[\mathbf{Q}_t - \mathbf{Q}_t^{\boldsymbol{\pi}_t}\right]\|_2^2\right] \leq \underbrace{\tilde{\mathcal{O}}\left(\frac{1}{\alpha}\right)}_{\substack{\text{Effect of} \\ \text{initialization}}} + \underbrace{\tilde{\mathcal{O}}\left(\Delta_{R,T}\right) + \tilde{\mathcal{O}}\left(\Delta_{P,T}\right) + \tilde{\mathcal{O}}\left(\frac{\Delta_{R,T}^{2/3} T^{1/3}}{\alpha^2}\right) + \tilde{\mathcal{O}}\left(\frac{\Delta_{P,T}^{2/3} T^{1/3}}{\alpha^2}\right)}_{\text{Error due to Non-Stationarity}}$$

$$+ \underbrace{\tilde{\mathcal{O}}\left(\gamma T\right) + \tilde{\mathcal{O}}\left(\frac{1}{\gamma}\right) + \tilde{\mathcal{O}}\left(\frac{\beta^2 T}{\gamma^2}\right) + \tilde{\mathcal{O}}\left(\frac{\Delta_{R,T}^{2/3} T^{1/3}}{\gamma^2}\right) + \tilde{\mathcal{O}}\left(\frac{\Delta_{P,T}^{2/3} T^{1/3}}{\gamma^2}\right)}_{\text{Error in Average Reward Estimate } (\eta_t) \text{ at Critic}}$$

$$+ \underbrace{\tilde{\mathcal{O}}\left(\beta T\right) + \tilde{\mathcal{O}}\left(\frac{\beta^2 T}{\alpha^2}\right)}_{\substack{\text{Bounds cumulative change} \\ \text{in policy over horizon } T}} + \underbrace{\tilde{\mathcal{O}}\left(\alpha T\right)}_{\substack{\text{Bounds cumulative} \\ \text{change in critic estimates}}},$$

*where $\tilde{\mathcal{O}}(\cdot)$ hides constants and logarithmic terms which can be found in Equation (19) and $\Delta_{R,T} = \sum_{t=0}^{T-1} \|\mathbf{r}_{t+1} - \mathbf{r}_t\|_\infty$, and $\Delta_{P,T} = \sum_{t=0}^{T-1} \|\mathbf{P}_{t+1} - \mathbf{P}_t\|_\infty$.*

*Proof.* Recall that $\boldsymbol{\psi}_t = \Pi_E\left[\mathbf{Q}_t - \mathbf{Q}_t^{\boldsymbol{\pi}_t}\right]$, E is the subspace orthogonal to the all ones vector $\mathbf{1}$ and the critic update equation (line 8 in Algorithm 1) can be expressed in vector form as $\mathbf{Q}_{t+1} = \Pi_{R_Q}\left[\mathbf{Q}_t + \alpha\left(\mathbf{r}_t(O_t) - \boldsymbol{\eta}_t(O_t) + \mathbf{A}(O_t)\mathbf{Q}_t\right)\right]$. Recall the notations $\mathbf{r}_t, \eta_t, \mathbf{A}(O_t), \bar{\mathbf{A}}^{\boldsymbol{\pi}_t, \mathbf{P}_t}, \mathbf{J}_t(O_t), \Gamma(\cdot), \phi_t$ from Appendix A. We therefore have

$$\|\boldsymbol{\psi}_{t+1}\|_2^2 = \|\Pi_E\left[\mathbf{Q}_{t+1} - \mathbf{Q}_{t+1}^{\boldsymbol{\pi}_{t+1}}\right]\|_2^2$$

$$\leq \|\Pi_E\left[\mathbf{Q}_t + \alpha\left(\mathbf{r}_t(O_t) - \boldsymbol{\eta}_t(O_t) + \mathbf{A}(O_t)\mathbf{Q}_t\right) - \mathbf{Q}_{t+1}^{\boldsymbol{\pi}_{t+1}}\right]\|_2^2$$

$$= \|\Pi_E\left[\boldsymbol{\psi}_t + \alpha\left(\mathbf{r}_t(O_t) - \boldsymbol{\eta}_t(O_t) + \mathbf{A}(O_t)\mathbf{Q}_t\right) + \mathbf{Q}_t^{\boldsymbol{\pi}_t} - \mathbf{Q}_{t+1}^{\boldsymbol{\pi}_{t+1}}\right]\|_2^2$$

$$\leq \|\boldsymbol{\psi}_t\|_2^2 + 2\alpha\boldsymbol{\psi}_t^\top\left(\mathbf{r}_t(O_t) - \boldsymbol{\eta}_t(O_t)\right) + \mathbf{A}(O_t)\mathbf{Q}_t$$

$$\quad + 2\boldsymbol{\psi}_t^\top \Pi_E\left[\mathbf{Q}_t^{\boldsymbol{\pi}_t} - \mathbf{Q}_{t+1}^{\boldsymbol{\pi}_{t+1}}\right] + 2\alpha^2\|\mathbf{r}_t(O_t) - \boldsymbol{\eta}_t(O_t) + \mathbf{A}(O_t)\mathbf{Q}_t\|_2^2 + 2\|\Pi_E\left[\mathbf{Q}_t^{\boldsymbol{\pi}_t} - \mathbf{Q}_{t+1}^{\boldsymbol{\pi}_{t+1}}\right]\|_2^2$$

$$\leq \|\boldsymbol{\psi}_t\|_2^2 + 2\alpha\boldsymbol{\psi}_t^\top\left(\mathbf{r}_t(O_t) - \boldsymbol{\eta}_t(O_t) + \mathbf{A}(O_t)\mathbf{Q}_t - \bar{\mathbf{A}}^{\boldsymbol{\pi}_t, \mathbf{P}_t}\boldsymbol{\psi}_t\right) + 2\alpha\boldsymbol{\psi}_t^\top \bar{\mathbf{A}}^{\boldsymbol{\pi}_t, \mathbf{P}_t}\boldsymbol{\psi}_t$$

$$\quad + 2\boldsymbol{\psi}_t^\top \Pi_E\left[\mathbf{Q}_t^{\boldsymbol{\pi}_t} - \mathbf{Q}_{t+1}^{\boldsymbol{\pi}_{t+1}}\right] + 2\alpha^2\|\mathbf{r}_t(O_t) - \boldsymbol{\eta}_t(O_t) + \mathbf{A}(O_t)\mathbf{Q}_t\|_2^2 + 2\|\Pi_E\left[\mathbf{Q}_t^{\boldsymbol{\pi}_t} - \mathbf{Q}_{t+1}^{\boldsymbol{\pi}_{t+1}}\right]\|_2^2$$

$$\leq \|\boldsymbol{\psi}_t\|_2^2 + 2\alpha\boldsymbol{\psi}_t^\top\left(\mathbf{r}_t(O_t) - \boldsymbol{\eta}_t(O_t) + \mathbf{A}(O_t)\mathbf{Q}_t^{\boldsymbol{\pi}_t}\right) + 2\alpha\boldsymbol{\psi}_t^\top\left(\mathbf{A}(O_t) - \bar{\mathbf{A}}^{\boldsymbol{\pi}_t, \mathbf{P}_t}\right)\boldsymbol{\psi}_t + 2\alpha\boldsymbol{\psi}_t^\top \bar{\mathbf{A}}^{\boldsymbol{\pi}_t, \mathbf{P}_t}\boldsymbol{\psi}_t$$

$$+ 2\boldsymbol{\psi}_t^\top \Pi_E \left[ \mathbf{Q}_t^{\boldsymbol{\pi}_t} - \mathbf{Q}_{t+1}^{\boldsymbol{\pi}_{t+1}} \right] + 2\alpha^2 \| \mathbf{r}_t(O_t) - \boldsymbol{\eta}_t(O_t) + \mathbf{A}(O_t)\mathbf{Q}_t \|_2^2 + 2\| \Pi_E \left[ \mathbf{Q}_t^{\boldsymbol{\pi}_t} - \mathbf{Q}_{t+1}^{\boldsymbol{\pi}_{t+1}} \right] \|_2^2$$

$$\leq \|\boldsymbol{\psi}_t\|_2^2 + 2\alpha\Gamma(\boldsymbol{\pi}_t, \mathbf{P}_t, \mathbf{r}_t, \boldsymbol{\psi}_t, O_t) + 2\alpha\boldsymbol{\psi}_t^\top (\mathbf{J}_t^{\boldsymbol{\pi}_t}(O_t) - \eta_t(O_t)) + 2\alpha\boldsymbol{\psi}_t^\top \bar{\mathbf{A}}^{\boldsymbol{\pi}_t, \mathbf{P}_t} \boldsymbol{\psi}_t$$

$$+ 2\boldsymbol{\psi}_t^\top \Pi_E \left[ \mathbf{Q}_t^{\boldsymbol{\pi}_t} - \mathbf{Q}_{t+1}^{\boldsymbol{\pi}_{t+1}} \right] + 2\alpha^2 \| \mathbf{r}_t(O_t) - \boldsymbol{\eta}_t(O_t) + \mathbf{A}(O_t)\mathbf{Q}_t \|_2^2 + 2\| \Pi_E \left[ \mathbf{Q}_t^{\boldsymbol{\pi}_t} - \mathbf{Q}_{t+1}^{\boldsymbol{\pi}_{t+1}} \right] \|_2^2$$

$$\overset{(a)}{\leq} \|\boldsymbol{\psi}_t\|_2^2 + 2\alpha\Gamma(\boldsymbol{\pi}_t, \mathbf{P}_t, \mathbf{r}_t, \boldsymbol{\psi}_t, O_t) + 2\alpha\|\boldsymbol{\psi}_t\|_2 \|\mathbf{J}_t^{\boldsymbol{\pi}_t}(O_t) - \eta_t(O_t)\|_2 + 2\alpha\boldsymbol{\psi}_t^\top \bar{\mathbf{A}}^{\boldsymbol{\pi}_t, \mathbf{P}_t} \boldsymbol{\psi}_t$$

$$+ 2\|\boldsymbol{\psi}_t\|_2 \| \Pi_E \left[ \mathbf{Q}_t^{\boldsymbol{\pi}_t} - \mathbf{Q}_{t+1}^{\boldsymbol{\pi}_{t+1}} \right] \|_2 + 2\alpha^2 \| \mathbf{r}_t(O_t) - \boldsymbol{\eta}_t(O_t) + \mathbf{A}(O_t)\mathbf{Q}_t \|_2^2 + 2\| \Pi_E \left[ \mathbf{Q}_t^{\boldsymbol{\pi}_t} - \mathbf{Q}_{t+1}^{\boldsymbol{\pi}_{t+1}} \right] \|_2^2$$

$$\overset{(b)}{\leq} \|\boldsymbol{\psi}_t\|_2^2 + 2\alpha\Gamma(\boldsymbol{\pi}_t, \mathbf{P}_t, \mathbf{r}_t, \boldsymbol{\psi}_t, O_t) + 2\alpha\|\boldsymbol{\psi}_t\|_2 |J_t^{\boldsymbol{\pi}_t} - \eta_t| - 2\alpha\lambda\|\boldsymbol{\psi}_t\|_2^2$$

$$+ 2\|\boldsymbol{\psi}_t\|_2 \| \Pi_E \left[ \mathbf{Q}_t^{\boldsymbol{\pi}_t} - \mathbf{Q}_{t+1}^{\boldsymbol{\pi}_{t+1}} \right] \|_2 + 2\alpha^2 \| \mathbf{r}_t(O_t) - \boldsymbol{\eta}_t(O_t) + \mathbf{A}(O_t)\mathbf{Q}_t \|_2^2 + 2\| \Pi_E \left[ \mathbf{Q}_t^{\boldsymbol{\pi}_t} - \mathbf{Q}_{t+1}^{\boldsymbol{\pi}_{t+1}} \right] \|_2^2$$

$$\leq (1 - 2\alpha\lambda)\|\boldsymbol{\psi}_t\|_2^2 + 2\alpha\Gamma(\boldsymbol{\pi}_t, \mathbf{P}_t, \mathbf{r}_t, \boldsymbol{\psi}_t, O_t) + 2\alpha\|\boldsymbol{\psi}_t\|_2 |J_t^{\boldsymbol{\pi}_t} - \eta_t|$$

$$+ 2\|\boldsymbol{\psi}_t\|_2 \| \Pi_E \left[ \mathbf{Q}_t^{\boldsymbol{\pi}_t} - \mathbf{Q}_{t+1}^{\boldsymbol{\pi}_{t+1}} \right] \|_2 + 2\alpha^2(2U_R + 2U_Q)^2 + 2\| \Pi_E \left[ \mathbf{Q}_t^{\boldsymbol{\pi}_t} - \mathbf{Q}_{t+1}^{\boldsymbol{\pi}_{t+1}} \right] \|_2^2,$$

where $(a)$ is due to Cauchy-Schwarz inequality, $(b)$ follows from $\boldsymbol{\psi}_t \in E$ and Lemma 5.1. Taking expectation, rearranging the terms, setting $\tau = \tau_T = \min\{i \geq 0 | m\rho^{i-1} \leq \min\{\beta, \alpha\}\}$ and summing over time, we have

$$\sum_{t=\tau_T}^{T-1} \lambda\mathbb{E} \left[ \|\boldsymbol{\psi}_t\|_2^2 \right]$$

$$\leq \underbrace{\sum_{t=\tau_T}^{T-1} \frac{\mathbb{E}[\|\boldsymbol{\psi}_t\|_2^2 - \|\boldsymbol{\psi}_{t+1}\|_2^2]}{2\alpha}}_{I_1} + \underbrace{\sum_{t=\tau_T}^{T-1} \mathbb{E} \left[ \Gamma(\boldsymbol{\pi}_t, \mathbf{P}_t, \mathbf{r}_t, \boldsymbol{\psi}_t, O_t) \right]}_{I_2} + \underbrace{\sum_{t=\tau_T}^{T-1} \mathbb{E} \left[ |\phi_t| \|\boldsymbol{\psi}_t\|_2 \right]}_{I_3} \qquad (16)$$

$$+ \underbrace{\sum_{t=\tau_T}^{T-1} \frac{\mathbb{E} \left[ \|\boldsymbol{\psi}_t\|_2 \| \Pi_E \left[ \mathbf{Q}_t^{\boldsymbol{\pi}_t} - \mathbf{Q}_{t+1}^{\boldsymbol{\pi}_{t+1}} \right] \|_2 \right]}{\alpha}}_{I_4} + \alpha(2U_R + 2U_Q)^2(T - \tau_T) + \underbrace{\sum_{t=\tau_T}^{T-1} \frac{\mathbb{E} \left[ \| \Pi_E \left[ \mathbf{Q}_t^{\boldsymbol{\pi}_t} - \mathbf{Q}_{t+1}^{\boldsymbol{\pi}_{t+1}} \right] \|_2^2 \right]}{\alpha}}_{I_5}.$$

We now bound each of the terms starting with the first term as

$$I_1 = \frac{\mathbb{E}[\|\boldsymbol{\psi}_{\tau_T}\|_2^2 - \|\boldsymbol{\psi}_T\|_2^2]}{2\alpha} \leq \frac{2U_Q^2}{\alpha}.$$

By Lemma G.9, we have

$$I_2 \leq \sum_{t=\tau_T}^{T-1} B_3\beta(\tau_T + 1)^2 + B_4\alpha\tau_T + B_5\Delta_{R,t-\tau_T+1,t} + B_6\tau_T\Delta_{P,t-\tau_T+1,t}$$

$$\leq B_3\beta(\tau_T + 1)^2(T - \tau_T) + B_4\alpha\tau_T(T - \tau_T) + B_5\tau_T\Delta_{R,T} + B_6\tau_T^2\Delta_{P,T}.$$

By the Cauchy-Schwarz inequality, we have

$$I_3 \leq \sum_{t=\tau_T}^{T-1} \sqrt{\mathbb{E}[\phi_t^2]}\sqrt{\mathbb{E}[\|\boldsymbol{\psi}_t\|_2^2]} \leq \left( \sum_{t=\tau_T}^{T-1} \mathbb{E}[\phi_t^2] \right)^{1/2} \left( \sum_{t=\tau_T}^{T-1} \mathbb{E}[\|\boldsymbol{\psi}_t\|_2^2] \right)^{1/2},$$

where $\sum_{t=\tau_T}^{T-1} \mathbb{E}[\phi_t^2]$ can be further bounded using Proposition 4.

We now upper bound the difference in state-action value function at consecutive timesteps as follows. For timesteps with small changes in the environment, we use Lemma G.8, and for timesteps with large changes in the environment, we use a naive upper bound. Define the set of timesteps $\mathcal{T}_Q := \{t : \|\mathbf{r}_{t+1} - \mathbf{r}_t\|_\infty \leq \delta_R, \|\mathbf{P}_{t+1} - \mathbf{P}_t\|_\infty \leq \delta_P\}$.

$$\sum_{t=\tau_T}^{T-1} \mathbb{E} \left[ \| \Pi_E \left[ \mathbf{Q}_t^{\boldsymbol{\pi}_t} - \mathbf{Q}_{t+1}^{\boldsymbol{\pi}_{t+1}} \right] \|_2^2 \right] \overset{(c)}{\leq} \sum_{t \in \mathcal{T}_Q} \mathbb{E} \left[ \| \Pi_E \left[ \mathbf{Q}_t^{\boldsymbol{\pi}_t} - \mathbf{Q}_{t+1}^{\boldsymbol{\pi}_{t+1}} \right] \|_2^2 \right] + \sum_{t \notin \mathcal{T}_Q} 4U_Q^2$$

$$\overset{(d)}{\leq} \sum_{t \in \mathcal{T}_Q} 3G_R^2\delta_R^2 + 3G_P^2\delta_P^2 + 3G_{\boldsymbol{\pi}}^2 B_2^2\beta^2 + \sum_{t \notin \mathcal{T}_Q} 4U_Q^2$$

$$\overset{(e)}{\leq} 3G_R^2\delta_R^2 T + 3G_P^2\delta_P^2 T + 3G_{\boldsymbol{\pi}}^2 B_2^2\beta^2 T + \frac{4U_Q^2\Delta_{R,T}}{\delta_R} + \frac{4U_Q^2\Delta_{P,T}}{\delta_P}$$

$$\overset{(f)}{\leq} W_1\Delta_{R,T}^{2/3}T^{1/3} + W_2\Delta_{P,T}^{2/3}T^{1/3} + 3G_{\boldsymbol{\pi}}^2 B_2^2\beta^2 T \tag{17}$$

where $(c)$ follows from the Lemma I.3, $(d)$ follows from Lemma G.8 and $(e)$ is obtained by choosing $\delta_R = \left(\frac{4U_Q^2\Delta_{R,T}}{3G_R^2 T}\right)^{1/3}$ and $\delta_P = \left(\frac{4U_Q^2\Delta_{P,T}}{3G_P^2 T}\right)^{1/3}$ and defining $W_1 = (3G_R^2)^{1/3}(4U_Q^2)^{2/3}$, $W_2 = (3G_P^2)^{1/3}(4U_Q^2)^{2/3}$.

To bound $I_4$, we use Cauchy-Schwarz Inequality and the bound in (17) above to get

$$I_4 \leq \left(\sum_{t=\tau_T}^{T-1} \frac{\mathbb{E}[\|\Pi_E\left[\mathbf{Q}_t^{\boldsymbol{\pi}_t} - \mathbf{Q}_{t+1}^{\boldsymbol{\pi}_{t+1}}\right]\|_2^2]}{\alpha^2}\right)^{1/2} \left(\sum_{t=\tau_T}^{T-1} \mathbb{E}[\|\boldsymbol{\psi}_t\|_2^2]\right)^{1/2}$$

$$\leq \left(\frac{W_1\Delta_{R,T}^{2/3}T^{1/3}}{\alpha^2} + \frac{W_2\Delta_{P,T}^{2/3}T^{1/3}}{\alpha^2} + \frac{3G_{\boldsymbol{\pi}}^2 B_2^2\beta^2 T}{\alpha^2}\right)^{1/2} \left(\sum_{t=\tau_T}^{T-1} \mathbb{E}[\|\boldsymbol{\psi}_t\|_2^2]\right)^{1/2}.$$

For the last term, again by using (17), we have

$$I_5 \leq \frac{W_1\Delta_{R,T}^{2/3}T^{1/3}}{\alpha} + \frac{W_2\Delta_{P,T}^{2/3}T^{1/3}}{\alpha} + \frac{3G_{\boldsymbol{\pi}}^2 B_2^2\beta^2 T}{\alpha}.$$

We substitute the bounds on $I_1, I_2, I_3, I_4, I_5$ (using Proposition 4) into (16) and use the squaring trick from Section C.3 in Wu et al. (2020). The above equation is of the form, $X \leq Y + Z\sqrt{X}$. Completing the squares and rearranging, we get $X \leq 2Y + Z^2$. Hence, we get the final result as

$$\sum_{t=\tau_T}^{T-1} \mathbb{E}\left[\|\boldsymbol{\psi}_t\|_2^2\right] \tag{18}$$

$$\leq \frac{4U_Q^2}{\alpha\lambda} + \frac{2B_3\beta(\tau_T+1)^2 T}{\lambda} + \frac{2\alpha(B_4 + 8U_R^2 + 8U_Q^2)\tau_T T}{\lambda} + \frac{2B_5\tau_T\Delta_{R,T}}{\lambda} + \frac{2B_6\tau_T^2\Delta_{P,T}}{\lambda}$$

$$+ \frac{8U_R^2}{\gamma\lambda^2} + \frac{4B_7\beta(\tau_T+1)^2 T}{\lambda^2} + \frac{2D_3\gamma\tau_T T}{\lambda^2} + \frac{4B_8(\tau_T+1)^2\Delta_{P,T}}{\lambda^2}$$

$$+ \frac{2D_4\beta^2 T}{\gamma^2\lambda^2} + \frac{6W_3\Delta_{R,T}^{2/3}T^{1/3}}{\gamma^2\lambda^2} + \frac{6W_4\Delta_{P,T}^{2/3}T^{1/3}}{\gamma^2\lambda^2}$$

$$+ \left(\frac{1}{\lambda^2} + \frac{1}{\lambda}\right)\left(\frac{W_1\Delta_{R,T}^{2/3}T^{1/3}}{\alpha^2} + \frac{W_2\Delta_{P,T}^{2/3}T^{1/3}}{\alpha^2} + \frac{3G_{\boldsymbol{\pi}}^2 B_2^2\beta^2 T}{\alpha^2}\right) \tag{19}$$

$$\leq \tilde{\mathcal{O}}\left(\frac{1}{\alpha}\right) + \tilde{\mathcal{O}}\left(\beta T\right) + \tilde{\mathcal{O}}\left(\alpha T\right) + \tilde{\mathcal{O}}\left(\Delta_{R,T}\right) + \tilde{\mathcal{O}}\left(\Delta_{P,T}\right) + \tilde{\mathcal{O}}\left(\gamma T\right) + \tilde{\mathcal{O}}\left(\frac{1}{\gamma}\right) + \tilde{\mathcal{O}}\left(\frac{\beta^2 T}{\gamma^2}\right)$$

$$+ \tilde{\mathcal{O}}\left(\frac{\Delta_{R,T}^{2/3}T^{1/3}}{\gamma^2}\right) + \tilde{\mathcal{O}}\left(\frac{\Delta_{P,T}^{2/3}T^{1/3}}{\gamma^2}\right) + \tilde{\mathcal{O}}\left(\frac{\Delta_{R,T}^{2/3}T^{1/3}}{\alpha^2}\right) + \tilde{\mathcal{O}}\left(\frac{\Delta_{P,T}^{2/3}T^{1/3}}{\alpha^2}\right) + \tilde{\mathcal{O}}\left(\frac{\beta^2 T}{\alpha^2}\right),$$

where $\tilde{\mathcal{O}}(\cdot)$ hides constants and logarithmic terms.

$\square$

## F AVERAGE REWARD

**Proposition 3.** *If Assumption 1 is satisfied, then the following holds true*

$$\sum_{t=\tau_T}^{T-1} \mathbb{E}\left[J_t^{\boldsymbol{\pi}_t} - r_t(s_t, a_t)\right] \leq D_1\beta(\tau_T+1)^2(T-\tau_T) + D_2(\tau_T+1)^2\Delta_{P,T}$$

*where $D_1 = L_{\boldsymbol{\pi}}B_2 + 4U_R\sqrt{|\mathcal{S}||\mathcal{A}|}B_2 + 4U_R$, $D_2 = 4U_R + L_P$ and $\Delta_{P,T} = \sum_{t=0}^{T-1} \|\mathbf{P}_{t+1} - \mathbf{P}_t\|_\infty$.*

*Proof.* Given time indices $t > \tau > 0$, recall the auxiliary Markov chain starting from $s_{t-\tau}$ constructed by conditioning on $s_{t-\tau}, \boldsymbol{\pi}_{t-\tau-1}, \mathbf{P}_{t-\tau}$ and rolling out by applying $\boldsymbol{\pi}_{t-\tau-1}, \mathbf{P}_{t-\tau}$ as

$$s_{t-\tau} \xrightarrow{\boldsymbol{\pi}_{t-\tau-1}} a_{t-\tau} \xrightarrow{\mathbf{P}_{t-\tau}} \tilde{s}_{t-\tau+1} \xrightarrow{\boldsymbol{\pi}_{t-\tau-1}} \tilde{a}_{t-\tau+1} \xrightarrow{\mathbf{P}_{t-\tau}} \dots \tilde{s}_t \xrightarrow{\boldsymbol{\pi}_{t-\tau-1}} \tilde{a}_t \xrightarrow{\mathbf{P}_{t-\tau}} \tilde{s}_{t+1} \xrightarrow{\boldsymbol{\pi}_{t-\tau-1}} \tilde{a}_{t+1}.$$

Also, recall that the original Markov chain is

$$s_{t-\tau} \xrightarrow{\boldsymbol{\pi}_{t-\tau-1}} a_{t-\tau} \xrightarrow{\mathbf{P}_{t-\tau}} s_{t-\tau+1} \xrightarrow{\boldsymbol{\pi}_{t-\tau}} a_{t-\tau+1} \xrightarrow{\mathbf{P}_{t-\tau+1}} \dots s_t \xrightarrow{\boldsymbol{\pi}_{t-1}} a_t \xrightarrow{\mathbf{P}_t} s_{t+1} \xrightarrow{\boldsymbol{\pi}_t} a_{t+1}.$$

Further, recall $J^{\boldsymbol{\pi}_{t-\tau-1}, \mathbf{P}_{t-\tau}, \mathbf{r}_t} := \sum_{s,a} d^{\boldsymbol{\pi}_{t-\tau-1}, \mathbf{P}_{t-\tau}}(s) \boldsymbol{\pi}_{t-\tau-1}(a|s) r_t(s,a)$.

We start by decomposing the term as

$$\mathbb{E}\left[J_t^{\boldsymbol{\pi}_t} - r_t(s_t, a_t)\right] = \underbrace{\mathbb{E}\left[J_t^{\boldsymbol{\pi}_t} - J^{\boldsymbol{\pi}_{t-\tau-1}, \mathbf{P}_{t-\tau}, \mathbf{r}_t}\right]}_{I_1} + \underbrace{\mathbb{E}\left[r_t(\tilde{s}_t, \tilde{a}_t) - r_t(s_t, a_t)\right]}_{I_2}$$

$$+ \underbrace{\mathbb{E}\left[J^{\boldsymbol{\pi}_{t-\tau-1}, \mathbf{P}_{t-\tau}, \mathbf{r}_t} - r_t(\tilde{s}_t, \tilde{a}_t)\right]}_{I_3}. \tag{20}$$

Note that $I_1$ is the difference in the average rewards between the two policies $\boldsymbol{\pi}_t, \boldsymbol{\pi}_{t-\tau-1}$ in two different environments $(\mathbf{P}_t, \mathbf{r}_t)$ and $(\mathbf{P}_{t-\tau}, \mathbf{r}_t)$ that share the same reward function. Hence, using Lemma G.5 and Lemma G.2 successively, we get

$$I_1 \leq \mathbb{E}\left[L_{\boldsymbol{\pi}} \|\boldsymbol{\pi}_t - \boldsymbol{\pi}_{t-\tau-1}\|_2 + L_P \|\mathbf{P}_t - \mathbf{P}_{t-\tau}\|_\infty\right]$$

$$\leq \mathbb{E}\left[L_{\boldsymbol{\pi}} \sum_{i=t-\tau}^t \|\boldsymbol{\pi}_i - \boldsymbol{\pi}_{i-1}\|_2 + L_P \sum_{i=t-\tau+1}^t \|\mathbf{P}_i - \mathbf{P}_{i-1}\|_\infty\right]$$

$$\leq L_{\boldsymbol{\pi}} B_2 \beta(\tau+1) + L_P \Delta_{P,t-\tau+1,t}, \tag{21}$$

where $\Delta_{P,t-\tau+1,t} = \sum_{i=t-\tau+1}^t \|\mathbf{P}_i - \mathbf{P}_{i-1}\|_\infty$.

For $I_2$, by Lemma I.2 and Lemma H.2 successively, we get

$$I_2 \leq 2U_R \cdot 2d_{TV}\left(P((s_t, a_t) \in \cdot|\mathcal{F}_{t-\tau}), P((\tilde{s}_t, \tilde{a}_t) \in \cdot|\mathcal{F}_{t-\tau})\right)$$

$$\leq 4U_R\sqrt{|\mathcal{S}||\mathcal{A}|}\mathbb{E}\left[\sum_{i=t-\tau}^t \|\boldsymbol{\pi}_i - \boldsymbol{\pi}_{t-\tau-1}\|_2 \Big| \mathcal{F}_{t-\tau}\right] + 4U_R \sum_{i=t-\tau}^t \|\mathbf{P}_i - \mathbf{P}_{t-\tau}\|_\infty$$

$$\leq 4U_R\sqrt{|\mathcal{S}||\mathcal{A}|}B_2\beta(\tau+1)^2 + 4U_R\tau\Delta_{P,t-\tau+1,t}. \tag{22}$$

Finally, we bound $I_3$ using Lemma H.7 as

$$I_3 \leq 4U_R m \rho^\tau. \tag{23}$$

Plugging the bounds on $I_1, I_2, I_3$ into Equation (20) and setting $\tau = \tau_T = \min\{i \geq 0 | m\rho^{i-1} \leq \min\{\beta, \alpha\}\}$,

$$\sum_{t=\tau_T}^{T-1} \mathbb{E}\left[J_t^{\boldsymbol{\pi}_t} - r_t(s_t, a_t)\right]$$

$$\leq \sum_{t=\tau_T}^{T-1} L_{\boldsymbol{\pi}}B_2\beta(\tau_T+1) + L_P\Delta_{P,t-\tau_T+1,t} + 4U_R\sqrt{|\mathcal{S}||\mathcal{A}|}B_2\beta(\tau_T+1)^2$$

$$+ 4U_R\tau_T\Delta_{P,t-\tau_T+1,t} + 4U_R m\rho^{\tau_T}$$

$$\leq (L_{\boldsymbol{\pi}} + 4U_R\sqrt{|\mathcal{S}||\mathcal{A}|})B_2\beta(\tau_T+1)^2(T-\tau_T) + (4U_R + L_P)(\tau_T+1)^2\Delta_{P,T} + 4U_R\beta(T-\tau_T).$$

$\square$

**Proposition 4.** *If Assumption 1 holds and $0 < \gamma < 1/2$, then we have the following*

$$\sum_{t=\tau}^{T-1} \mathbb{E}\left[(J_t^{\boldsymbol{\pi}^t} - \eta_t)^2\right] \leq \frac{4U_R^2}{\gamma} + 2B_7\beta(\tau_T+1)^2 T + D_3\gamma\tau_T T + 2B_8(\tau_T+1)^2\Delta_{P,T}$$

$$+ \frac{D_4\beta^2 T}{\gamma^2} + \frac{3W_3\Delta_{R,T}^{2/3}T^{1/3}}{\gamma^2} + \frac{3W_4\Delta_{P,T}^{2/3}T^{1/3}}{\gamma^2}$$

*where $D_3 = 4U_R F_6 + 8U_R^2$, $D_4 = 9L_{\boldsymbol{\pi}}^2 B_2^2$, $\Delta_{R,T} = \sum_{t=0}^{T-1} \|\mathbf{r}_{t+1} - \mathbf{r}_t\|_\infty$, $\Delta_{P,T} = \sum_{t=0}^{T-1} \|\mathbf{P}_{t+1} - \mathbf{P}_t\|_\infty$, $W_3 = (3)^{1/3}(4U_R^2)^{2/3}$ and $W_4 = (3L_P^2)^{1/3}(4U_R^2)^{2/3}$.*

*Proof.* Recall that $\phi_t := \eta_t - J_t^{\boldsymbol{\pi}^t}$. Using the average reward update equation (line 7 in Algorithm 1), we have

$$\phi_{t+1}^2 = \left(\eta_t + \gamma(r_t(s_t, a_t) - \eta_t) - J_{t+1}^{\boldsymbol{\pi}^{t+1}}\right)^2$$

$$= \left(\phi_t + J_t^{\boldsymbol{\pi}^t} - J_{t+1}^{\boldsymbol{\pi}^{t+1}} + \gamma(r_t(s_t, a_t) - \eta_t)\right)^2$$

$$\leq \phi_t^2 + 2\gamma\phi_t(r_t(s_t, a_t) - \eta_t) + 2\phi_t(J_t^{\boldsymbol{\pi}^t} - J_{t+1}^{\boldsymbol{\pi}^{t+1}}) + 2(J_t^{\boldsymbol{\pi}} - J_{t+1}^{\boldsymbol{\pi}^{t+1}})^2 + 2\gamma^2(r_t(s_t, a_t) - \eta_t)^2$$

$$= (1 - 2\gamma)\phi_t^2 + 2\gamma\phi_t(r_t(s_t, a_t) - J_t^{\boldsymbol{\pi}^t}) + 2\phi_t(J_t^{\boldsymbol{\pi}^t} - J_{t+1}^{\boldsymbol{\pi}^{t+1}})$$

$$+ 2(J_t^{\boldsymbol{\pi}} - J_{t+1}^{\boldsymbol{\pi}^{t+1}})^2 + 2\gamma^2(r_t(s_t, a_t) - \eta_t)^2$$

$$= (1 - 2\gamma)\phi_t^2 + 2\gamma\Lambda(\boldsymbol{\pi}_t, \mathbf{P}_t, \mathbf{r}_t, \eta_t, O_t) + 2\phi_t(J_t^{\boldsymbol{\pi}^t} - J_{t+1}^{\boldsymbol{\pi}^{t+1}})$$

$$+ 2(J_t^{\boldsymbol{\pi}} - J_{t+1}^{\boldsymbol{\pi}^{t+1}})^2 + 2\gamma^2(r_t(s_t, a_t) - \eta_t)^2.$$

Rearranging and setting $\tau = \tau_T = \min\{i \geq 0 | m\rho^{i-1} \leq \min\{\beta, \alpha\}\}$, we have

$$\sum_{t=\tau_T}^{T-1} \mathbb{E}[\phi_t^2] \leq \underbrace{\sum_{t=\tau_T}^{T-1} \frac{\mathbb{E}[\phi_t^2 - \phi_{t+1}^2]}{2\gamma}}_{I_1} + \underbrace{\sum_{t=\tau_T}^{T-1} \mathbb{E}[\Lambda(\boldsymbol{\pi}_t, \mathbf{P}_t, \mathbf{r}_t, \eta_t, O_t)]}_{I_2} + \underbrace{\sum_{t=\tau_T}^{T-1} \frac{\mathbb{E}[\phi_t(J_t^{\boldsymbol{\pi}^t} - J_{t+1}^{\boldsymbol{\pi}^{t+1}})]}{\gamma}}_{I_3}$$

$$+ \underbrace{\sum_{t=\tau_T}^{T-1} \frac{\mathbb{E}[(J_t^{\boldsymbol{\pi}^t} - J_{t+1}^{\boldsymbol{\pi}^{t+1}})^2]}{\gamma}}_{I_4} + \underbrace{\sum_{t=\tau_T}^{T-1} \gamma\mathbb{E}[(r_t(s_t, a_t) - \eta_t)^2]}_{I_5}.$$

We now analyze each of these terms starting with the first term as

$$I_1 = \frac{\mathbb{E}[\phi_{\tau_T}^2 - \phi_T^2]}{2\gamma} \leq \frac{2U_R^2}{\gamma}.$$

By Lemma G.10 and the average reward update equation, we have

$$I_2 \leq \sum_{t=\tau_T}^{T-1} B_7\beta(\tau_T+1)^2 + F_6|\eta_t - \eta_{t-\tau_T}| + F_7\tau\Delta_{P,t-\tau_T+1,t}$$

$$\leq B_7\beta(\tau_T+1)^2(T - \tau_T) + 2U_R F_6\gamma\tau_T(T - \tau_T) + B_8(\tau_T+1)^2\Delta_{P,T}.$$

We now upper bound the difference in average reward at consecutive timesteps as follows. For timesteps with small changes in the environment, we use Lemma G.5, and for timesteps with large changes in the environment, we use a naive upper bound. Define the set of timesteps $\mathcal{T}_J := \{t : \|\mathbf{r}_{t+1} - \mathbf{r}_t\|_\infty \leq \delta_R, \|\mathbf{P}_{t+1} - \mathbf{P}_t\|_\infty \leq \delta_P\}$.

$$\sum_{t=\tau_T}^{T-1} \mathbb{E}\left[(J_{t+1}^{\boldsymbol{\pi}^{t+1}} - J_t^{\boldsymbol{\pi}^t})^2\right] \leq \sum_{t\in\mathcal{T}_J} \mathbb{E}\left[(J_{t+1}^{\boldsymbol{\pi}^{t+1}} - J_t^{\boldsymbol{\pi}^t})^2\right] + \sum_{t\notin\mathcal{T}_J} 4U_R^2$$

$$\overset{(a)}{\leq} \sum_{t\in\mathcal{T}_J} 3\delta_R^2 + 3L_P^2\delta_P^2 + 3L_{\boldsymbol{\pi}}^2 B_2^2\beta^2 + \sum_{t\notin\mathcal{T}_J} 4U_R^2$$

$$\overset{(b)}{\leq} 3\delta_R^2 T + 3L_P^2 \delta_P^2 T + 3L_\pi^2 B_2^2 \beta^2 T + \frac{4U_R^2 \Delta_{R,T}}{\delta_R} + \frac{4U_R^2 \Delta_{P,T}}{\delta_P}$$

$$\leq W_3 \Delta_{R,T}^{2/3} T^{1/3} + W_4 \Delta_{P,T}^{2/3} T^{1/3} + 3L_\pi^2 B_2^2 \beta^2 T \tag{24}$$

where $(a)$ follows from Lemma G.5 and $(b)$ is obtained by choosing $\delta_R = \left( \frac{4U_R^2 \Delta_{R,T}}{3T} \right)^{1/3}$ and

$\delta_P = \left( \frac{4U_R^2 \Delta_{P,T}}{3L_P^2 T} \right)^{1/3}$ and defining $W_3 = (3)^{1/3}(4U_R^2)^{2/3}$, $W_2 = (3L_P^2)^{1/3}(4U_R^2)^{2/3}$.

Now we bound $I_3$ as

$$I_3 \overset{(c)}{\leq} \left( \sum_{t=\tau_T}^{T-1} \mathbb{E}[\phi_t^2] \right)^{1/2} \left( \sum_{t=\tau_T}^{T-1} \frac{\mathbb{E}[(J_{t+1}^{\pi_{t+1}} - J_t^{\pi_t})^2]}{\gamma^2} \right)^{1/2}$$

$$\overset{(d)}{\leq} \left( \sum_{t=\tau_T}^{T-1} \mathbb{E}[\phi_t^2] \right)^{1/2} \left( \frac{W_3 \Delta_{R,T}^{2/3} T^{1/3}}{\gamma^2} + \frac{W_4 \Delta_{P,T}^{2/3} T^{1/3}}{\gamma^2} + \frac{3L_\pi^2 B_2^2 \beta^2 T}{\gamma^2} \right)^{1/2}$$

where $(c)$ is by Cauchy-Schwarz inequality, $(d)$ follows from Equation (24).

Again, by Equation (24), we have

$$I_4 \leq \frac{W_3 \Delta_{R,T}^{2/3} T^{1/3}}{\gamma} + \frac{W_4 \Delta_{P,T}^{2/3} T^{1/3}}{\gamma} + \frac{3L_\pi^2 B_2^2 \beta^2 T}{\gamma}.$$

For the final term, we have

$$I_5 \leq 4U_R^2 \gamma (T - \tau_T).$$

Putting everything together, we have

$$\sum_{t=\tau_T}^{T-1} \mathbb{E}[\phi_t^2] \leq \frac{2U_R^2}{\gamma} + B_7 \beta (\tau_T + 1)^2 (T - \tau_T) + 2U_R F_6 \gamma \tau_T (T - \tau_T) + B_8 (\tau_T + 1)^2 \Delta_{P,T}$$

$$+ \left( \sum_{t=\tau_T}^{T-1} \mathbb{E}[\phi_t^2] \right)^{1/2} \left( \frac{W_3 \Delta_{R,T}^{2/3} T^{1/3}}{\gamma^2} + \frac{W_4 \Delta_{P,T}^{2/3} T^{1/3}}{\gamma^2} + \frac{3L_\pi^2 B_2^2 \beta^2 T}{\gamma^2} \right)^{1/2}$$

$$+ \frac{W_3 \Delta_{R,T}^{2/3} T^{1/3}}{\gamma} + \frac{W_4 \Delta_{P,T}^{2/3} T^{1/3}}{\gamma} + \frac{3L_\pi^2 B_2^2 \beta^2 T}{\gamma} + 4U_R^2 \gamma (T - \tau_T).$$

Now, we use the squaring trick from Section C.3, Wu et al. (2020). The above equation is of the form, $X \leq Y + Z\sqrt{X}$. Completing the squares and rearranging, we get $X \leq 2Y + Z^2$. Hence,

$$\sum_{t=\tau_T}^{T-1} \mathbb{E}[\phi_t^2] \leq \frac{4U_R^2}{\gamma} + 2B_7 \beta (\tau_T + 1)^2 (T - \tau_T) + 4U_R F_6 \gamma \tau_T (T - \tau_T) + 2B_8 (\tau_T + 1)^2 \Delta_{P,T}$$

$$+ \frac{9L_\pi^2 B_2^2 \beta^2 T}{\gamma^2} + \frac{3W_3 \Delta_{R,T}^{2/3} T^{1/3}}{\gamma^2} + \frac{3W_4 \Delta_{P,T}^{2/3} T^{1/3}}{\gamma^2} + 8U_R^2 \gamma (T - \tau_T).$$

$\square$

# G   TECHNICAL LEMMAS

**Lemma G.1.** *If Assumption 1 holds, for any $t, t' \geq 0$, we have*

$$\sum_s d^{\pi_{t'}^\star, \mathbf{P}_{t'}}(s) \left[ \frac{\log Z_t(s)}{\beta} - V_t^{\pi_t}(s) \right] \leq \frac{J_{t+1}^{\pi_{t+1}} - J_t^{\pi_t}}{C} + \epsilon \log |\mathcal{A}| + \|\mathbf{Q}_t^{\pi_t} - \mathbf{Q}_t\|_\infty + \frac{B_1 \beta}{C}$$

$$+ \frac{\|\mathbf{r}_{t+1} - \mathbf{r}_t\|_\infty}{C} + \frac{L_P \|\mathbf{P}_{t+1} - \mathbf{P}_t\|_\infty}{C}$$

where $Z_t(s) = \sum_{a' \in \mathcal{A}} (\pi_t(a'|s))^{1-\beta\epsilon} \exp(\beta Q_t(s, a'))$, $C$ is defined in Assumption 1 and other constants in Appendix B.

*Proof.* We have

$$J_{t+1}^{\pi_{t+1}} - J_t^{\pi_t}$$

$$= J_{t+1}^{\pi_{t+1}} - J_t^{\pi_{t+1}} + J_t^{\pi_{t+1}} - J_t^{\pi_t}$$

$$\overset{(a)}{=} J_{t+1}^{\pi_{t+1}} - J_t^{\pi_{t+1}} + \sum_{s,a} d^{\pi_{t+1}, \mathbf{P}_t}(s) \pi_{t+1}(a|s) \left[ Q_t^{\pi_t}(s, a) - V_t^{\pi_t}(s) + Q_t(s, a) - Q_t(s, a) \right]$$

$$\overset{(b)}{=} J_{t+1}^{\pi_{t+1}} - J_t^{\pi_{t+1}} + \sum_{s,a} d^{\pi_{t+1}, \mathbf{P}_t}(s) \pi_{t+1}(a|s) \Bigg[ Q_t^{\pi_t}(s, a) - V_t^{\pi_t}(s)$$

$$+ \frac{\log Z_t(s)}{\beta} + \frac{1}{\beta} \log \frac{\pi_{t+1}(a|s)}{(\pi_t(a|s))^{1-\beta\epsilon}} - Q_t(s, a) \Bigg]$$

$$\overset{(c)}{\geq} J_{t+1}^{\pi_{t+1}} - J_t^{\pi_{t+1}} + \sum_{s,a} d^{\pi_{t+1}, \mathbf{P}_t}(s) \pi_{t+1}(a|s) \left[ Q_t^{\pi_t}(s, a) - V_t^{\pi_t}(s) + \frac{\log Z_t(s)}{\beta} + \epsilon \log \pi_t(a|s) - Q_t(s, a) \right]$$

$$\geq J_{t+1}^{\pi_{t+1}} - J_t^{\pi_{t+1}} + \underbrace{\sum_{s,a} d^{\pi_{t+1}, \mathbf{P}_t}(s) \pi_t(a|s) \left[ \frac{\log Z_t(s)}{\beta} - Q_t(s, a) \right] + \sum_{s,a} d^{\pi_{t+1}, \mathbf{P}_t}(s) \pi_{t+1}(a|s) \epsilon \log \pi_t(a|s)}_{I_1}$$

$$+ \underbrace{\sum_{s,a} d^{\pi_{t+1}, \mathbf{P}_t}(s) (\pi_{t+1}(a|s) - \pi_t(a|s)) \left[ Q_t^{\pi_t}(s, a) - Q_t(s, a) \right]}_{I_2} \qquad (25)$$

where $(a)$ follows from Lemma G.3, $(b)$ follows from the actor update (line 10 in Algorithm 1), and $(c)$ is due to the non-negativity of KL-Divergence.

Next, we bound the last two terms in (25). Under Assumption 1, we have

$$I_1 = \sum_{s,a} d^{\pi_{t'}^\star, \mathbf{P}_{t'}}(s) \left( \frac{d^{\pi_{t+1}, \mathbf{P}_t}(s)}{d^{\pi_{t'}^\star, \mathbf{P}_{t'}}(s)} \right) \pi_t(a|s) \left[ \frac{\log Z_t(s)}{\beta} - Q_t(s, a) \right]$$

$$+ \sum_{s,a} d^{\pi_{t'}^\star, \mathbf{P}_{t'}}(s) \left( \frac{d^{\pi_{t+1}, \mathbf{P}_t}(s)}{d^{\pi_{t'}^\star, \mathbf{P}_{t'}}(s)} \right) \pi_{t+1}(a|s) \epsilon \log \pi_t(a|s)$$

$$\geq C \sum_{s,a} d^{\pi_{t'}^\star, \mathbf{P}_{t'}}(s) \pi_t(a|s) \left[ \frac{\log Z_t(s)}{\beta} - Q_t(s, a) \right] + \sum_{s,a} d^{\pi_{t'}^\star, \mathbf{P}_{t'}}(s) \left( \frac{d^{\pi_{t+1}, \mathbf{P}_t}(s)}{d^{\pi_{t'}^\star, \mathbf{P}_{t'}}(s)} \right) \pi_{t+1}(a|s) \epsilon \log \pi_t(a|s)$$

$$\overset{(d)}{\geq} C \sum_s d^{\pi_{t'}^\star, \mathbf{P}_{t'}}(s) \left[ \frac{\log Z_t(s)}{\beta} - V_t^{\pi_t}(s) \right] C\epsilon \log |\mathcal{A}| + C \sum_{s,a} d^{\pi_{t'}^\star, \mathbf{P}_{t'}}(s) \pi_t(a|s) \left[ Q_t^{\pi_t}(s, a) - Q_t(s, a) \right]$$

$$(26)$$

where $(d)$ follows from Lemma H.1.

Further, by Lemma G.2, we have

$$I_2 \geq -2U_Q \sum_{s,a} d^{\pi_{t+1}, \mathbf{P}_t}(s) (\pi_{t+1}(a|s) - \pi_t(a|s)) \geq -2U_Q \cdot 2\beta U_Q \sqrt{|\mathcal{A}|} \geq -B_1 \beta. \qquad (27)$$

Plugging the bounds from (26) and (27) into (25) and rearranging, we have

$$\sum_s d^{\pi_{t'}^\star, \mathbf{P}_{t'}}(s) \left[ \frac{\log Z_t(s)}{\beta} - V_t^{\pi_t}(s) \right]$$

$$\leq \frac{J_{t+1}^{\boldsymbol{\pi}_{t+1}} - J_t^{\boldsymbol{\pi}_t}}{C} + \epsilon \log|\mathcal{A}| + \sum_{s,a} d^{\boldsymbol{\pi}_{t'}^\star, \mathbf{P}_{t'}}(s)\pi_t(a|s)\left[Q_t(s,a) - Q_t^{\boldsymbol{\pi}_t}(s,a)\right] + \frac{B_1\beta}{C} + \frac{J_t^{\boldsymbol{\pi}_{t+1}} - J_{t+1}^{\boldsymbol{\pi}_{t+1}}}{C}$$

$$\leq \frac{J_{t+1}^{\boldsymbol{\pi}_{t+1}} - J_t^{\boldsymbol{\pi}_t}}{C} + \epsilon \log|\mathcal{A}| + \|\mathbf{Q}_t^{\boldsymbol{\pi}_t} - \mathbf{Q}_t\|_\infty + \frac{B_1\beta}{C} + \frac{\|\mathbf{r}_{t+1} - \mathbf{r}_t\|_\infty}{C} + \frac{L_P\|\mathbf{P}_{t+1} - \mathbf{P}_t\|_\infty}{C}$$

where the last inequality follows from Lemma G.5. $\qquad\square$

**Lemma G.2.** *For $t \geq 0$, policy $\boldsymbol{\pi}_t$ satisfies*

$$\|\boldsymbol{\pi}_{t+1} - \boldsymbol{\pi}_t\|_2 \leq B_2\beta$$

*where $B_2 = 2U_Q$.*

*Proof.* From the update(5), we have the following recursive equation

$$\boldsymbol{\theta}_{t+1} = (1 - \beta\epsilon)\boldsymbol{\theta}_t + \beta\mathbf{Q}_t$$

$$= (1 - \beta\epsilon)^{t+1}\boldsymbol{\theta}_0 + \beta\sum_{i=0}^{t}(1 - \beta\epsilon)^{t-i-1}\mathbf{Q}_i.$$

By starting from $\boldsymbol{\theta}_0 = 0$ and 1-Lipschitzness of the softmax parameterization (Beck, 2017) and Lemma I.3, we have

$$\|\boldsymbol{\pi}_{t+1} - \boldsymbol{\pi}_t\|_2 \leq \|\beta\mathbf{Q}_t - \beta\epsilon\boldsymbol{\theta}_t\|_2$$

$$\leq \beta\|\mathbf{Q}_t\|_2 + \beta\epsilon\|\boldsymbol{\theta}_t\|_2$$

$$\leq \beta U_Q + \beta\epsilon\frac{U_Q}{\epsilon} \leq 2\beta U_Q.$$

$\qquad\square$

**Lemma G.3** (Average-reward Performance Difference Lemma (Murthy & Srikant, 2023)). *The average rewards for any two policies $\boldsymbol{\pi}, \boldsymbol{\pi}'$ at time $t$ satisfy*

$$J_t^{\boldsymbol{\pi}} - J_t^{\boldsymbol{\pi}'} = \sum_{s\in\mathcal{S}} d^{\boldsymbol{\pi}, \mathbf{P}_t}(s)\sum_{a\in\mathcal{A}}\boldsymbol{\pi}(a|s)\left[Q_t^{\boldsymbol{\pi}'}(s,a) - V_t^{\boldsymbol{\pi}'}(s)\right].$$

**Lemma G.4.** *For any $t, t' \geq 0$, it holds that*

$$J_t^{\boldsymbol{\pi}_t^\star} - J_{t'}^{\boldsymbol{\pi}_{t'}^\star} \leq \|\mathbf{r}_t - \mathbf{r}_{t'}\|_\infty + U_Q\|\mathbf{P}_t - \mathbf{P}_{t'}\|_\infty.$$

*where $\boldsymbol{\pi}_t^\star$ represents the optimal policy for MDP $\mathcal{M}_t(\mathcal{S}, \mathcal{A}, \mathbf{P}_t, \mathbf{r}_t)$.*

*Proof.* Consider the linear programming formulation of an MDP $\mathcal{M}(\mathcal{S}, \mathcal{A}, \mathbf{P}, \mathbf{r})$ (Puterman, 2014)

$$\min_{J, V(s)} J$$

$$\text{such that } J + V(s) \geq r(s,a) + \sum_{s'} P(s'|s,a)V(s') \; \forall s \in \mathcal{S}, a \in \mathcal{A}. \qquad (28)$$

If the optimal solution for $\mathcal{M}_{t'}(\mathcal{S}, \mathcal{A}, \mathbf{P}_{t'}, \mathbf{r}_{t'})$ is $J_{t'}^\star, \mathbf{V}_{t'}^\star$, we have

$$J_{t'}^\star\mathbf{1} \geq \mathbf{r}_{t'} + (\mathbf{P}_{t'} - \mathbf{I})\mathbf{V}_{t'}^\star.$$

Now for $\mathcal{M}_t(\mathcal{S}, \mathcal{A}, \mathbf{P}_t, \mathbf{r}_t)$, we know

$$J_t^\star \leq \|\mathbf{r}_t + (\mathbf{P}_t - \mathbf{I})\mathbf{V}_{t'}^\star\|_\infty$$

$$\leq \|\mathbf{r}_{t'} + (\mathbf{P}_{t'} - \mathbf{I})\mathbf{V}_{t'}^\star + (\mathbf{r}_t - \mathbf{r}_{t'}) + (\mathbf{P}_t - \mathbf{P}_{t'})\mathbf{V}_{t'}^\star\|_\infty$$

$$\leq \|J_{t'}^\star\mathbf{1}\|_\infty + \|\mathbf{r}_t - \mathbf{r}_{t'}\|_\infty + \|(\mathbf{P}_t - \mathbf{P}_{t'})\mathbf{V}_{t'}^\star\|_\infty.$$

Hence, we have

$$J_t^\star - J_{t'}^\star \leq \|\mathbf{r}_t - \mathbf{r}_{t'}\|_\infty + \|(\mathbf{P}_t - \mathbf{P}_{t'})\mathbf{V}_{t'}^\star\|_\infty$$

$$J_t^{\boldsymbol{\pi}_t^\star} - J_{t'}^{\boldsymbol{\pi}_{t'}^\star} \leq \|\mathbf{r}_t - \mathbf{r}_{t'}\|_\infty + U_Q\|\mathbf{P}_t - \mathbf{P}_{t'}\|_\infty.$$

$\qquad\square$

The next result bounds the difference in the average rewards between two policies $\boldsymbol{\pi}, \boldsymbol{\pi}'$ under two different environments $(\mathbf{r}_t, \mathbf{P}_t)$ and $(\mathbf{r}_{t'}, \mathbf{P}_{t'})$.

**Lemma G.5.** *There exist constants $L_{\boldsymbol{\pi}} = 4U_R(M+1)\sqrt{|\mathcal{S}||\mathcal{A}|}$ and $L_P = 4U_RM$ such that for all policies $\boldsymbol{\pi}, \boldsymbol{\pi}'$ and timesteps $t, t'$, it holds that*

$$J_t^{\boldsymbol{\pi}} - J_{t'}^{\boldsymbol{\pi}'} \leq L_\pi \|\boldsymbol{\pi} - \boldsymbol{\pi}'\|_2 + \|\mathbf{r}_t - \mathbf{r}_{t'}\|_\infty + L_P\|\mathbf{P}_t - \mathbf{P}_{t'}\|_\infty.$$

*Proof.*

$$J_t^{\boldsymbol{\pi}} - J_{t'}^{\boldsymbol{\pi}'} = \underbrace{J_t^{\boldsymbol{\pi}} - J_t^{\boldsymbol{\pi}'}}_{T_1} + \underbrace{J_t^{\boldsymbol{\pi}'} - J_{t'}^{\boldsymbol{\pi}'}}_{T_2}, \tag{29}$$

where $T_1$ is the difference in the average rewards between two policies $\boldsymbol{\pi}, \boldsymbol{\pi}'$ under the same environments $(\mathbf{r}_t, \mathbf{P}_t)$, while $T_2$ is the difference in the average rewards with the same policy $\boldsymbol{\pi}'$, but under two different environments $(\mathbf{r}_t, \mathbf{P}_t)$ and $(\mathbf{r}_{t'}, \mathbf{P}_{t'})$.

$$\begin{aligned} T_1 = J_t^{\boldsymbol{\pi}} - J_t^{\boldsymbol{\pi}'} &= \mathbb{E}_{s\sim d^{\boldsymbol{\pi},\mathbf{P}_t}, a\sim\boldsymbol{\pi}, s'\sim d^{\boldsymbol{\pi}',\mathbf{P}_t}, a'\sim\boldsymbol{\pi}'}[r_t(s,a) - r_t(s',a')] \\ &= 4U_R d_{TV}\left(d^{\boldsymbol{\pi},\mathbf{P}_t} \otimes \boldsymbol{\pi}, d^{\boldsymbol{\pi}',\mathbf{P}_t} \otimes \boldsymbol{\pi}'\right) \\ &\overset{(a)}{\leq} L_{\boldsymbol{\pi}}\|\boldsymbol{\pi} - \boldsymbol{\pi}'\|_2, \end{aligned} \tag{30}$$

where $(a)$ follows from Lemma I.1, where $\otimes$ denotes the Kronecker product. Next, we bound $T_2$.

$$\begin{aligned} T_2 = J_t^{\boldsymbol{\pi}'} - J_{t'}^{\boldsymbol{\pi}'} &= \sum_{s,a} d^{\boldsymbol{\pi}',\mathbf{P}_t}(s)\boldsymbol{\pi}'(a|s)r_t(s,a) - d^{\boldsymbol{\pi}',\mathbf{P}_{t'}}(s)\boldsymbol{\pi}'(a|s)r_{t'}(s,a) \\ &\leq \sum_{s,a} \left| d^{\boldsymbol{\pi}',\mathbf{P}_t}(s)\boldsymbol{\pi}'(a|s)r_t(s,a) - d^{\boldsymbol{\pi}',\mathbf{P}_t}(s)\boldsymbol{\pi}'(a|s)r_{t'}(s,a) \right| \\ &\quad + \sum_{s,a} \left| d^{\boldsymbol{\pi}',\mathbf{P}_t}(s)\boldsymbol{\pi}'(a|s)r_{t'}(s,a) - d^{\boldsymbol{\pi}',\mathbf{P}_{t'}}(s)\boldsymbol{\pi}'(a|s)r_{t'}(s,a) \right| \\ &\leq \|\mathbf{r}_t - \mathbf{r}_{t'}\|_\infty + 4U_R d_{TV}(d^{\boldsymbol{\pi}',\mathbf{P}_t} \otimes \boldsymbol{\pi}', d^{\boldsymbol{\pi}',\mathbf{P}_{t'}} \otimes \boldsymbol{\pi}') \\ &\overset{(b)}{\leq} \|\mathbf{r}_t - \mathbf{r}_{t'}\|_\infty + L_P\|\mathbf{P}_t - \mathbf{P}_{t'}\|_\infty \end{aligned} \tag{31}$$

where $(b)$ also follows from Lemma I.1. Substituting the bounds from (30) and (31) into (29), we get the result. $\square$

In the next result, we bound the value function difference for two different policies $\boldsymbol{\pi}, \boldsymbol{\pi}'$, given the underlying environment $\{\mathbf{r}_t, \mathbf{P}_t\}$. Consequently, the value-function difference depends only on the difference between the two policies.

**Lemma G.6.** *For any policies $\boldsymbol{\pi}, \boldsymbol{\pi}'$, we have*

$$\|\mathbf{Q}_t^{\boldsymbol{\pi}} - \mathbf{Q}_t^{\boldsymbol{\pi}'}\|_2 \leq G_\pi\|\boldsymbol{\pi} - \boldsymbol{\pi}'\|_2$$

*where $G_\pi = 2U_Q\sqrt{|\mathcal{S}||\mathcal{A}|}$.*

*Proof.*

$$Q_t^{\boldsymbol{\pi}}(s,a) \overset{(a)}{=} r_t(s,a) - J_t^{\boldsymbol{\pi}} + \mathbb{E}_{s'\sim P_t(\cdot|s,a)}[V_t^{\boldsymbol{\pi}}(s')]$$

$$\Rightarrow \frac{\partial Q_t^{\boldsymbol{\pi}}(s,a)}{\partial \boldsymbol{\pi}} = \frac{-\partial J_t^{\boldsymbol{\pi}}}{\partial \boldsymbol{\pi}} + \sum_{s'\in\mathcal{S}} P_t(s'|s,a)\frac{\partial V_t^{\boldsymbol{\pi}}(s')}{\partial \boldsymbol{\pi}}$$

$$\left\|\frac{\partial Q_t^{\boldsymbol{\pi}}(s,a)}{\partial \boldsymbol{\pi}}\right\|_2 \leq 2\left\|\frac{\partial J_t^{\boldsymbol{\pi}}}{\partial \boldsymbol{\pi}}\right\|_2 \tag{32}$$

$$\left\|\frac{\partial Q_t^{\boldsymbol{\pi}}(s,a)}{\partial \boldsymbol{\pi}}\right\|_2 \overset{(b)}{\leq} 2\left\|d^{\boldsymbol{\pi},\mathbf{P}_t}(s)Q_t^{\boldsymbol{\pi}}(s,a)\right\|_2 \leq 2U_Q \tag{33}$$

It follows from mean-value theorem that

$$|Q_t^{\boldsymbol{\pi}}(s,a) - Q_t^{\boldsymbol{\pi}'}(s,a)| \leq 2U_Q \|\boldsymbol{\pi} - \boldsymbol{\pi}'\|_2, \text{ for all } s, a$$

$$\Rightarrow \|\mathbf{Q}_t^{\boldsymbol{\pi}} - \mathbf{Q}_t^{\boldsymbol{\pi}'}\|_2 \leq G_{\boldsymbol{\pi}} \|\boldsymbol{\pi} - \boldsymbol{\pi}'\|_2,$$

where $(a)$ is by using the Bellman equation, and $(b)$ follows from Policy Gradient Theorem (Sutton & Barto, 2018) and Lemma I.3. $\qquad\square$

Complementary to Lemma G.6, in the next result, we bound the difference between the value functions when following the same policy $\boldsymbol{\pi}$ in two different environments $\{\mathbf{r}_t, \mathbf{P}_t\}$ and $\{\mathbf{r}_{t'}, \mathbf{P}_{t'}\}$.

**Lemma G.7.** *For any timesteps $t, t' \geq 0$, we have*

$$\|\Pi_E\left[\mathbf{Q}_t^{\boldsymbol{\pi}} - \mathbf{Q}_{t'}^{\boldsymbol{\pi}}\right]\|_2 \leq G_R \|\mathbf{r}_t - \mathbf{r}_{t'}\|_\infty + G_P \|\mathbf{P}_t - \mathbf{P}_{t'}\|_\infty$$

*where $G_R = 2\lambda^{-1}\sqrt{|\mathcal{S}||\mathcal{A}|}$ and $G_P = (\lambda^{-1}L_P + 4U_R\lambda^{-1}M + 4U_R\lambda^{-2}(M+1))\sqrt{|\mathcal{S}||\mathcal{A}|}$.*

*Proof.* Recall the diagonal matrix $D^{\boldsymbol{\pi}, \mathbf{P}_t} = diag\left(d^{\boldsymbol{\pi}, \mathbf{P}_t}(s)\pi(a|s)\right)$, where $d^{\boldsymbol{\pi}, \mathbf{P}_t}(\cdot)$ denotes the stationary distribution induced over the states, while $\mathbf{1}$ denotes the all ones vector. E denotes the subspace orthogonal to the all ones vector. Pseudo-inverse of a matrix is represented by $\mathbf{X}^\dagger$. Now, we have

$$\|\Pi_E\left[\mathbf{Q}_t^{\boldsymbol{\pi}} - \mathbf{Q}_{t'}^{\boldsymbol{\pi}}\right]\|_2 \overset{(a)}{=} \|(\bar{\mathbf{A}}^{\boldsymbol{\pi}, \mathbf{P}_t})^\dagger D^{\boldsymbol{\pi}, \mathbf{P}_t}(J_t^{\boldsymbol{\pi}}\mathbf{1} - \mathbf{r}_t) - (\bar{\mathbf{A}}^{\boldsymbol{\pi}, \mathbf{P}_{t'}})^\dagger D^{\boldsymbol{\pi}, \mathbf{P}_{t'}}(J_{t'}^{\boldsymbol{\pi}}\mathbf{1} - \mathbf{r}_{t'})\|_2$$

$$\leq \|(\bar{\mathbf{A}}^{\boldsymbol{\pi}, \mathbf{P}_t})^\dagger D^{\boldsymbol{\pi}, \mathbf{P}_t}(J_t^{\boldsymbol{\pi}}\mathbf{1} - \mathbf{r}_t) - (\bar{\mathbf{A}}^{\boldsymbol{\pi}, \mathbf{P}_t})^\dagger D^{\boldsymbol{\pi}, \mathbf{P}_{t'}}(J_{t'}^{\boldsymbol{\pi}}\mathbf{1} - \mathbf{r}_{t'})\|_2$$

$$\qquad + \|(\bar{\mathbf{A}}^{\boldsymbol{\pi}, \mathbf{P}_t})^\dagger D^{\boldsymbol{\pi}, \mathbf{P}_{t'}}(J_{t'}^{\boldsymbol{\pi}}\mathbf{1} - \mathbf{r}_{t'}) - (\bar{\mathbf{A}}^{\boldsymbol{\pi}, \mathbf{P}_{t'}})^\dagger D^{\boldsymbol{\pi}, \mathbf{P}_{t'}}(J_{t'}^{\boldsymbol{\pi}}\mathbf{1} - \mathbf{r}_{t'})\|_2$$

$$\leq \|(\bar{\mathbf{A}}^{\boldsymbol{\pi}, \mathbf{P}_t})^\dagger\|_2 \left(\|D^{\boldsymbol{\pi}, \mathbf{P}_t}J_t^{\boldsymbol{\pi}}\mathbf{1} - D^{\boldsymbol{\pi}, \mathbf{P}_{t'}}J_{t'}^{\boldsymbol{\pi}}\mathbf{1}\|_2 + \|D^{\boldsymbol{\pi}, \mathbf{P}_t}\mathbf{r}_t - D^{\boldsymbol{\pi}, \mathbf{P}_{t'}}\mathbf{r}_{t'}\|_2\right)$$

$$\qquad + \|(\bar{\mathbf{A}}^{\boldsymbol{\pi}, \mathbf{P}_t})^\dagger D^{\boldsymbol{\pi}, \mathbf{P}_{t'}}(J_{t'}^{\boldsymbol{\pi}}\mathbf{1} - \mathbf{r}_{t'}) - (\bar{\mathbf{A}}^{\boldsymbol{\pi}, \mathbf{P}_{t'}})^\dagger D^{\boldsymbol{\pi}, \mathbf{P}_{t'}}(J_{t'}^{\boldsymbol{\pi}}\mathbf{1} - \mathbf{r}_{t'})\|_2$$

$$\overset{(b)}{\leq} \lambda^{-1}\left(\|D^{\boldsymbol{\pi}, \mathbf{P}_t}(J_t^{\boldsymbol{\pi}} - J_{t'}^{\boldsymbol{\pi}})\mathbf{1}\|_2 + \|(D^{\boldsymbol{\pi}, \mathbf{P}_t} - D^{\boldsymbol{\pi}, \mathbf{P}_{t'}})J_{t'}^{\boldsymbol{\pi}}\mathbf{1}\|_2 + \|D^{\boldsymbol{\pi}, \mathbf{P}_t}\mathbf{r}_t - D^{\boldsymbol{\pi}, \mathbf{P}_{t'}}\mathbf{r}_{t'}\|_2\right)$$

$$\qquad + \|(\bar{\mathbf{A}}^{\boldsymbol{\pi}, \mathbf{P}_t})^\dagger D^{\boldsymbol{\pi}, \mathbf{P}_{t'}}(J_{t'}^{\boldsymbol{\pi}}\mathbf{1} - \mathbf{r}_{t'}) - (\bar{\mathbf{A}}^{\boldsymbol{\pi}, \mathbf{P}_{t'}})^\dagger D^{\boldsymbol{\pi}, \mathbf{P}_{t'}}(J_{t'}^{\boldsymbol{\pi}}\mathbf{1} - \mathbf{r}_{t'})\|_2$$

$$\leq \lambda^{-1}\left(\sqrt{|\mathcal{S}||\mathcal{A}|}\|D^{\boldsymbol{\pi}, \mathbf{P}_t}\|_2|J_t^{\boldsymbol{\pi}} - J_{t'}^{\boldsymbol{\pi}}| + \|D^{\boldsymbol{\pi}, \mathbf{P}_t} - D^{\boldsymbol{\pi}, \mathbf{P}_{t'}}\|_2 \cdot U_R\sqrt{|\mathcal{S}||\mathcal{A}|} + \|D^{\boldsymbol{\pi}, \mathbf{P}_t}\mathbf{r}_t - D^{\boldsymbol{\pi}, \mathbf{P}_{t'}}\mathbf{r}_{t'}\|_2\right)$$

$$\qquad + \|(\bar{\mathbf{A}}^{\boldsymbol{\pi}, \mathbf{P}_t})^\dagger D^{\boldsymbol{\pi}, \mathbf{P}_{t'}}(J_{t'}^{\boldsymbol{\pi}}\mathbf{1} - \mathbf{r}_{t'}) - (\bar{\mathbf{A}}^{\boldsymbol{\pi}, \mathbf{P}_{t'}})^\dagger D^{\boldsymbol{\pi}, \mathbf{P}_{t'}}(J_{t'}^{\boldsymbol{\pi}}\mathbf{1} - \mathbf{r}_{t'})\|_2$$

$$\overset{(c)}{\leq} \lambda^{-1}\sqrt{|\mathcal{S}||\mathcal{A}|}\left(\|\mathbf{r}_t - \mathbf{r}_{t'}\|_\infty + L_P\|\mathbf{P}_t - \mathbf{P}_{t'}\|_\infty + 2U_R d_{TV}(d^{\boldsymbol{\pi}, \mathbf{P}_t} \otimes \boldsymbol{\pi}, d^{\boldsymbol{\pi}, \mathbf{P}_{t'}} \otimes \boldsymbol{\pi})\right)$$

$$\qquad + \lambda^{-1}\|D^{\boldsymbol{\pi}, \mathbf{P}_t}\mathbf{r}_t - D^{\boldsymbol{\pi}, \mathbf{P}_{t'}}\mathbf{r}_{t'}\|_2 \tag{34}$$

$$\qquad + \|(\bar{\mathbf{A}}^{\boldsymbol{\pi}, \mathbf{P}_t})^\dagger D^{\boldsymbol{\pi}, \mathbf{P}_{t'}}(J_{t'}^{\boldsymbol{\pi}}\mathbf{1} - \mathbf{r}_{t'}) - (\bar{\mathbf{A}}^{\boldsymbol{\pi}, \mathbf{P}_{t'}})^\dagger D^{\boldsymbol{\pi}, \mathbf{P}_{t'}}(J_{t'}^{\boldsymbol{\pi}}\mathbf{1} - \mathbf{r}_{t'})\|_2$$

$$\overset{(d)}{\leq} \lambda^{-1}\sqrt{|\mathcal{S}||\mathcal{A}|}\left(\|\mathbf{r}_t - \mathbf{r}_{t'}\|_\infty + L_P\|\mathbf{P}_t - \mathbf{P}_{t'}\|_\infty + 2U_R M\|\mathbf{P}_t - \mathbf{P}_{t'}\|_\infty\right)$$

$$\qquad + \lambda^{-1}\|D^{\boldsymbol{\pi}, \mathbf{P}_t}\mathbf{r}_t - D^{\boldsymbol{\pi}, \mathbf{P}_{t'}}\mathbf{r}_{t'}\|_2 \tag{35}$$

$$\qquad + \|(\bar{\mathbf{A}}^{\boldsymbol{\pi}, \mathbf{P}_t})^\dagger D^{\boldsymbol{\pi}, \mathbf{P}_{t'}}(J_{t'}^{\boldsymbol{\pi}}\mathbf{1} - \mathbf{r}_{t'}) - (\bar{\mathbf{A}}^{\boldsymbol{\pi}, \mathbf{P}_{t'}})^\dagger D^{\boldsymbol{\pi}, \mathbf{P}_{t'}}(J_{t'}^{\boldsymbol{\pi}}\mathbf{1} - \mathbf{r}_{t'})\|_2$$

$$\overset{(e)}{\leq} \lambda^{-1}\sqrt{|\mathcal{S}||\mathcal{A}|}\left(2\|\mathbf{r}_t - \mathbf{r}_{t'}\|_\infty + L_P\|\mathbf{P}_t - \mathbf{P}_{t'}\|_\infty + 4U_R M\|\mathbf{P}_t - \mathbf{P}_{t'}\|_\infty\right)$$

$$\qquad + \|(\bar{\mathbf{A}}^{\boldsymbol{\pi}, \mathbf{P}_t})^\dagger D^{\boldsymbol{\pi}, \mathbf{P}_{t'}}(J_{t'}^{\boldsymbol{\pi}}\mathbf{1} - \mathbf{r}_{t'}) - (\bar{\mathbf{A}}^{\boldsymbol{\pi}, \mathbf{P}_{t'}})^\dagger D^{\boldsymbol{\pi}, \mathbf{P}_{t'}}(J_{t'}^{\boldsymbol{\pi}}\mathbf{1} - \mathbf{r}_{t'})\|_2$$

$$\leq \lambda^{-1}\sqrt{|\mathcal{S}||\mathcal{A}|}\left(2\|\mathbf{r}_t - \mathbf{r}_{t'}\|_\infty + L_P\|\mathbf{P}_t - \mathbf{P}_{t'}\|_\infty + 4U_R M\|\mathbf{P}_t - \mathbf{P}_{t'}\|_\infty\right)$$

$$\qquad + \|(\bar{\mathbf{A}}^{\boldsymbol{\pi}, \mathbf{P}_t})^\dagger - (\bar{\mathbf{A}}^{\boldsymbol{\pi}, \mathbf{P}_{t'}})^\dagger\|_2 \cdot 2U_R$$

$$\overset{(f)}{\leq} \lambda^{-1}\sqrt{|\mathcal{S}||\mathcal{A}|}\left(2\|\mathbf{r}_t - \mathbf{r}_{t'}\|_\infty + L_P\|\mathbf{P}_t - \mathbf{P}_{t'}\|_\infty + 4U_R M\|\mathbf{P}_t - \mathbf{P}_{t'}\|_\infty\right)$$

$$\qquad + 2U_R\lambda^{-2}\|\bar{\mathbf{A}}^{\boldsymbol{\pi}, \mathbf{P}_t} - \bar{\mathbf{A}}^{\boldsymbol{\pi}, \mathbf{P}_{t'}}\|_2$$

$$\overset{(g)}{\leq} \lambda^{-1}\sqrt{|\mathcal{S}||\mathcal{A}|}\left(2\|\mathbf{r}_t - \mathbf{r}_{t'}\|_\infty + L_P\|\mathbf{P}_t - \mathbf{P}_{t'}\|_\infty + 4U_R M\|\mathbf{P}_t - \mathbf{P}_{t'}\|_\infty\right)$$

$$\qquad + 2U_R\lambda^{-2} \cdot 2(M+1)\sqrt{|\mathcal{S}||\mathcal{A}|}\|\mathbf{P}_t - \mathbf{P}_{t'}\|_\infty$$

$$\leq G_R\|\mathbf{r}_t - \mathbf{r}_{t'}\|_\infty + G_P\|\mathbf{P}_t - \mathbf{P}_{t'}\|_\infty$$

where $(a)$ is because $\mathbb{E}\left[\mathbf{r}(O) - \mathbf{J}(O) + \mathbf{A}(O)\mathbf{Q}^{\boldsymbol{\pi}}\right] = 0$ (see TD limiting point (6) in Section 5.1) $(b)$ is from Lemma 5.1, $(c)$ is by Lemma G.5, $(d)$ is due to lemma I.1, $(e)$ is using the same process as the last step for the second term, $(f)$ is because $\|\mathbf{X}^\dagger - \mathbf{Y}^\dagger\|_2 \leq \|\mathbf{X}^\dagger(\mathbf{X} - \mathbf{Y})\mathbf{Y}^\dagger\|_2 \leq \|\mathbf{X}^\dagger\|_2\|\mathbf{X} - \mathbf{Y}\|_2\|\mathbf{Y}^\dagger\|_2$ and $(g)$ is by Lemma I.3 and Lemma I.1.

$\square$

We combine Lemma G.6 and Lemma G.7 to bound the difference between the value functions resulting from, respectively, following policy $\boldsymbol{\pi}_t$ at time $t$ (in environment $\{\mathbf{r}_t, \mathbf{P}_t\}$) and policy $\boldsymbol{\pi}_{t+1}$ at time $t + 1$ (in environment $\{\mathbf{r}_{t+1}, \mathbf{P}_{t+1}\}$).

**Lemma G.8.** *For any $t \geq 0$, we have*

$$\|\Pi_E\left[\mathbf{Q}_{t+1}^{\boldsymbol{\pi}_{t+1}} - \mathbf{Q}_t^{\boldsymbol{\pi}_t}\right]\|_2 \leq G_R\|\mathbf{r}_{t+1} - \mathbf{r}_t\|_\infty + G_P\|\mathbf{P}_{t+1} - \mathbf{P}_t\|_\infty + G_{\boldsymbol{\pi}}B_2\beta.$$

*See Appendix B for constants.*

*Proof.*
$$\|\Pi_E\left[\mathbf{Q}_{t+1}^{\boldsymbol{\pi}_{t+1}} - \mathbf{Q}_t^{\boldsymbol{\pi}_t}\right]\|_2 \leq \|\Pi_E\left[\mathbf{Q}_{t+1}^{\boldsymbol{\pi}_{t+1}} - \mathbf{Q}_t^{\boldsymbol{\pi}_{t+1}}\right]\|_2 + \|\Pi_E\left[\mathbf{Q}_t^{\boldsymbol{\pi}_{t+1}} - \mathbf{Q}_t^{\boldsymbol{\pi}_t}\right]\|_2$$

$$\overset{(a)}{\leq} G_R\|\mathbf{r}_{t+1} - \mathbf{r}_t\|_\infty + G_P\|\mathbf{P}_{t+1} - \mathbf{P}_t\|_\infty + G_{\boldsymbol{\pi}}\|\boldsymbol{\pi}_{t+1} - \boldsymbol{\pi}_t\|_2$$

$$\overset{(b)}{\leq} G_R\|\mathbf{r}_{t+1} - \mathbf{r}_t\|_\infty + G_P\|\mathbf{P}_{t+1} - \mathbf{P}_t\|_\infty + G_{\boldsymbol{\pi}}B_2\beta$$

where $(a)$ is by Lemma G.7 and Lemma G.6 and $(b)$ is from Lemma G.2. $\square$

**Lemma G.9.** *If Assumption 1 holds, for any $t > \tau$, we have*

$$\mathbb{E}\left[\Gamma(\boldsymbol{\pi}_t, \mathbf{P}_t, \mathbf{r}_t, \boldsymbol{\psi}_t, O_t)\right] \leq B_3\beta(\tau + 1)^2 + B_4\alpha\tau + B_5\Delta_{R,t-\tau+1,t} + B_6\tau\Delta_{P,t-\tau+1,t}$$

*where $B_3 = (F_{1\boldsymbol{\pi}} + F_2 G_{\boldsymbol{\pi}} + F_3\sqrt{|\mathcal{S}||\mathcal{A}|} + F_4)B_2$, $B_4 = F_2(2U_R + 2U_Q)$, $B_5 = F_2 G_R$ and $B_6 = F_{1\mathbf{P}} + F_2 G_P + F_3$, $\Delta_{R,t-\tau+1,t} = \sum_{i=t-\tau+1}^t \|\mathbf{r}_i - \mathbf{r}_{i-1}\|_\infty$ and $\Delta_{P,t-\tau+1,t} = \sum_{i=t-\tau+1}^t \|\mathbf{P}_i - \mathbf{P}_{i-1}\|_\infty$.*

*Proof.* Recall from Appendix A, the definition

$$\Gamma(\boldsymbol{\pi}, \mathbf{P}, \mathbf{r}, \boldsymbol{\psi}, O) = \boldsymbol{\psi}^\top\left(\mathbf{r}(O) - \mathbf{J}^{\boldsymbol{\pi},\mathbf{P},\mathbf{r}}(O) + \mathbf{A}(O)\mathbf{Q}^{\boldsymbol{\pi},\mathbf{P},\mathbf{r}}\right) + \boldsymbol{\psi}^\top\left(\mathbf{A}(O) - \bar{\mathbf{A}}^{\boldsymbol{\pi},\mathbf{P}}\right)\boldsymbol{\psi}.$$

We first decompose $\Gamma(\cdot)$ into the following four terms

$$\mathbb{E}\left[\Gamma(\boldsymbol{\pi}_t, \mathbf{P}_t, \mathbf{r}_t, \boldsymbol{\psi}_t, O_t)\right] \leq \underbrace{\mathbb{E}\left[\Gamma(\boldsymbol{\pi}_t, \mathbf{P}_t, \mathbf{r}_t, \boldsymbol{\psi}_t, O_t) - \Gamma(\boldsymbol{\pi}_{t-\tau-1}, \mathbf{P}_{t-\tau}, \mathbf{r}_t, \boldsymbol{\psi}_t, O_t)\right]}_{I_1}$$

$$+ \underbrace{\mathbb{E}\left[\Gamma(\boldsymbol{\pi}_{t-\tau-1}, \mathbf{P}_{t-\tau}, \mathbf{r}_t, \boldsymbol{\psi}_t, O_t) - \Gamma(\boldsymbol{\pi}_{t-\tau-1}, \mathbf{P}_{t-\tau}, \mathbf{r}_t, \boldsymbol{\psi}_{t-\tau}, O_t)\right]}_{I_2}$$

$$+ \underbrace{\mathbb{E}\left[\Gamma(\boldsymbol{\pi}_{t-\tau-1}, \mathbf{P}_{t-\tau}, \mathbf{r}_t, \boldsymbol{\psi}_{t-\tau}, O_t) - \Gamma(\boldsymbol{\pi}_{t-\tau-1}, \mathbf{P}_{t-\tau}, \mathbf{r}_t, \boldsymbol{\psi}_{t-\tau}, \tilde{O}_t)\right]}_{I_3}$$

$$+ \underbrace{\mathbb{E}\left[\Gamma(\boldsymbol{\pi}_{t-\tau-1}, \mathbf{P}_{t-\tau}, \mathbf{r}_t, \boldsymbol{\psi}_{t-\tau}, \tilde{O}_t)\right]}_{I_4}.$$

We now bound each term as follows.

$$I_1 \overset{(a)}{\leq} F_{1\boldsymbol{\pi}}\mathbb{E}\left[\|\boldsymbol{\pi}_t - \boldsymbol{\pi}_{t-\tau-1}\|_2\right] + F_{1\mathbf{P}}\|\mathbf{P}_t - \mathbf{P}_{t-\tau}\|_\infty$$

$$\leq F_{1\boldsymbol{\pi}}\mathbb{E}\left[\sum_{i=t-\tau}^t \|\boldsymbol{\pi}_i - \boldsymbol{\pi}_{i-1}\|_2\right] + F_{1\mathbf{P}}\sum_{i=t-\tau+1}^t \|\mathbf{P}_i - \mathbf{P}_{i-1}\|_\infty$$

$$\overset{(b)}{\leq} F_{1\boldsymbol{\pi}}B_2\beta(\tau + 1) + F_{1\mathbf{P}}\Delta_{P,t-\tau+1,t}$$

where $(a)$ is by Lemma H.3 and $(b)$ is due to Lemma H.2. For the second term, we have

For the third term, we have

$$I_3 \overset{(e)}{\leq} F_3 \sqrt{|\mathcal{S}||\mathcal{A}|} \mathbb{E}\left[\sum_{i=t-\tau}^{t} \|\boldsymbol{\pi}_i - \boldsymbol{\pi}_{t-\tau-1}\|_2 \Big| \mathcal{F}_{t-\tau}\right] + F_3 \sum_{i=t-\tau}^{t} \|\mathbf{P}_i - \mathbf{P}_{t-\tau}\|_\infty$$

$$\overset{(f)}{\leq} F_3 \sqrt{|\mathcal{S}||\mathcal{A}|} B_2 \beta(\tau+1)^2 + F_3 \tau \Delta_{P,t-\tau+1,t}.$$

where $(e)$ is due to Lemma H.5 and $(f)$ follows from lemma H.2. For the last term, by Lemma H.6, we have

$$I_4 \leq F_4 m \rho^\tau.$$

We get the final result by putting all the four terms together. $\qquad\square$

**Lemma G.10.** *If Assumption 1 holds, for any $t > \tau$, we have*

$$\mathbb{E}[\Lambda(\boldsymbol{\pi}_t, \mathbf{P}_t, \mathbf{r}_t, \eta_t, O_t)] \leq B_7 \beta(\tau+1)^2 + F_6 |\eta_t - \eta_{t-\tau}| + B_8 \tau \Delta_{P,t-\tau+1,t}$$

*where $B_7 = (F_5 L_{\boldsymbol{\pi}} + F_7 \sqrt{|\mathcal{S}||\mathcal{A}|} + F_8) B_2$, $B_8 = F_7 + F_5 L_P$ and $\Delta_{P,t-\tau+1,t} = \sum_{i=t-\tau+1}^{t} \|\mathbf{P}_i - \mathbf{P}_{i-1}\|_\infty$.*

*Proof.* Recall from Appendix A, the definition

$$\Lambda(\boldsymbol{\pi}, \mathbf{P}, \mathbf{r}, \eta, O) = (\eta - J^{\boldsymbol{\pi},\mathbf{P},\mathbf{r}})(r(s,a) - J^{\boldsymbol{\pi},\mathbf{P},\mathbf{r}})$$

We first decompose $\Lambda(\boldsymbol{\pi}_t, \mathbf{P}_t, \mathbf{r}_t, \eta_t, O_t)$ into the following four terms

$$\mathbb{E}[\Lambda(\boldsymbol{\pi}_t, \mathbf{P}_t, \mathbf{r}_t, \eta_t, O_t)] = \underbrace{\mathbb{E}[\Lambda(\boldsymbol{\pi}_t, \mathbf{P}_t, \mathbf{r}_t, \eta_t, O_t) - \Lambda(\boldsymbol{\pi}_{t-\tau-1}, \mathbf{P}_{t-\tau}, \mathbf{r}_t, \eta_t, O_t)]}_{I_1}$$

$$+ \underbrace{\mathbb{E}[\Lambda(\boldsymbol{\pi}_{t-\tau-1}, \mathbf{P}_{t-\tau}, \mathbf{r}_t, \eta_t, O_t) - \Lambda(\boldsymbol{\pi}_{t-\tau-1}, \mathbf{P}_{t-\tau}, \mathbf{r}_t, \eta_{t-\tau}, O_t)]}_{I_2}$$

$$+ \underbrace{\mathbb{E}[\Lambda(\boldsymbol{\pi}_{t-\tau-1}, \mathbf{P}_{t-\tau}, \mathbf{r}_t, \eta_{t-\tau}, O_t) - \Lambda(\boldsymbol{\pi}_{t-\tau-1}, \mathbf{P}_{t-\tau}, \mathbf{r}_t, \eta_{t-\tau}, \tilde{O}_t)]}_{I_3}$$

$$+ \underbrace{\mathbb{E}[\Lambda(\boldsymbol{\pi}_{t-\tau-1}, \mathbf{P}_{t-\tau}, \mathbf{r}_t, \eta_{t-\tau}, \tilde{O}_t)]}_{I_4}.$$

We now bound each term as follows.

$$I_1 \overset{(a)}{\leq} F_5 L_{\boldsymbol{\pi}} \mathbb{E}[\|\boldsymbol{\pi}_t - \boldsymbol{\pi}_{t-\tau-1}\|_2] + F_5 L_P \|\mathbf{P}_t - \mathbf{P}_{t-\tau}\|_\infty$$

$$\leq F_5 L_{\boldsymbol{\pi}} \mathbb{E}\left[\sum_{i=t-\tau}^{t} \|\boldsymbol{\pi}_i - \boldsymbol{\pi}_{i-1}\|_2\right] + F_5 L_P \sum_{i=t-\tau+1}^{t} \|\mathbf{P}_i - \mathbf{P}_{i-1}\|_\infty$$

$$\overset{(b)}{\leq} F_5 L_{\boldsymbol{\pi}} B_2 \beta(\tau+1) + F_5 L_P \Delta_{P,t-\tau+1,t}$$

where $(a)$ follows from Lemma H.8, and $(b)$ is due to Lemma H.2. For the second term $I_2$, we have

$$I_2 \overset{(c)}{\leq} F_6 |\eta_t - \eta_{t-\tau}|$$

where $(c)$ is by Lemma H.9. For the third term $I_3$, we have

$$I_3 \overset{(d)}{\leq} F_7 \sqrt{|\mathcal{S}||\mathcal{A}|} \mathbb{E}\left[\sum_{i=t-\tau}^{t} \|\boldsymbol{\pi}_i - \boldsymbol{\pi}_{t-\tau-1}\|_2 \Big| \mathcal{F}_{t-\tau}\right] + F_7 \sum_{i=t-\tau}^{t} \|\mathbf{P}_i - \mathbf{P}_{t-\tau}\|_\infty$$

$$\overset{(e)}{\leq} F_7 \sqrt{|\mathcal{S}||\mathcal{A}|} B_2 \beta(\tau+1)^2 + F_7 \Delta_{P,t-\tau+1,t}.$$

where $(d)$ is due to Lemma H.10 and $(e)$ follows from lemma H.2. For the last term, by Lemma H.11, we have

$$I_4 \leq F_8 m \rho^\tau.$$

We get the final result by putting all the four terms together. $\qquad\square$

## H    AUXILIARY LEMMAS

**Lemma H.1.** *For all $t, s, a$ in Algorithm 1, we have*

$$\pi_t(a|s) \geq \frac{e^{-2U_Q/\epsilon}}{|\mathcal{A}|}.$$

*Proof.* From the actor update (5), we have the following recursive equation

$$\boldsymbol{\theta}_{t+1} = (1 - \beta\epsilon)\boldsymbol{\theta}_t + \beta\mathbf{Q}_t$$

$$\boldsymbol{\theta}_{t+1} = (1 - \beta\epsilon)^{t+1}\boldsymbol{\theta}_0 + \beta\sum_{i=0}^{t}(1 - \beta\epsilon)^{t-i-1}\mathbf{Q}_i.$$

Starting from $\boldsymbol{\theta}_0 = 0$, it holds that

$$-\beta U_Q\frac{1 - (1 - \beta\epsilon)^t}{\beta\epsilon} \leq \theta_{t+1,s,a} \leq \beta U_Q\frac{1 - (1 - \beta\epsilon)^t}{\beta\epsilon}$$

$$\max_{t,s,a,a'} \theta_{t,s,a'} - \theta_{t,s,a} = \frac{2U_Q}{\epsilon}$$

We hence have $\forall t, s, a$,

$$\pi_t(a|s) = \frac{e^{\theta_{t,s,a}}}{\sum_{a'} e^{\theta_{t,s,a'}}} \geq \frac{1}{|\mathcal{A}|e^{\max_{t,s,a,a'}\theta_{t,s,a'} - \theta_{t,s,a}}} \geq \frac{e^{-2U_Q/\epsilon}}{|\mathcal{A}|}.$$

$\square$

**Lemma H.2.** *For any timesteps $t > \tau > 0$, the policies generated by Algorithm 1 satisfy*

$$\sum_{i=t-\tau}^{t} \|\boldsymbol{\pi}_i - \boldsymbol{\pi}_{t-\tau-1}\|_2 \leq B_2\beta(\tau + 1)^2$$

*and reward and transition probability matrices satisfy*

$$\sum_{i=t-\tau}^{t} \|\mathbf{r}_i - \mathbf{r}_{t-\tau}\|_\infty \leq \tau \sum_{i=t-\tau+1}^{t} \|\mathbf{r}_i - \mathbf{r}_{i-1}\|_\infty$$

$$\sum_{i=t-\tau}^{t} \|\mathbf{P}_i - \mathbf{P}_{t-\tau}\|_\infty \leq \tau \sum_{i=t-\tau+1}^{t} \|\mathbf{P}_i - \mathbf{P}_{i-1}\|_\infty.$$

*Proof.* By triangle inequality, we have

$$\sum_{i=t-\tau}^{t} \|\boldsymbol{\pi}_i - \boldsymbol{\pi}_{t-\tau-1}\|_2 \leq \sum_{i=t-\tau}^{t} \|\sum_{j=t-\tau}^{i} \boldsymbol{\pi}_j - \boldsymbol{\pi}_{j-1}\|_2$$

$$\leq \sum_{i=t-\tau}^{t}\sum_{j=t-\tau}^{i} \|\boldsymbol{\pi}_j - \boldsymbol{\pi}_{j-1}\|_2$$

$$\overset{(a)}{\leq} B_2\beta(\tau + 1)^2$$

where $(a)$ is by Lemma G.2. The rest follow similarly using triangle inequality. $\square$

**Lemma H.3.** *For any $\boldsymbol{\pi}, \boldsymbol{\pi}', \mathbf{P}, \mathbf{P}', \mathbf{r}, \boldsymbol{\psi}$ and $O = (s, a, s', a')$, we have*

$$|\Gamma(\boldsymbol{\pi}, \mathbf{P}, \mathbf{r}, \boldsymbol{\psi}, O) - \Gamma(\boldsymbol{\pi}', \mathbf{P}', \mathbf{r}, \boldsymbol{\psi}, O)| \leq F_{1\boldsymbol{\pi}}\|\boldsymbol{\pi} - \boldsymbol{\pi}'\|_2 + F_{1\mathbf{P}}\|\mathbf{P} - \mathbf{P}'\|_\infty$$

*where $F_{1\boldsymbol{\pi}} = 2U_Q L_{\boldsymbol{\pi}} + 4U_Q G_{\boldsymbol{\pi}} + 8U_Q^2(M + 2)|\mathcal{S}||\mathcal{A}|$, $F_{1\mathbf{P}} = 2U_Q L_P + 4U_Q G_P + 8U_Q^2(M + 1)\sqrt{|\mathcal{S}||\mathcal{A}|}$.*

*Proof.*

$$|\Gamma(\boldsymbol{\pi}, \mathbf{P}, \mathbf{r}, \boldsymbol{\psi}, O) - \Gamma(\boldsymbol{\pi}', \mathbf{P}', \mathbf{r}, \boldsymbol{\psi}, O)|$$

$$= |\boldsymbol{\psi}^\top (\mathbf{J}^{\boldsymbol{\pi}', \mathbf{P}', \mathbf{r}}(O) - \mathbf{J}^{\boldsymbol{\pi}, \mathbf{P}, \mathbf{r}}(O)) + \boldsymbol{\psi}^\top \mathbf{A}(O) \left( \mathbf{Q}^{\boldsymbol{\pi}, \mathbf{P}, \mathbf{r}} - \mathbf{Q}^{\boldsymbol{\pi}', \mathbf{P}', \mathbf{r}} \right) + \boldsymbol{\psi}^\top \left( \bar{\mathbf{A}}^{\boldsymbol{\pi}', \mathbf{P}'} - \bar{\mathbf{A}}^{\boldsymbol{\pi}, \mathbf{P}} \right) \boldsymbol{\psi}|$$

$$\overset{(a)}{\leq} \|\boldsymbol{\psi}\|_\infty |J^{\boldsymbol{\pi}', \mathbf{P}', \mathbf{r}} - J^{\boldsymbol{\pi}, \mathbf{P}, \mathbf{r}}| + \|\boldsymbol{\psi}\|_2 \|\mathbf{A}(O)\|_2 \left\| \mathbf{Q}^{\boldsymbol{\pi}, \mathbf{P}, \mathbf{r}} - \mathbf{Q}^{\boldsymbol{\pi}', \mathbf{P}', \mathbf{r}} \right\|_2$$
$$+ \|\boldsymbol{\psi}\|_\infty \left\| \bar{\mathbf{A}}^{\boldsymbol{\pi}', \mathbf{P}'} - \bar{\mathbf{A}}^{\boldsymbol{\pi}, \mathbf{P}} \right\|_\infty \|\boldsymbol{\psi}\|_1$$

$$\overset{(b)}{\leq} 2U_Q L_{\boldsymbol{\pi}} \|\boldsymbol{\pi} - \boldsymbol{\pi}'\|_2 + 2U_Q L_P \|\mathbf{P} - \mathbf{P}'\|_\infty + \|\boldsymbol{\psi}\|_2 \|\mathbf{A}(O)\|_2 \left\| \mathbf{Q}^{\boldsymbol{\pi}, \mathbf{P}, \mathbf{r}} - \mathbf{Q}^{\boldsymbol{\pi}', \mathbf{P}', \mathbf{r}} \right\|_2$$
$$+ \|\boldsymbol{\psi}\|_\infty \left\| \bar{\mathbf{A}}^{\boldsymbol{\pi}', \mathbf{P}'} - \bar{\mathbf{A}}^{\boldsymbol{\pi}, \mathbf{P}} \right\|_\infty \|\boldsymbol{\psi}\|_1$$

$$\overset{(c)}{\leq} 2U_Q L_{\boldsymbol{\pi}} \|\boldsymbol{\pi} - \boldsymbol{\pi}'\|_2 + 2U_Q L_P \|\mathbf{P} - \mathbf{P}'\|_\infty + 4U_Q \cdot G_{\boldsymbol{\pi}} \|\boldsymbol{\pi} - \boldsymbol{\pi}'\|_2 + 4U_Q G_P \|\mathbf{P} - \mathbf{P}'\|_\infty$$
$$+ \|\boldsymbol{\psi}\|_\infty \left\| \bar{\mathbf{A}}^{\boldsymbol{\pi}', \mathbf{P}'} - \bar{\mathbf{A}}^{\boldsymbol{\pi}, \mathbf{P}} \right\|_\infty \|\boldsymbol{\psi}\|_1$$

$$\overset{(d)}{\leq} 2U_Q L_{\boldsymbol{\pi}} \|\boldsymbol{\pi} - \boldsymbol{\pi}'\|_2 + 2U_Q L_P \|\mathbf{P} - \mathbf{P}'\|_\infty + 4U_Q G_{\boldsymbol{\pi}} \|\boldsymbol{\pi} - \boldsymbol{\pi}'\|_2 + 4U_Q G_P \|\mathbf{P} - \mathbf{P}'\|_\infty$$
$$+ 2U_Q \cdot 2d_{TV} \left( d^{\boldsymbol{\pi}', \mathbf{P}'} \otimes \boldsymbol{\pi}' \otimes \mathbf{P}' \otimes \boldsymbol{\pi}', d^{\boldsymbol{\pi}, \mathbf{P}} \otimes \boldsymbol{\pi} \otimes \mathbf{P} \otimes \boldsymbol{\pi} \right) \cdot 2U_Q \sqrt{|\mathcal{S}||\mathcal{A}|}$$

$$\overset{(e)}{\leq} 2U_Q L_{\boldsymbol{\pi}} \|\boldsymbol{\pi} - \boldsymbol{\pi}'\|_2 + 2U_Q L_P \|\mathbf{P} - \mathbf{P}'\|_\infty + 4U_Q G_{\boldsymbol{\pi}} \|\boldsymbol{\pi} - \boldsymbol{\pi}'\|_2 + 4U_Q G_P \|\mathbf{P} - \mathbf{P}'\|_\infty$$
$$+ 8U_Q^2 (M+2)|\mathcal{S}||\mathcal{A}| \|\boldsymbol{\pi} - \boldsymbol{\pi}'\|_2 + 8U_Q^2 (M+1) \sqrt{|\mathcal{S}||\mathcal{A}|} \|\boldsymbol{\pi} - \boldsymbol{\pi}'\|_\infty$$

where $(a)$ follows from Holder's inequality; $(b)$ is due to Lemma G.5; $(c)$ is by Lemma G.8 and Lemma I.3 ($\|\mathbf{A}(O)\|_1 \leq 1$); $(d)$ is by Lemma I.3 and $(e)$ uses Lemma I.1. $\qquad\square$

**Lemma H.4.** *For any $\boldsymbol{\pi}, \mathbf{P}, \mathbf{r}, \boldsymbol{\psi}, \boldsymbol{\psi}'$ and $O = (s, a, s', a')$, we have*

$$|\Gamma(\boldsymbol{\pi}, \mathbf{P}, \mathbf{r}, \boldsymbol{\psi}, O) - \Gamma(\boldsymbol{\pi}, \mathbf{P}, \mathbf{r}, \boldsymbol{\psi}', O)| \leq F_2 \|\boldsymbol{\psi} - \boldsymbol{\psi}'\|_2$$

*where $F_2 = 2U_R + 18U_Q$.*

*Proof.*

$$|\Gamma(\boldsymbol{\pi}, \mathbf{P}, \mathbf{r}, \boldsymbol{\psi}, O) - \Gamma(\boldsymbol{\pi}, \mathbf{P}, \mathbf{r}, \boldsymbol{\psi}', O)|$$
$$\leq \left( \|\mathbf{r}(O)\|_2 + \|\mathbf{J}^{\boldsymbol{\pi}, \mathbf{P}, \mathbf{r}}(O)\|_2 + \|\mathbf{A}(O)\|_2 \|\mathbf{Q}^{\boldsymbol{\pi}, \mathbf{P}, \mathbf{r}}\|_2 \right) \|\boldsymbol{\psi} - \boldsymbol{\psi}'\|_2$$
$$+ \|\mathbf{A}(O) - \bar{\mathbf{A}}^{\boldsymbol{\pi}, \mathbf{P}}\|_2 \|\boldsymbol{\psi} - \boldsymbol{\psi}'\|_2 (\|\boldsymbol{\psi}\|_2 + \|\boldsymbol{\psi}'\|_2)$$
$$\leq (2U_R + 18U_Q) \|\boldsymbol{\psi} - \boldsymbol{\psi}'\|_2.$$

$\qquad\square$

**Lemma H.5.** *Consider an observation from the original Markov chain by $O_t = (s_t, a_t, s_{t+1}, a_{t+1})$ and auxiliary Markov chain by $\tilde{O}_t = (\tilde{s}_t, \tilde{a}_t, \tilde{s}_{t+1}, \tilde{a}_{t+1})$. Conditioned on $\mathcal{F}_{t-\tau} = \{s_{t-\tau}, \boldsymbol{\pi}_{t-\tau-1}, \mathbf{P}_{t-\tau}\}$, we have*

$$\mathbb{E}\left[ \Gamma(\boldsymbol{\pi}_{t-\tau-1}, \mathbf{P}_{t-\tau}, \mathbf{r}_t, \boldsymbol{\psi}_{t-\tau}, O_t) - \Gamma(\boldsymbol{\pi}_{t-\tau-1}, \mathbf{P}_{t-\tau}, \mathbf{r}_t, \boldsymbol{\psi}_{t-\tau}, \tilde{O}_t) \big| \mathcal{F}_{t-\tau} \right]$$
$$\leq F_3 \sqrt{|\mathcal{S}||\mathcal{A}|} \mathbb{E}\left[ \sum_{i=t-\tau}^t \|\boldsymbol{\pi}_i - \boldsymbol{\pi}_{t-\tau-1}\|_2 \big| \mathcal{F}_{t-\tau} \right] + F_3 \sum_{i=t-\tau}^t \|\mathbf{P}_i - \mathbf{P}_{t-\tau}\|_\infty$$

*where $F_3 = 16U_R U_Q + 24U_Q^2 \sqrt{|\mathcal{S}||\mathcal{A}|}$.*

*Proof.* Consider the original and auxiliary Markov chains whose construction is described in Appendix A.

$$\mathbb{E}\left[ \Gamma(\boldsymbol{\pi}_{t-\tau-1}, \mathbf{P}_{t-\tau}, \mathbf{r}_t, \boldsymbol{\psi}_{t-\tau}, O_t) - \Gamma(\boldsymbol{\pi}_{t-\tau-1}, \mathbf{P}_{t-\tau}, \mathbf{r}_t, \boldsymbol{\psi}_{t-\tau}, \tilde{O}_t) \big| \mathcal{F}_{t-\tau} \right]$$
$$= \boldsymbol{\psi}_{t-\tau}^\top \mathbb{E}\left[ \mathbf{r}_t(O_t) - \mathbf{r}_t(\tilde{O}_t) + \mathbf{J}^{\boldsymbol{\pi}_{t-\tau-1}, \mathbf{P}_{t-\tau}, \mathbf{r}_t}(\tilde{O}_t) - \mathbf{J}^{\boldsymbol{\pi}_{t-\tau-1}, \mathbf{P}_{t-\tau}, \mathbf{r}_t}(O_t) \big| \mathcal{F}_{t-\tau} \right]$$

$$+ \boldsymbol{\psi}_{t-\tau}^{\top} \mathbb{E}\left[\left(\mathbf{A}(O_t) - \mathbf{A}(\tilde{O}_t)\right) \mathbf{Q}^{\boldsymbol{\pi}_{t-\tau-1}, \mathbf{P}_{t-\tau}, \mathbf{r}_t} \big| \mathcal{F}_{t-\tau}\right]$$

$$+ \boldsymbol{\psi}_{t-\tau}^{\top} \mathbb{E}\left[\left(\mathbf{A}(O_t) - \mathbf{A}(\tilde{O}_t)\right) \big| \mathcal{F}_{t-\tau}\right] \boldsymbol{\psi}_{t-\tau}$$

$$\leq \|\boldsymbol{\psi}_{t-\tau}\|_{\infty} \left\|\mathbb{E}\left[\mathbf{r}_t(O_t) - \mathbf{r}_t(\tilde{O}_t) + \mathbf{J}_t^{\boldsymbol{\pi}_{t-\tau-1}}(\tilde{O}_t) - \mathbf{J}_t^{\boldsymbol{\pi}_{t-\tau-1}}(O_t) \big| \mathcal{F}_{t-\tau}\right]\right\|_1$$

$$+ \|\boldsymbol{\psi}_{t-\tau}\|_{\infty} \left\|\mathbb{E}\left[\mathbf{A}(O_t) - \mathbf{A}(\tilde{O}_t) \big| \mathcal{F}_{t-\tau}\right]\right\|_1 \|\mathbf{Q}^{\boldsymbol{\pi}_{t-\tau-1}, \mathbf{P}_{t-\tau}, \mathbf{r}_t}\|_1$$

$$+ \|\boldsymbol{\psi}_{t-\tau}\|_{\infty} \left\|\mathbb{E}\left[\mathbf{A}(O_t) - \mathbf{A}(\tilde{O}_t) \big| \mathcal{F}_{t-\tau}\right]\right\|_1 \|\boldsymbol{\psi}_{t-\tau}\|_1$$

$$\leq 2U_Q \cdot 4U_R \cdot 2d_{TV}\left(P(O_t \in \cdot | \mathcal{F}_{t-\tau}), P(\tilde{O}_t \in \cdot | \mathcal{F}_{t-\tau})\right)$$

$$+ 2U_Q \cdot 4d_{TV}\left(P(O_t \in \cdot | \mathcal{F}_{t-\tau}), P(\tilde{O}_t \in \cdot | \mathcal{F}_{t-\tau})\right) \cdot U_Q \sqrt{|\mathcal{S}||\mathcal{A}|}$$

$$+ 2U_Q \cdot 4d_{TV}\left(P(O_t \in \cdot | \mathcal{F}_{t-\tau}), P(\tilde{O}_t \in \cdot | \mathcal{F}_{t-\tau})\right) \cdot 2U_Q \sqrt{|\mathcal{S}||\mathcal{A}|}$$

$$\leq (16U_R U_Q + 24U_Q^2 \sqrt{|\mathcal{S}||\mathcal{A}|}) \left(\sqrt{|\mathcal{S}||\mathcal{A}|} \mathbb{E}\left[\sum_{i=t-\tau}^{t} \|\boldsymbol{\pi}_i - \boldsymbol{\pi}_{t-\tau-1}\|_2 \Big| \mathcal{F}_{t-\tau}\right] + \sum_{i=t-\tau}^{t} \|\mathbf{P}_i - \mathbf{P}_{t-\tau}\|_{\infty}\right)$$

where the last inequality is from Lemma I.2. $\qquad\square$

**Lemma H.6.** *Consider an observation from the original Markov chain by $O_t = (s_t, a_t, s_{t+1}, a_{t+1})$ and auxiliary Markov chain by $\tilde{O}_t = (\tilde{s}_t, \tilde{a}_t, \tilde{s}_{t+1}, \tilde{a}_{t+1})$. Conditioned on $\mathcal{F}_{t-\tau} = \{s_{t-\tau}, \boldsymbol{\pi}_{t-\tau-1}, \mathbf{P}_{t-\tau}\}$, we have*

$$\mathbb{E}\left[\Gamma(\boldsymbol{\pi}_{t-\tau-1}, \mathbf{P}_{t-\tau}, \mathbf{r}_t, \boldsymbol{\psi}_{t-\tau}, \tilde{O}_t) \big| \mathcal{F}_{t-\tau}\right] \leq F_4 m \rho^{\tau}$$

*where $F_4 = 8U_R U_Q + 24U_Q^2 \sqrt{|\mathcal{S}||\mathcal{A}|}$.*

*Proof.* Consider the original and auxiliary Markov chains whose construction is described in Appendix A. Also, consider the observation tuple $O_t' = (s_t', a_t', s_{t+1}', a_{t+1}')$ where $s_t' \sim d^{\boldsymbol{\pi}_{t-\tau-1}, \mathbf{P}_{t-\tau}}(\cdot)$, $a_t' \sim \boldsymbol{\pi}_{t-\tau-1}(\cdot | s_t')$, $s_{t+1}' \sim \mathbf{P}_{t-\tau}(\cdot | s_t', a_t')$ and $a_{t+1}' \sim \boldsymbol{\pi}_{t-\tau-1}(\cdot | s_{t+1}')$. From the definition of $\Gamma(\cdot)$ and the TD limit point equation (6), it follows that

$$\mathbb{E}\left[\Gamma(\boldsymbol{\pi}_{t-\tau-1}, \mathbf{P}_{t-\tau}, \mathbf{r}_t, \boldsymbol{\psi}_{t-\tau}, O_t') \big| \mathcal{F}_{t-\tau}\right] = 0$$

Hence, we have

$$\mathbb{E}\left[\Gamma(\boldsymbol{\pi}_{t-\tau-1}, \mathbf{P}_{t-\tau}, \mathbf{r}_t, \boldsymbol{\psi}_{t-\tau}, \tilde{O}_t) \big| \mathcal{F}_{t-\tau}\right]$$

$$\leq \mathbb{E}\left[\Gamma(\boldsymbol{\pi}_{t-\tau-1}, \mathbf{P}_{t-\tau}, \mathbf{r}_t, \boldsymbol{\psi}_{t-\tau}, \tilde{O}_t) - \Gamma(\boldsymbol{\pi}_{t-\tau-1}, \mathbf{P}_{t-\tau}, \mathbf{r}_t, \boldsymbol{\psi}_{t-\tau}, O_t') \big| \mathcal{F}_{t-\tau}\right]$$

$$\leq \|\boldsymbol{\psi}_{t-\tau}\|_{\infty} \left\|\mathbb{E}\left[\mathbf{r}_t(\tilde{O}_t) - \mathbf{J}^{\boldsymbol{\pi}_{t-\tau-1}, \mathbf{P}_{t-\tau}, \mathbf{r}_t}(\tilde{O}_t) - \mathbf{r}_t(O_t') + \mathbf{J}^{\boldsymbol{\pi}_{t-\tau-1}, \mathbf{P}_{t-\tau}, \mathbf{r}_t}(O_t') \big| \mathcal{F}_{t-\tau}\right]\right\|_1$$

$$+ \|\boldsymbol{\psi}_{t-\tau}\|_{\infty} \left\|\mathbb{E}\left[\left(\mathbf{A}(\tilde{O}_t) - \mathbf{A}(O_t')\right) \mathbf{Q}^{\boldsymbol{\pi}_{t-\tau-1}, \mathbf{P}_{t-\tau}, \mathbf{r}_t} \big| \mathcal{F}_{t-\tau}\right]\right\|_1$$

$$+ \|\boldsymbol{\psi}_{t-\tau}\|_{\infty} \left\|\mathbb{E}\left[\left(\mathbf{A}(\tilde{O}_t) - \mathbf{A}(O_t')\right) \boldsymbol{\psi}_{t-\tau} \big| \mathcal{F}_{t-\tau}\right]\right\|_1$$

$$\leq 2U_Q \cdot 4U_R \cdot 2d_{TV}\left(P(\tilde{O}_t \in \cdot | \mathcal{F}_{t-\tau}), P(O_t' \in \cdot | \mathcal{F}_{t-\tau})\right)$$

$$+ 2U_Q \cdot 4d_{TV}\left(P(\tilde{O}_t \in \cdot | \mathcal{F}_{t-\tau}), P(O_t' \in \cdot | \mathcal{F}_{t-\tau})\right) \cdot U_Q \sqrt{|\mathcal{S}||\mathcal{A}|}$$

$$+ 2U_Q \cdot 4d_{TV}\left(P(\tilde{O}_t \in \cdot | \mathcal{F}_{t-\tau}), P(O_t' \in \cdot | \mathcal{F}_{t-\tau})\right) \cdot 2U_Q \sqrt{|\mathcal{S}||\mathcal{A}|}$$

$$= F_4 \sum_{s,a,s',a'} |P(\tilde{s}_t = s | \mathcal{F}_{t-\tau}) \pi_{t-\tau-1}(a|s) P_{t-\tau}(s'|s,a) \pi_{t-\tau-1}(a'|s')$$

$$- P(s_t' = s | \mathcal{F}_{t-\tau}) \pi_{t-\tau-1}(a|s) P_{t-\tau}(s'|s,a) \pi_{t-\tau-1}(a'|s')|$$

$$= F_4 \sum_{s,a,s',a'} \pi_{t-\tau-1}(a|s) P(s'|s,a) \pi_{t-\tau-1}(a'|s') |P(\tilde{s}_t = s | \mathcal{F}_{t-\tau}) - P(s_t' = s | \mathcal{F}_{t-\tau})|$$

$$= F_4 \sum_s |P(\tilde{s}_t = s|\mathcal{F}_{t-\tau}) - P(s'_t = s|\mathcal{F}_{t-\tau})|$$

$$\leq F_4 m \rho^\tau$$

where the last inequality follows from Assumption 1. $\qquad \square$

**Lemma H.7.** *Consider an observation from the original Markov chain by $O_t = (s_t, a_t, s'_t, a'_t)$ and auxiliary Markov chain by $\tilde{O}_t = (\tilde{s}_t, \tilde{a}_t, \tilde{s}_{t+1}, \tilde{a}_{t+1})$. Conditioned on $\mathcal{F}_{t-\tau} = \{s_{t-\tau}, \boldsymbol{\pi}_{t-\tau-1}, \mathbf{P}_{t-\tau}\}$, we have*

$$\mathbb{E}\left[J^{\boldsymbol{\pi}_{t-\tau-1}, \mathbf{P}_{t-\tau}, \mathbf{r}_t} - r_t(\tilde{s}_t, \tilde{a}_t)|\mathcal{F}_{t-\tau}\right] \leq 4U_R m \rho^\tau$$

*where $J^{\boldsymbol{\pi}_{t-\tau-1}, \mathbf{P}_{t-\tau}, \mathbf{r}_t} = \sum_{s,a} d^{\boldsymbol{\pi}_{t-\tau-1}, \mathbf{P}_{t-\tau}}(s)\boldsymbol{\pi}_{t-\tau-1}(a|s)r_t(s,a)$.*

*Proof.* Consider the observation tuple $O'_t = (s'_t, a'_t, s'_{t+1}, a'_{t+1})$ where $s'_t \sim d^{\boldsymbol{\pi}_{t-\tau-1}, \mathbf{P}_{t-\tau}}(\cdot)$, $a'_t \sim \boldsymbol{\pi}_{t-\tau-1}(\cdot|s'_t)$, $s'_{t+1} \sim \mathbf{P}_{t-\tau}(\cdot|s'_t, a'_t)$ and $a'_{t+1} \sim \boldsymbol{\pi}_{t-\tau-1}(\cdot|s'_{t+1})$. Then, by definition of $J^{\boldsymbol{\pi}_{t-\tau-1}, \mathbf{P}_{t-\tau}, \mathbf{r}_t}$, we have

$$\mathbb{E}\left[J^{\boldsymbol{\pi}_{t-\tau-1}, \mathbf{P}_{t-\tau}, \mathbf{r}_t} - r_t(s'_t, a'_t)|\mathcal{F}_{t-\tau}\right] = 0.$$

Hence, we have

$$\mathbb{E}\left[J^{\boldsymbol{\pi}_{t-\tau-1}, \mathbf{P}_{t-\tau}, \mathbf{r}_t} - r_t(\tilde{s}_t, \tilde{a}_t)|\mathcal{F}_{t-\tau}\right]$$
$$= \mathbb{E}\left[J^{\boldsymbol{\pi}_{t-\tau-1}, \mathbf{P}_{t-\tau}, \mathbf{r}_t} - r_t(s'_t, a'_t) - r_t(\tilde{s}_t, \tilde{a}_t) + r_t(s'_t, a'_t)|\mathcal{F}_{t-\tau}\right]$$
$$= \mathbb{E}\left[r_t(s'_t, a'_t) - r_t(\tilde{s}_t, \tilde{a}_t)|\mathcal{F}_{t-\tau}\right]$$
$$\leq 2U_R \cdot 2d_{TV}\left(d^{\boldsymbol{\pi}_{t-\tau-1}, \mathbf{P}_{t-\tau}} \otimes \boldsymbol{\pi}_{t-\tau-1}, P((\tilde{s}_t, \tilde{a}_t) \in \cdot|\mathcal{F}_{t-\tau})\right)$$
$$\overset{(a)}{\leq} 4U_R d_{TV}\left(d^{\boldsymbol{\pi}_{t-\tau-1}, \mathbf{P}_{t-\tau}}, P(\tilde{s}_t \in \cdot|\mathcal{F}_{t-\tau})\right)$$
$$\overset{(b)}{\leq} 4U_R m \rho^\tau$$

where $(a)$ follows from Lemma B.1 in (Wu et al., 2020) and $(b)$ is by Assumption 1. $\qquad \square$

**Lemma H.8.** *For any $\boldsymbol{\pi}, \boldsymbol{\pi}', \mathbf{P}, \mathbf{P}', \mathbf{r}, \eta$, and $O = (s, a, s', a')$, we have*

$$|\Lambda(\boldsymbol{\pi}, \mathbf{P}, \mathbf{r}, \eta, O) - \Lambda(\boldsymbol{\pi}', \mathbf{P}', \mathbf{r}, \eta, O)| \leq F_5 L_{\boldsymbol{\pi}}\|\boldsymbol{\pi} - \boldsymbol{\pi}'\|_2 + F_5 L_P \|\mathbf{P} - \mathbf{P}'\|_\infty,$$

*where $F_5 = 4U_R$.*

*Proof.*
$$|\Lambda(\boldsymbol{\pi}, \mathbf{P}, \mathbf{r}, \eta, O) - \Lambda(\boldsymbol{\pi}', \mathbf{P}', \mathbf{r}, \eta, O)|$$
$$\leq |(\eta - J^{\boldsymbol{\pi}, \mathbf{P}, \mathbf{r}})(r(s,a) - J^{\boldsymbol{\pi}, \mathbf{P}, \mathbf{r}}) - (\eta - J^{\boldsymbol{\pi}', \mathbf{P}', \mathbf{r}})(r(s,a) - J^{\boldsymbol{\pi}', \mathbf{P}', \mathbf{r}})|$$
$$\leq |(\eta - J^{\boldsymbol{\pi}, \mathbf{P}, \mathbf{r}})(r(s,a) - J^{\boldsymbol{\pi}, \mathbf{P}, \mathbf{r}}) - (\eta - J^{\boldsymbol{\pi}, \mathbf{P}, \mathbf{r}})(r(s,a) - J^{\boldsymbol{\pi}', \mathbf{P}', \mathbf{r}})|$$
$$\quad + |(\eta - J^{\boldsymbol{\pi}, \mathbf{P}, \mathbf{r}})(r(s,a) - J^{\boldsymbol{\pi}', \mathbf{P}', \mathbf{r}}) - (\eta - J^{\boldsymbol{\pi}', \mathbf{P}', \mathbf{r}})(r(s,a) - J^{\boldsymbol{\pi}', \mathbf{P}', \mathbf{r}})|$$
$$\leq 4U_R|J^{\boldsymbol{\pi}, \mathbf{P}, \mathbf{r}} - J^{\boldsymbol{\pi}', \mathbf{P}', \mathbf{r}}| \overset{(a)}{\leq} 4U_R L_{\boldsymbol{\pi}}\|\boldsymbol{\pi} - \boldsymbol{\pi}'\|_2 + 4U_R L_P\|\mathbf{P} - \mathbf{P}'\|_\infty$$

where $(a)$ follows from Lemma G.5. $\qquad \square$

**Lemma H.9.** *For any $\boldsymbol{\pi}, \mathbf{P}, \mathbf{r}, \eta, \eta'$ and $O = (s, a, s', a')$, we have*

$$|\Lambda(\boldsymbol{\pi}, \mathbf{P}, \mathbf{r}, \eta, O) - \Lambda(\boldsymbol{\pi}, \mathbf{P}, \mathbf{r}, \eta', O)| \leq F_6|\eta - \eta'|$$

*where $F_6 = 2U_R$.*

*Proof.* Recall the definition of $\Lambda(\cdot)$ in Appendix A. It is straightforward to see that

$$|\Lambda(\boldsymbol{\pi}, \mathbf{P}, \mathbf{r}, \eta, O) - \Lambda(\boldsymbol{\pi}, \mathbf{P}, \mathbf{r}, \eta', O)| \leq 2U_R|\eta - \eta'|$$

$\qquad \square$

**Lemma H.10.** *Consider an observation from the original Markov chain by $O_t = (s_t, a_t, s_{t+1}, a_{t+1})$ and auxiliary Markov chain by $\tilde{O}_t = (\tilde{s}_t, \tilde{a}_t, \tilde{s}_{t+1}, \tilde{a}_{t+1})$. Conditioned on $\mathcal{F}_{t-\tau} = \{s_{t-\tau}, \boldsymbol{\pi}_{t-\tau-1}, \mathbf{P}_{t-\tau}\}$, we have*

$$\mathbb{E}\left[\Lambda(\boldsymbol{\pi}_{t-\tau-1}, \mathbf{P}_{t-\tau}, \mathbf{r}_t, \eta_{t-\tau}, O_t) - \Lambda(\boldsymbol{\pi}_{t-\tau-1}, \mathbf{P}_{t-\tau}, \mathbf{r}_t, \eta_{t-\tau}, \tilde{O}_t) \big| \mathcal{F}_{t-\tau}\right]$$

$$\leq F_7 \sqrt{|\mathcal{S}||\mathcal{A}|} \mathbb{E}\left[\sum_{i=t-\tau}^t \|\boldsymbol{\pi}_i - \boldsymbol{\pi}_{t-\tau-1}\|_2 \Big| \mathcal{F}_{t-\tau}\right] + F_7 \sum_{i=t-\tau}^t \|\mathbf{P}_i - \mathbf{P}_{t-\tau}\|_\infty$$

*where $F_7 = 8U_R^2$.*

*Proof.*

$$\mathbb{E}\left[\Lambda(\boldsymbol{\pi}_{t-\tau-1}, \mathbf{P}_{t-\tau}, \mathbf{r}_t, \eta_{t-\tau}, O_t) - \Lambda(\boldsymbol{\pi}_{t-\tau-1}, \mathbf{P}_{t-\tau}, \mathbf{r}_t, \eta_{t-\tau}, \tilde{O}_t) \big| \mathcal{F}_{t-\tau}\right]$$

$$= (\eta_{t-\tau} - J^{\boldsymbol{\pi}_{t-\tau-1}, \mathbf{P}_{t-\tau}, \mathbf{r}_t}) \mathbb{E}\left[r_t(s_t, a_t) - r_t(\tilde{s}_t, \tilde{a}_t) \big| \mathcal{F}_{t-\tau}\right]$$

$$\leq 2U_R \cdot 4U_R d_{TV}\left(P(O_t \in \cdot | \mathcal{F}_{t-\tau}), P(\tilde{O}_t \in \cdot | \mathcal{F}_{t-\tau})\right)$$

$$\overset{(a)}{\leq} F_7 \sqrt{|\mathcal{S}||\mathcal{A}|} \mathbb{E}\left[\sum_{i=t-\tau}^t \|\boldsymbol{\pi}_i - \boldsymbol{\pi}_{t-\tau-1}\|_2 \Big| \mathcal{F}_{t-\tau}\right] + F_7 \sum_{i=t-\tau}^t \|\mathbf{P}_i - \mathbf{P}_{t-\tau}\|_\infty$$

where $(a)$ follows from Lemma I.2. $\qquad\square$

**Lemma H.11.** *Consider an observation from the original Markov chain by $O_t = (s_t, a_t, s_{t+1}, a_{t+1})$ and auxiliary Markov chain by $\tilde{O}_t = (\tilde{s}_t, \tilde{a}_t, \tilde{s}_{t+1}, \tilde{a}_{t+1})$. Conditioned on $\mathcal{F}_{t-\tau} = \{s_{t-\tau}, \boldsymbol{\pi}_{t-\tau-1}, \mathbf{P}_{t-\tau}\}$, we have*

$$\mathbb{E}\left[\Lambda(\boldsymbol{\pi}_{t-\tau-1}, \mathbf{P}_{t-\tau}, \mathbf{r}_t, \eta_{t-\tau}, \tilde{O}_t) \big| \mathcal{F}_{t-\tau}\right] \leq F_8 m \rho^\tau$$

*where $F_8 = 8U_R^2$.*

*Proof.* Consider the observation tuple $O_t' = (s_t', a_t', s_{t+1}', a_{t+1}')$ where $s_t' \sim d^{\boldsymbol{\pi}_{t-\tau-1}, \mathbf{P}_{t-\tau}}(\cdot)$, $a_t' \sim \boldsymbol{\pi}_{t-\tau-1}(\cdot | s_t')$, $s_{t+1}' \sim \mathbf{P}_{t-\tau}(\cdot | s_t', a_t')$ and $a_{t+1}' \sim \boldsymbol{\pi}_{t-\tau-1}(\cdot | s_{t+1}')$.

We know

$$\mathbb{E}\left[\Lambda(\boldsymbol{\pi}_{t-\tau-1}, \mathbf{P}_{t-\tau}, \mathbf{r}_t, \eta_{t-\tau}, O_t') \big| \mathcal{F}_{t-\tau}\right] = 0.$$

Hence, we have

$$\mathbb{E}\left[\Lambda(\boldsymbol{\pi}_{t-\tau-1}, \mathbf{P}_{t-\tau}, \mathbf{r}_t, \eta_{t-\tau}, \tilde{O}_t) \big| \mathcal{F}_{t-\tau}\right]$$

$$= \mathbb{E}\left[\Lambda(\boldsymbol{\pi}_{t-\tau-1}, \mathbf{P}_{t-\tau}, \mathbf{r}_t, \eta_{t-\tau}, \tilde{O}_t) - \Lambda(\boldsymbol{\pi}_{t-\tau-1}, \mathbf{P}_{t-\tau}, \mathbf{r}_t, \eta_{t-\tau}, O_t') \big| \mathcal{F}_{t-\tau}\right]$$

$$= \mathbb{E}\left[(\eta_{t-\tau} - J^{\boldsymbol{\pi}_{t-\tau-1}, \mathbf{P}_{t-\tau}, \mathbf{r}_t})(r_t(\tilde{s}_t, \tilde{a}_t) - r_t(s_t', a_t')) \big| \mathcal{F}_{t-\tau}\right]$$

$$\leq 2U_R \cdot 4U_R d_{TV}\left(d^{\boldsymbol{\pi}_{t-\tau-1}, \mathbf{P}_{t-\tau}} \otimes \boldsymbol{\pi}_{t-\tau-1}, P((\tilde{s}_t, \tilde{a}_t) \in \cdot | \mathcal{F}_{t-\tau})\right)$$

$$\overset{(a)}{\leq} 2U_R \cdot 4U_R d_{TV}\left(d^{\boldsymbol{\pi}_{t-\tau-1}, \mathbf{P}_{t-\tau}}, P(\tilde{s}_t \in \cdot | \mathcal{F}_{t-\tau})\right)$$

$$\overset{(b)}{\leq} 8U_R^2 m \rho^\tau$$

where $(a)$ follows from Lemma B.1 in (Wu et al., 2020) and $(b)$ is by Assumption 1. $\qquad\square$

## I PRELIMINARY LEMMAS

**Lemma I.1.** *For any policies $\boldsymbol{\pi}, \boldsymbol{\pi}'$ and transition probabilities matrices $\mathbf{P}, \mathbf{P}'$, it holds that*

$$d_{TV}\left(d^{\boldsymbol{\pi}, \mathbf{P}}, d^{\boldsymbol{\pi}', \mathbf{P}'}\right) \leq M\sqrt{|\mathcal{S}||\mathcal{A}|} \|\boldsymbol{\pi} - \boldsymbol{\pi}'\|_2 + M\|\mathbf{P} - \mathbf{P}'\|_\infty,$$

$$d_{TV}\left(d^{\boldsymbol{\pi},\mathbf{P}}\otimes\boldsymbol{\pi}, d^{\boldsymbol{\pi}',\mathbf{P}'}\otimes\boldsymbol{\pi}'\right) \leq (M+1)\sqrt{|\mathcal{S}||\mathcal{A}|}\|\boldsymbol{\pi}-\boldsymbol{\pi}'\|_2 + M\|\mathbf{P}-\mathbf{P}'\|_\infty,$$

$$d_{TV}\left(d^{\boldsymbol{\pi},\mathbf{P}}\otimes\boldsymbol{\pi}\otimes\mathbf{P}, d^{\boldsymbol{\pi}',\mathbf{P}'}\otimes\boldsymbol{\pi}'\otimes\mathbf{P}'\right) \leq (M+1)\sqrt{|\mathcal{S}||\mathcal{A}|}\|\boldsymbol{\pi}-\boldsymbol{\pi}'\|_2 + (M+1)\|\mathbf{P}-\mathbf{P}'\|_\infty,$$

$$d_{TV}\left(d^{\boldsymbol{\pi},\mathbf{P}}\otimes\boldsymbol{\pi}\otimes\mathbf{P}\otimes\boldsymbol{\pi}, d^{\boldsymbol{\pi}',\mathbf{P}'}\otimes\boldsymbol{\pi}'\otimes\mathbf{P}'\otimes\boldsymbol{\pi}'\right) \leq (M+2)\sqrt{|\mathcal{S}||\mathcal{A}|}\|\boldsymbol{\pi}-\boldsymbol{\pi}'\|_2 + (M+1)\|\mathbf{P}-\mathbf{P}'\|_\infty$$

*where $\otimes$ denotes the Kronecker product, and $M := \left(\lceil\log_\rho m^{-1}\rceil + \frac{1}{1-\rho}\right)$.*

*Proof.* Recall that $d^{\boldsymbol{\pi},\mathbf{P}}(\cdot)$ is the stationary distribution induced over the states by a Markov chain with transition probabilities $\mathbf{P}$ following policy $\boldsymbol{\pi}$. Define the matrices $\mathbf{K}, \mathbf{K}' \in \mathbb{R}^{|\mathcal{S}|\times|\mathcal{S}|}$ such that $\mathbf{K}(s,s') = \sum_{a\in\mathcal{A}} P(s'|s,a)\pi(a|s)$ and $\mathbf{K}'(s,s') = \sum_{a\in\mathcal{A}} P'(s'|s,a)\pi'(a|s)$. Further denote the total variation norm as $\|\cdot\|_{TV}$. Note that $\|\mathbf{P}-\mathbf{P}'\|_\infty = \max_{s,a}\sum_{s'}|P(s'|s,a) - P'(s'|s,a)|$.

From Theorem 3.1 in (Mitrophanov, 2005), we have,

$$d_{TV}\left(d^{\boldsymbol{\pi},\mathbf{P}}, d^{\boldsymbol{\pi}',\mathbf{P}'}\right) \leq M \sup_{\|q\|_{TV}=1}\left\|\int_{\mathcal{S}} q(ds)(\mathbf{K}-\mathbf{K}')(s,\cdot)\right\|_{TV}$$

$$\leq M \sup_{\|q\|_{TV}=1}\int_{\mathcal{S}}\left|\int_{\mathcal{S}} q(ds)(\mathbf{K}-\mathbf{K}')(s,ds')\right|$$

$$\leq M \sup_{\|q\|_{TV}=1}\int_{\mathcal{S}}\int_{\mathcal{S}}|q(ds)|\left|\sum_{a\in\mathcal{A}} P(ds'|s,a)\pi(a|s) - P'(ds'|s,a)\pi'(a|s)\right|$$

$$\leq M \sup_{\|q\|_{TV}=1}\int_{\mathcal{S}}\int_{\mathcal{S}}\sum_a|q(ds)|\,|P(ds'|s,a)\pi(a|s) - P(ds'|s,a)\pi'(a|s)|$$

$$+ M \sup_{\|q\|_{TV}=1}\int_{\mathcal{S}}\int_{\mathcal{S}}\sum_a|q(ds)|\,|P(ds'|s,a)\pi'(a|s) - P'(ds'|s,a)\pi'(a|s)|$$

$$\leq M\sqrt{|\mathcal{S}||\mathcal{A}|}\|\boldsymbol{\pi}-\boldsymbol{\pi}'\|_2 + M\|\mathbf{P}-\mathbf{P}'\|_\infty.$$

For the second inequality, we have,

$$d_{TV}\left(d^{\boldsymbol{\pi},\mathbf{P}}\otimes\boldsymbol{\pi}, d^{\boldsymbol{\pi}',\mathbf{P}'}\otimes\boldsymbol{\pi}'\right) \leq \frac{1}{2}\int_{\mathcal{S}}\sum_a\left|d^{\boldsymbol{\pi},\mathbf{P}}(ds)\pi(a|s) - d^{\boldsymbol{\pi}',\mathbf{P}'}(ds)\pi'(a|s)\right|$$

$$\leq \frac{1}{2}\int_{\mathcal{S}}\sum_a\left|d^{\boldsymbol{\pi},\mathbf{P}}(ds)\pi(a|s) - d^{\boldsymbol{\pi},\mathbf{P}}(ds)\pi'(a|s)\right|$$

$$+ \frac{1}{2}\int_{\mathcal{S}}\sum_a\left|d^{\boldsymbol{\pi},\mathbf{P}}(ds)\pi'(a|s) - d^{\boldsymbol{\pi}',\mathbf{P}'}(ds)\pi'(a|s)\right|$$

$$\leq \sqrt{|\mathcal{S}||\mathcal{A}|}\|\boldsymbol{\pi}-\boldsymbol{\pi}'\|_2 + d_{TV}\left(d^{\boldsymbol{\pi},\mathbf{P}}, d^{\boldsymbol{\pi}',\mathbf{P}'}\right)$$

$$\leq (M+1)\sqrt{|\mathcal{S}||\mathcal{A}|}\|\boldsymbol{\pi}-\boldsymbol{\pi}'\|_2 + M\|\mathbf{P}-\mathbf{P}'\|_\infty.$$

The rest follow in a similar manner. $\qquad\square$

**Lemma I.2.** *Consider observations $O_t = (s_t, a_t, s_{t+1}, a_{t+1})$ and $\tilde{O}_t = (\tilde{s}_t, \tilde{a}_t, \tilde{s}_{t+1}, \tilde{a}_{t+1})$ and define $\mathcal{F}_{t-\tau} := \{s_{t-\tau}, \boldsymbol{\pi}_{t-\tau-1}, \mathbf{P}_{t-\tau}\}$. We have*

$$d_{TV}\left(P(O_t \in \cdot|\mathcal{F}_{t-\tau}), P(\tilde{O}_t \in \cdot|\mathcal{F}_{t-\tau})\right) \leq \sqrt{|\mathcal{S}||\mathcal{A}|}\sum_{i=t-\tau}^t \mathbb{E}\left[\|\pi_i - \pi_{t-\tau-1}\|_2\Big|\mathcal{F}_{t-\tau}\right] + \|\mathbf{P}_i - \mathbf{P}_{t-\tau}\|_\infty.$$

*Proof.*
$$d_{TV}\left(P(O_t \in \cdot|\mathcal{F}_{t-\tau}), P(\tilde{O}_t \in \cdot|\mathcal{F}_{t-\tau})\right)$$

$$= \frac{1}{2}\sum_{s,a,s',a'}|P(\overbrace{s_t = s, a_t = a}^{\mathcal{H}_t}, s_{t+1} = s', a_{t+1} = a'|\mathcal{F}_{t-\tau})$$

$$- P(\tilde{s}_t = s, \tilde{a}_t = a, \tilde{s}_{t+1} = s', \tilde{a}_{t+1} = a'|\mathcal{F}_{t-\tau})|$$

$$= \frac{1}{2} \sum_{s,a,s',a'} |P(s_t = s, a_t = a|\mathcal{F}_{t-\tau})P_t(s'|s,a)\mathbb{E}\left[\pi_t(a'|s')|\mathcal{F}_{t-\tau}, \mathcal{H}_t\right]$$

$$- P(\tilde{s}_t = s, \tilde{a}_t = a|\mathcal{F}_{t-\tau})P_{t-\tau}(s'|s,a)\pi_{t-\tau-1}(a'|s')|$$

$$\leq \frac{1}{2} \sum_{s,a,s',a'} |P(s_t = s, a_t = a|\mathcal{F}_{t-\tau})P_t(s'|s,a)\mathbb{E}\left[\pi_t(a'|s')|\mathcal{F}_{t-\tau}, \mathcal{H}_t\right]$$

$$- P(\tilde{s}_t = s, \tilde{a}_t = a|\mathcal{F}_{t-\tau})P_t(s'|s,a)\pi_{t-\tau-1}(a'|s')|$$

$$+ \frac{1}{2} \sum_{s,a,s',a'} |P(\tilde{s}_t = s, \tilde{a} = a|\mathcal{F}_{t-\tau})P_t(s'|s,a)\pi_{t-\tau-1}(a'|s')$$

$$- P(\tilde{s}_t = s, \tilde{a}_t = a|\mathcal{F}_{t-\tau})P_{t-\tau}(s'|s,a)\pi_{t-\tau-1}(a'|s')|$$

$$= \frac{1}{2} \sum_{s,a,s',a'} P_t(s'|s,a)P(s_t = s, a_t = a|\mathcal{F}_{t-\tau})|\mathbb{E}\left[\pi_t(a'|s')|\mathcal{F}_{t-\tau}, \mathcal{H}_t\right] - \pi_{t-\tau-1}(a'|s')|$$

$$+ \frac{1}{2} \sum_{s,a} |P(s_t = s, a_t = a|\mathcal{F}_{t-\tau}) - P(\tilde{s}_t = s, \tilde{a}_t = a|\mathcal{F}_{t-\tau})|$$

$$+ \frac{1}{2} \sum_{s,a,s',a'} P(\tilde{s}_t = s, \tilde{a}_t = a|\mathcal{F}_{t-\tau})\pi_{t-\tau-1}(a'|s')|P_t(s'|s,a) - P_{t-\tau}(s'|s,a)|$$

$$\leq \sqrt{|\mathcal{S}||\mathcal{A}|}\mathbb{E}\left[\|\pi_t - \pi_{t-\tau-1}\|_2\Big|\mathcal{F}_{t-\tau}\right] + d_{TV}\left(P(O_{t-1} \in \cdot|\mathcal{F}_{t-\tau}), P(\tilde{O}_{t-1} \in \cdot|\mathcal{F}_{t-\tau})\right)$$

$$+ \|\mathbf{P}_t - \mathbf{P}_{t-\tau}\|_\infty.$$

Finally, recursing backwards until $\tau$ yields the result. $\qquad\square$

**Lemma I.3.** *If an observation is denoted as $O = (s, a, s', a')$, then the following hold for all $t, t'$*

1. $\|Q_t^\pi\|_2 \leq U_Q$; $\|Q_t\|_2 \leq R_Q = U_Q$

2. $\|\mathbf{A}(O)\|_\infty \leq 2$; $\|\mathbf{A}(O)\|_2 \leq \sqrt{2}$

3. $\|\bar{\mathbf{A}}^{\boldsymbol{\pi},\mathbf{P}} - \bar{\mathbf{A}}^{\boldsymbol{\pi}',\mathbf{P}'}\|_\infty \leq 2d_{TV}\left(d^{\boldsymbol{\pi},\mathbf{P}} \otimes \boldsymbol{\pi} \otimes \mathbf{P} \otimes \boldsymbol{\pi}, d^{\boldsymbol{\pi}',\mathbf{P}'} \otimes \boldsymbol{\pi}' \otimes \mathbf{P}' \otimes \boldsymbol{\pi}'\right)$

4. $\|\boldsymbol{\psi}_{t+1} - \boldsymbol{\psi}_t\|_2 \leq \|\mathbf{Q}_{t+1} - \mathbf{Q}_t\|_2 + \|\mathbf{Q}_{t+1}^{\pi_{t+1}} - \mathbf{Q}_t^{\pi_t}\|_2$

*Proof.* We have the following.

1. See the projection operator $\Pi_{R_Q}(\cdot)$ used in Algorithm 1 and discussed further in section 5.1.

2. Follows from the definition of $\mathbf{A}(O)$ in section 5.1

3. Follows from the definition of $\bar{\mathbf{A}}^{\boldsymbol{\pi},\mathbf{P}}$ in section 5.1 and

$$\|\bar{\mathbf{A}}^{\boldsymbol{\pi},\mathbf{P}} - \bar{\mathbf{A}}^{\boldsymbol{\pi}',\mathbf{P}'}\|_\infty = \max_{s,a} \sum_{s',a'} |d^{\boldsymbol{\pi},\mathbf{P}}(s,a)\boldsymbol{\pi}(a|s)\mathbf{P}(s'|s,a)\boldsymbol{\pi}(a'|s')$$

$$- d^{\boldsymbol{\pi}',\mathbf{P}'}(s,a)\boldsymbol{\pi}'(a|s)\mathbf{P}'(s'|s,a)\boldsymbol{\pi}'(a'|s')|$$

4. By the definition of $\boldsymbol{\psi}_t = \mathbf{Q}_t - \mathbf{Q}_t^{\pi_t}$ and triangle inequality

$\qquad\square$

## J  ADDITIONAL RELATED WORK

| Setting | Algorithm | Regret | Model Free | Policy Based |
|---|---|---|---|---|
| Non-Stationary Infinite Horizon Average Reward | Lower Bound | $\Omega\left(\|\mathcal{S}\|^{\frac{1}{3}}\|\mathcal{A}\|^{\frac{1}{3}}D^{\frac{2}{3}}\Delta_T^{\frac{1}{3}}T^{\frac{2}{3}}\right)$ | - | - |
| | Jaksch et al. (2010) | $\tilde{\mathcal{O}}\left(\|\mathcal{S}\|\|\mathcal{A}\|^{\frac{1}{2}}DL^{\frac{1}{3}}T^{\frac{2}{3}}\right)$ | $\times$ | - |
| | Gajane et al. (2018) | $\tilde{\mathcal{O}}\left(\|\mathcal{S}\|^{\frac{2}{3}}\|\mathcal{A}\|^{\frac{1}{3}}D^{\frac{2}{3}}L^{\frac{1}{3}}T^{\frac{2}{3}}\right)$ | $\times$ | - |
| | Ortner et al. (2020) | $\tilde{\mathcal{O}}\left(\|\mathcal{S}\|\|\mathcal{A}\|^{\frac{1}{2}}D\Delta_T^{\frac{1}{3}}T^{\frac{2}{3}}\right)$ | $\times$ | - |
| | Cheung et al. (2020) | $\tilde{\mathcal{O}}\left(\|\mathcal{S}\|^{\frac{2}{3}}\|\mathcal{A}\|^{\frac{1}{2}}D\Delta_T^{\frac{1}{4}}T^{\frac{3}{4}}\right)$ | $\times$ | - |
| | Wei & Luo (2021) | $\tilde{\mathcal{O}}\left(\Delta_T^{\frac{1}{3}}T^{\frac{2}{3}}\right)$ | $\times$ | - |
| | This Work | $\tilde{\mathcal{O}}\left(\|\mathcal{S}\|^{\frac{1}{2}}\|\mathcal{A}\|^{\frac{1}{2}}\Delta_T^{\frac{5}{9}}T^{\frac{8}{9}}\right)$ | $\checkmark$ | $\checkmark$ |
| Non-Stationary Episodic | Lower Bound | $\Omega\left(\|\mathcal{S}\|^{\frac{1}{3}}\|\mathcal{A}\|^{\frac{1}{3}}\Delta_T^{\frac{1}{3}}H^{\frac{2}{3}}T^{\frac{2}{3}}\right)$ | - | - |
| | Domingues et al. (2021) | $\tilde{\mathcal{O}}\left(\|\mathcal{S}\|\|\mathcal{A}\|^{\frac{1}{2}}\Delta_T^{\frac{1}{3}}H^{\frac{4}{3}}T^{\frac{2}{3}}\right)$ | $\checkmark$ | $\times$ |
| | Wei & Luo (2021) | $\tilde{\mathcal{O}}\left(\Delta_T^{\frac{1}{3}}T^{\frac{2}{3}}\right)$ | $\checkmark$ | $\times$ |
| | Feng et al. (2023) | $\tilde{\mathcal{O}}\left(\tilde{d}^{\frac{1}{2}}H^2T^{\frac{1}{2}}\right)$ | $\checkmark$ | $\times$ |
| | Mao et al. (2024) | $\tilde{\mathcal{O}}\left(\|\mathcal{S}\|^{\frac{1}{3}}\|\mathcal{A}\|^{\frac{1}{3}}\Delta_T^{\frac{1}{3}}HT^{\frac{2}{3}}\right)$ | $\checkmark$ | $\times$ |
| Non-Stationary Episodic Linear MDP | Zhou et al. (2020) | $\tilde{\mathcal{O}}\left(d^{\frac{4}{3}}\Delta_T^{\frac{1}{3}}H^{\frac{4}{3}}T^{\frac{2}{3}}\right)$ | $\checkmark$ | $\times$ |
| | Touati & Vincent (2020) | $\tilde{\mathcal{O}}\left(d^{\frac{5}{4}}\Delta_T^{\frac{1}{4}}H^{\frac{5}{4}}T^{\frac{3}{4}}\right)$ | $\checkmark$ | $\times$ |
| Stationary Infinite Horizon Discounted Reward | Khodadadian et al. (2022) | $\tilde{\mathcal{O}}\left(T^{\frac{5}{6}}\right)$ | $\checkmark$ | $\checkmark$ |
| Stationary Infinite Horizon Average Reward | Wang et al. (2024) | $\tilde{\mathcal{O}}\left(T^{\frac{2}{3}}\right)$ | $\checkmark$ | $\checkmark$ |

Table 1: Regret comparison across Non-Stationary and Stationary RL algorithms with variation budget $\Delta_T$, time horizon $T$, episode length $H$, size of the state-action space $\|\mathcal{S}\|$, $\|\mathcal{A}\|$, maximum diameter of MDP $D$, dimension of feature space $d$ and dynamic Bellman Eluder dimension $\tilde{d}$.

## K    SIMULATION SETUP

**Synthetic Environment.**    We empirically evaluate the performance of our algorithm NS-NAC on a synthetic non-stationary MDP, comparing it with three baseline algorithms: SW-UCRL2-CW (Cheung et al. (2023)), Var-UCRL2 (Ortner et al. (2020)), and RestartQ-UCB (Mao et al. (2024)). The synthetic MDP environment simulates non-stationary dynamics by alternating between two sets of transition matrices and reward functions over the time horizon $T$. The switching frequency, controlled by $n_{\text{switches}}$, determines the degree of non-stationarity and the variation budget $\Delta_{P,T}$ for transitions and $\Delta_{R,T}$ for rewards. The MDP consists of $|\mathcal{S}|$ states and $|\mathcal{A}|$ actions per state, with two sets of transition probabilities and rewards sampled at initialization. Further, to benchmark the effect of the dynamic changes, the optimal policy is recalculated at each switching step $t_{switch}$ by solving a linear programming problem (Puterman (2014)).

The environment alternates between these two sets of transitions and rewards, $(\mathbf{P}_1, \mathbf{r}_1)$ and $(\mathbf{P}_2, \mathbf{r}_2)$, every $T/n_{\text{switches}}$ steps. The transition probabilities, $\mathbf{P}_1$ and $\mathbf{P}_2$, are drawn from a Dirichlet distribution with a concentration parameter set to $0.5$, ensuring a moderate degree of randomness in the state transitions. The first reward matrix $\mathbf{r}_1$ is drawn from a Beta distribution with shape parameters $\alpha = 0.5$ and $\beta = 0.5$, leading to rewards spread across the interval $[0, 1]$, with a higher probability near the extremes of $0$ and $1$. The second reward matrix $\mathbf{r}_2$ is sampled from a Beta distribution with shape parameters $\alpha = 0.2$ and $\beta = 0.9$, producing rewards skewed toward lower values, introducing diversity in the reward structure. We use 5 random seeds to initialize the matrices, with the standard deviation capturing the variability across these runs.

**Varying $T$.**    We evaluate the performance of different algorithms in a synthetic environment with $|\mathcal{S}| = 50$ and $|\mathcal{A}| = 4$ under varying time horizons $T$. Specifically, the time horizon $T$ is varied over the values $50 \times 10^3, 70 \times 10^3, 100 \times 10^3, 150 \times 10^3, 180 \times 10^3, 200 \times 10^3$, and $250 \times 10^3$. For each $T$, we set $n_{\text{switches}} = 1000$, resulting in a transition variation budget $\Delta_{P,T} \approx 300$, indicating significant environmental changes across the time horizon. The reward function is kept stationary (no switching between $\mathbf{r}_1$ and $\mathbf{r}_2$), and therefore $\Delta_{R,T} = 0$.

**Varying $\Delta_T$.**    We investigate the impact of changing variation budget by adjusting the number of switches $n_{\text{switches}}$ while keeping the number of states $|\mathcal{S}| = 50$, actions $|\mathcal{A}| = 4$, and the time horizon $T = 50 \times 10^3$ constant. The number of switches is varied across $10, 45, 100$, and $1000$, with both the reward function and the transition dynamics being non-stationary. The observed variation in rewards $\Delta_{R,T}$ is $9, 48, 98$, and $1000$, respectively, and the observed variation in transitions $\Delta_{P,T}$ is $4, 14, 30$, and $303$, respectively, corresponding to different levels of non-stationarity.

**Varying $|\mathcal{S}|$.**    We study the effect of varying the number of states while keeping the time horizon $T$, number of actions, and variation budget $\Delta_T$ constant. Specifically, the time horizon $T$ is fixed at $50 \times 10^3$ steps, and the number of states is varied across the values $100, 150, 175$, and $200$, corresponding to environments with different state sizes while keeping the number of actions fixed at $4$. The $n_{\text{switches}}$ is adjusted to $75, 100, 120$, and $150$, respectively, in order to maintain a consistent $\Delta_{P,T}$ of around $14$ for all environments. The reward function is kept stationary (no switching between $\mathbf{r}_1$ and $\mathbf{r}_2$), and therefore $\Delta_{R,T} = 0$.

**Varying $|\mathcal{A}|$.**    We examine the effect of varying the number of actions while keeping the time horizon $T$, number of states, and variation budget $\Delta_T$ constant. Specifically, the time horizon $T$ is fixed at $50 \times 10^3$ steps, and the number of actions is varied across the values $5, 10, 20$, and $25$, corresponding to environments with different action sizes while keeping the number of states fixed at $50$. The $n_{\text{switches}}$ is kept constant at $45$ across all experiments to maintain a consistent variation budget $\Delta_{P,T}$ of around $14$ for all environments. The reward function is kept stationary (no switching between $\mathbf{r}_1$ and $\mathbf{r}_2$), and therefore $\Delta_{R,T} = 0$.

**Parameters.**    The true variation budgets, $\Delta_{P,T}$ and $\Delta_{R,T}$, are provided to each algorithm, while the remaining hyperparameters are configured according to the optimal expressions derived in their respective papers. For SW-UCRL2-CW, the parameters include the window size $W_*$ and the confidence widening parameter $\eta_*$, both set using the optimal expressions given in the paper, and the confidence parameter $\delta = 0.05$. For Var-UCRL2, the true values of the variation budgets for transitions probabilities $\Delta_{P,T}$ and rewards $\Delta_{R,T}$, along with the confidence parameter $\delta = 0.05$, are

used. In RestartQ-UCB, the ending times of the stages $L$, confidence parameter $\delta = 0.05$, initial number of samples $N_0$, and number of epochs $D$ are configured as described in the original paper with $H = 1$ (to adapt from episodic setting for which the algorithm is designed to infinite horizon setting in our work). Further, for NS-NAC, we tune the step-sizes by grid search. The effect of different choices of step-sizes can be observed in Figure 2.

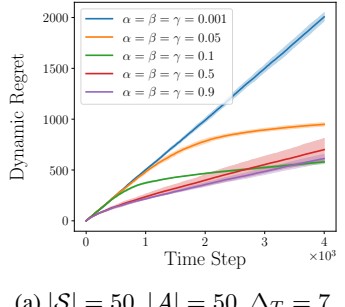

(a) $|\mathcal{S}| = 50$, $|\mathcal{A}| = 50$, $\Delta_T = 7$

Figure 2: Performance of NS-NAC with different step-sizes in an environment with 17 abrupt, randomly scheduled switches over $T = 4 \times 10^3$ steps.

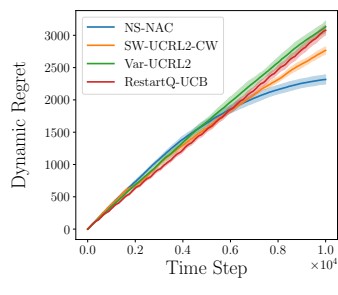 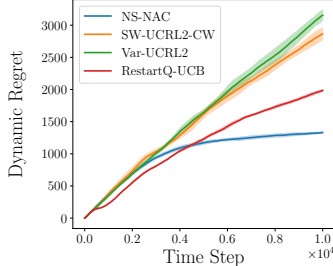

(a) $|\mathcal{S}| = 50$, $|\mathcal{A}| = 4$, $\Delta_T = 15$      (b) $|\mathcal{S}| = 50$, $|\mathcal{A}| = 4$, $\Delta_T = 0.06$

Figure 3: Performance of NS-NAC and baseline algorithms in various non-stationary settings. (a) Dynamic regret for a single instance over $T = 1 \times 10^4$ steps in an environment with 50 abrupt, randomly scheduled switches. (b) Dynamic regret for a single instance over $T = 1 \times 10^4$ steps in an environment with small, continuous changes.

**Additional Environments.** We conducted further experiments to evaluate the adaptability of NS-NAC and baseline algorithms across diverse non-stationary settings. Figure 3a illustrates performance in an environment with 50 abrupt and randomly scheduled switches (between $\mathbf{P}_1$ and $\mathbf{P}_2$), simulating scenarios with non-periodic unpredictability. Figure 3b captures performance in a continuously changing environment, where the transition from $\mathbf{P}_1$ to $\mathbf{P}_2$ occurred gradually over $T = 10^5$ steps resulting $\Delta_T = 0.06$. This scenario reflects real-world conditions where systems experience smooth drift rather than abrupt changes. The results highlight NS-NAC's effectiveness in handling both abrupt and gradual changes, consistently matching the performance of baseline methods.

