# OpenReview forum: "Natural Policy Gradient for Average Reward Non-Stationary RL"
_ICLR.cc/2025/Conference — Submitted to ICLR 2025_

### Official Review · Reviewer_yb2x · 2024-10-26

**Soundness:** 3
**Presentation:** 3
**Contribution:** 2
**Rating:** 6
**Confidence:** 3

**Summary:**

The paper studies a natural actor-critic algorithm for minimizing dynamic regret in non-stationary MDPs with fixed variation budget. The main contribution is to derive a regret bound that scales sublinearly in $T$. The paper makes an assumption on the contraction of the Bellman operator. The authors showed that this assumption can be removed by adding an entropy regularization term with controlled weight.

**Strengths:**

The paper tries to fill in an important gap in the literature, which is to solve non-stationary RL problems with a policy gradient algorithm. The paper is very well written with background information, algorithm development, and main proof steps clearly presented. The authors are able to show that the natural actor critic algorithm achieves a sublinear regret, which is the first time such result appears in the literature.

**Weaknesses:**

1) My main concern is that the paper seems to have limited technical innovation. The most challenging component of the analysis is to handle samples from a time-varying Markov chain induced by changing policies and transition probabilities. However, as the authors noted, the technique for doing so is standard if only the policies are changing. To handle the changes in $P$ seems to requires additionally bounding the TV distance in the samples by the shift in $P$, which is a straightforward extension. It is possible that I underestimate the amount of efforts and novelty in the analysis. In that case, I suggest that authors clearly state the key technical challenge that prevents existing analysis from being applied with small modification, as well as the novel analytical techniques that authors come up with.

2) The regret of the proposed algorithm is much inferior to that of the state-of-the-art value-based algorithms (for example, Mao et al., 2024), especially in terms of $T$. Though I understand that the techniques used to analyze gradient-based and value-based algorithms are quite different, the gap still undermines the contribution of the work. This work differs from Mao et al., 2024 in so many aspects that I cannot see where the degradation in the regret comes from (Q learning and policy gradient algorithm difference, actively restarting versus passive forgetting, episodic MDP versus infinite-horizon). Can the authors provide some discussion on why they think they cannot get a regret on the same order as Mao et al., 2024. An infinite-horizon average-reward MDP does not seem to be fundamentally harder than an episodic one, since the lower bounds in Mao et al., 2024 are similar for the two settings.

3) The statement "The projection step in line 9 ... ensures a unique solution to the Bellman equations in the average reward setting" is incorrect. Whatever algorithm you use to find a solution has nothing to do with whether the solution exists/is unique. The actual reason for using the projection is to guarantee boundedness of the critic iterates. It is by Assumption 2 that the critic solution is unique. It is known that Assumption 2 does not hold in any average-reward MDP (see Zhang et al., 2021). Any solution to the Bellman equation remains a solution when added with an all-one vector. Making an assumption that never holds weakens the theoretical results, even though the same assumption has been made in a number of existing works. However, the issue can be circumvented by slightly altering the critic convergence metric. Essentially, if we use an argument similar to the one in Zhang et al., 2021, Assumption 2 does not need to be assumed but can be shown to hold in a proper subspace. The entropy regularization approach discussed in line 385-397 is actually unnecessary if its sole purpose is to remove Assumption 2.

References

Zhang, S., Zhang, Z. and Maguluri, S.T., 2021. Finite Sample Analysis of Average-Reward TD Learning and $ Q $-Learning. Advances in Neural Information Processing Systems, 34, pp.1230-1242.

**Questions:**

The remark "Note that $N$ is an artifact of the proof ... to optimize the regret upper bound" is repeated a few times in the appendix. This should be moved to the main paper after Theorem 1. In this work $N$ is chosen as a function $T$ agnostic of the environmental change. If we (roughly) know how frequently the environment changes, is there a better way of choosing $N$ that can further enhance the bound?

Minor suggestion: I think the paper would benefit from trimming Section 1 for a quick introduction into the subject. It is natural for the audience to understand why RL in a non-stationary environment is a challenging and important problem to study (same for average reward vs discounted reward). A lot of the references in the first two paragraphs of Section 1 can actually be removed.

---

### Official Review · Reviewer_suBn · 2024-11-01

**Soundness:** 3
**Presentation:** 3
**Contribution:** 3
**Rating:** 6
**Confidence:** 2

**Summary:**

This paper considers the natural policy gradient in the non-stationary Markov decision processes. The authors propose the NS-NAC algorithm to solve this problem and derive the performance analysis of the algorithm. The authors also relax the assumptions of eigenvalues with the entropy regularization.

**Strengths:**

1.	This work is the first to derive the theoretical analysis of the natural gradient descent in the non-stationary Markov decision process.
2.	The authors show that the entropy regularization can remove the strong assumption on eigenvalues.

**Weaknesses:**

1.	The parameter configurations in Theorem 1 require further clarification. It is not clear whether the current configuration is the optimal one. For example, for parameter $N$, it should be chosen such that $N/\beta$ and $\Delta _{T}T/N$ have the same order. However, the current configuration does not satisfy this .

2.	The suboptimality of the algorithm requires more discussion. As mentioned by the authors, the regret of NAC for an infinite horizon stationary MDP can be $O(T^{2/3})$. It is not clear whether this can be achieved for non-stationary setting under the similar assumptions.

3.	The writing for the entropy regularization section is unclear. According to my understanding, people usually include the entropy term in the reward definition, and the regret is measured under this newly defined reward. However, the authors do not define this reward and just adopt the same notation for the un-regularized and regularized case. It is confusing since it looks like that adding a regularized term in the algorithm can relax the assumption for the un-regularized problem.

**Questions:**

Same as the Weakness

---

### Official Review · Reviewer_ZkkY · 2024-11-01

**Soundness:** 3
**Presentation:** 3
**Contribution:** 2
**Rating:** 5
**Confidence:** 3

**Summary:**

The paper presents Non-Stationary Natural Actor-Critic (NS-NAC), a model-free policy gradient method for time-varying environments. The algorithm uses a two timescale techniques such that the learning rates can balance the exploration-exploitation challenge in non-stationary RL problem. A solid bound on the dynamic regret is derived and presents an interpretation aligns with the intuition. Although the optimality of the derived bound is unclear, the numerical experiments present a sub-linear rate across different parameters.

**Strengths:**

* The paper is clearly written and provides a comprehensive overview of relevant literature.
* The paper presents the first theoretical result on dynamic regret for model-free policy-based algorithm. Although the bound might not be optimal, it provides an intuitive interpretation on the performance of the algorithm. The proof techniques developed in the paper may be of independent interest.
* The paper carries out a comparison study with current state-of-art methods in non-stationary RL. The effectiveness is demonstrated by the experiments.

**Weaknesses:**

The novelty of the proposed algorihtm is a bit weak. The NAC algorithm is kind of standard. The primary innovation in NS-NAC appears to be the two-timescale technique, which essentially coordinates the learning rates between actor and critic updates to ensure slower policy learning relative to Q-function evaluation.

**Questions:**

1. Regarding Assumption 2 and the discussion of projection radius, I'm curious about its practical implementation. Since $\lambda$ may not be available in practice, could you elaborate on how to select an appropriate projection radius? Additionally, it would be valuable to understand the algorithm's sensitivity to different projection radius values - specifically, how robust is the performance across various choices of this parameter?
2. The optimal choice of $\alpha$ and $\beta$ depends on variation parameter $\Delta_T$, which I think might not be available in practice. However, the choice of $\alpha$ and $\beta$ is important to the algorithm. I am curious that would different choice of $\alpha$ and $\beta$ introduce stability issues in the training procedure? How does the author choose the learning rate here?
3. In the optimal bound (8), the dependence on $\Delta_T$ is $\Delta_T^{1/9}$, which can be even smaller than the corresponding term $\Delta_T^{1/3}$ in the lower bound (Theorem 2). Do we have any intuitive explanations?
4. In line 241, I guess "lest" is a typo?

---

### Official Review · Reviewer_QoU5 · 2024-11-03

**Soundness:** 3
**Presentation:** 4
**Contribution:** 3
**Rating:** 6
**Confidence:** 4

**Summary:**

This paper introduces a model-free policy-based approach, non-stationary natural actor-critic (NS-NAC), for reinforcement learning in non-stationary environments with infinite-horizon average-reward setting, where the reward functions and transition probabilities are time varying.
The NS-NAC algorithm extends average reward policy optimization to handle non-stationarity by interpreting learning rates as adapting factors. A bound on the dynamic regret is shown, and competitive performance of the NS-NAC algorithm is illustrated by comparison with several baseline algorithms.

**Strengths:**

To the best of my knowledge, it is the first paper about the policy-based method for non-stationary RL. The authors support their approach with theoretical insights, including regret bounds and Lyapunov-based analysis, and demonstrate empirical effectiveness in comparison to established baseline algorithms across various environments. The paper is well-written and easy to read.

**Weaknesses:**

1. The paper seems to assume the optimal policy of a non-stationary average-reward MDP (i.e., $\pi^* = \arg\max_\pi\lim_{T\rightarrow\infty}\sum_{t=0}^Tr_t(s_t,a_t)/T$) is equivalent to the combination of optimal policies of each MDP $M_t$ environment at each time $t$ (i.e., $\pi_t^*$ in (4)). Although not explicitly stated, the authors seem to indicate this equivalence by the definition of the dynamic regret and statement such as "the agent chases a moving target, namely, the time-varying optimal policy $\pi_t^*$". However, this equivalence does not hold in general (please see my first comment in Questions below). Please clarify whether this equivlance holds in the paper. If the equivalence does not hold, then please explain why the dynamic regret is defined in the way of (4).

2. The premise of the problem is not clear, and it is not clear how to choose the learning rates in practice since the choice depends on quantities that are unknown in advance. Please see my sencond comment in Questions below.

3. Assumption 1 lacks justification. Please give examples that satisfy Assumption 1 and explain how to verify it in practice.

**Questions:**

1. The authors use the long-term average reward for the optimal policy in the environment $M_t$ to define regret at stage $t$.  However,  $M_t$ only occurs in time $t$ for once. It is possible that a sub-optimal action at $t^{th}$ stage may lead to a good starting state for the future stages. So, considering the optimal policy in the environment $M_t$ at each stage may not lead to the optimal long-run average reward.
Is it a good performance measure?

2. Is the cumulative change in the reward and transition probabilities, $\Delta_T$, known in advance? If it is not known in advance and is only revealed at each time stage, how to choose the learning rates since the choice of learning rates require to know this quantity in advance?

3. Could you explain more on how adaptive learning rate help to overcome two challenges: Explore-for-Change vs Exploit and Forgetting Old Environments? From the current choice of learning rates, they are just constants. Is it possible to choose them as changeable depending on the change speed of the environment? This sounds more meaningful.

4. Why using the two-timescale learning rates in the algorithm? Please explain its intuition, and also explain how two-timescale affects the regret analysis.

---

### Official Review · Reviewer_nm7s · 2024-11-03

**Soundness:** 4
**Presentation:** 4
**Contribution:** 3
**Rating:** 8
**Confidence:** 4

**Summary:**

This paper tackles the challenging problem of non-stationary average reward reinforcement learning (RL). While prior work in average reward RL has largely concentrated on value-based methods or policy-based methods in stationary environments, this study advances the theoretical understanding of policy-based approaches for the non-stationary setting. Specifically, the authors introduce the Non-Stationary Natural Actor-Critic (NS-NAC) algorithm and establish a sublinear dynamic regret bound, achievable through an optimal choice of learning rates and perturbation budgets.

**Strengths:**

The paper makes a significant theoretical contribution by proving the convergence of a natural actor-critic algorithm in a non-stationary Markov Decision Process (MDP) environment. Although I'd argue the NS-NAC algorithm is not novel (see Weaknesses), the theoretical analysis which considers non-stationarity requires a non-trivial and novel extension of previous work, particularly from Wu et al. (2020) and Khodadadian et al. (2022). This extension has the potential to meaningfully advance our understanding of policy gradient methods within the average reward context.

The paper is generally well-written and clear, with only minor areas requiring clarification.

The ablation study in the experiments provides solid empirical support for the derived sublinear regret bound, enhancing the practical validity of the theoretical results.

The proof sketches effectively outline the key steps, clearly linking previous work to the new theoretical contributions presented in this paper.

**Weaknesses:**

**Assumption 2 cannot be satisfied in the tabular setting**: It is my understanding that this paper has a serious technical error, which is that the tabular formulation of the average reward critic in NS-NAC means that Assumption 2 is not satisfiable under any circumstances. Specifically, in the tabular setting, the $\bar{A}$ that they constructed will always have a row sum of 0. Therefore $A$ is singular, and $0$ must be an eigenvalue which does not satisfy Assumption 2 which states that the largest eigenvalue of $\bar{A}$ is $-\lambda$ for $\lambda > 0$. To see that the row sums are always 0, given some observation $O(t) = (s_t, a_t, s_{t+1}, a_{t+1})$, if $(s_t, a_t) = (s_{t+1}, a_{t+1})$, the entire A matrix is $0$. So clearly in this case the row sums are 0. If $(s_t, a_t) \neq (s_{t+1}, a_{t+1})$ then one can readily check that there is a $-1$ on the main diagonal of A(O_t) and a single $+1$ value somewhere else on the same row. This means that for the row of $A(O_t)$ corresponding to $(s_t, a_t)$ there is a $-1$ entry and a $+1$ entry and every other entry is 0. This means means all the row sums are still 0 when $(s_t, a_t) \neq (s_{t+1}, a_{t+1})$. Then, checking that the row sums of $\bar{A}$ are also 0 is trivial by the linearity of expectation (i.e.  $\sum_j \bar{A}_{(s,a),j} = E_{s,a,s',a'}[\sum_j A_{(s,a),j}] = E[0] = 0$). Since Assumption 2 cannot be verified for the tabular setting, the authors cannot guarantee that there exists a unique solution to the problem which invalidates the rest of their analysis.

Its worth noting that this problem of invertibility of $\bar{A}$ is the reason why the convergence of tabular average reward TD cannot be proven simply by invoking the convergence results for average reward TD with linear function approximation from works like Tsitsiklis and van Roy 1999 or Wu et al 2020 but with one-hot features. If I am incorrect in my analysis please let me know and I'm happy to change my score. However, this error renders the rest of the analysis in the paper to be incorrect and so it is for this reason I am recommending rejection.

There are a few other technical concerns and points for clarification:

- **Algorithm 1:** The input includes an exploration parameter, but the algorithm description does not utilize it. I assume the authors meant to include the epsilon greedy action selection but it appears it was forgotten.
- **Non-stationarity in Notation:** The Bellman equation for the value function should index policy $\pi$ by $t$ to reflect non-stationarity and to be consistent with subsequent notations.
- **Experiment Plots:** Uncertainty measures are inconsistently reported. Some plots use shaded regions, while others use error bars. Consistency here would be nice.
- **Uniformly distributed switches in Experiments**: In the experiment setup in section L, it appears that the switches betweent the two transition matrices are uniformly spread out throughout the time horizon. This is not captured in any of the theoretical assumptions for the Algorithm. Additionally, this setting seems like the best case scenario for the algorithm as the learning rate schedule (which is fixed and depends on $\delta T / T$) could be tuned to provide the best performance based on the consistent switching times. It would be nice to see an experiment where the switching times are randomly distributed throughout $T$. At the same time, I do recognize that this is primarily a theoretical paper with substantial theoretical contributions.

**Novelty of the Proposed Algorithm**

While the proposed NS-NAC algorithm is presented as a novel approach, it appears to align closely with a two-timescale natural gradient actor-critic method from Khodadadian et al 2022. The main distinction lies in the derivation of the two timescale learning rates $\beta_t$ and $\gamma_t$, which are optimally set based on a perturbation budget. However, this alone may not justify categorizing NS-NAC as a new algorithm. Instead, it could be more accurately described as a traditional natural actor-critic with a specialized learning rate schedule for application to the non-stationary setting. This is analogous to tabular TD learning using $1/t$ versus $1/t^2$ learning rates, which represent variations within the same algorithm rather than fundamentally new methods. If there are other differences between NS-NAC and the original NAC but applied to a non-stationary environment, additional clarification would be helpful. Again, the analysis of this NAS algorithm is already important itself, regardless if this "NS-NAC" algorithm is truly "new".

**Questions:**

Is there a regret result that depends independently on the total variation metrics $\Delta_{R,T}$ and $\Delta_{P,T}$ instead of just $\Delta_{T} = \Delta_{R,T} + \Delta_{P,T}$? Or do the perturbations to each contribute equally to the overall regret?

---

### Official Review · Reviewer_UzLT · 2024-11-03

**Soundness:** 3
**Presentation:** 2
**Contribution:** 2
**Rating:** 5
**Confidence:** 3

**Summary:**

This paper studies the non-stationary reinforcement learning in average reward setting. The paper proposed natural policy gradient based actor-critic algorithm (NS-NAC), which aims to minimize the dynamic regret, a metric in non-stationary setting. The proposed algorithm achieves a regret upper bound of $\tilde{\mathcal{O}}(|S||A|\Delta_T T^{\cfrac{8}{9}})$, where the $S\times A$ is the state-action space and $\Delta_T$ non-stationarity level and $T$ is time horizon. In addition, synthetic empirical results shown to suggest that NS-NAC has comparative performance with some related algorithms.

**Strengths:**

* The paper explored the model-free approach in non-stationary RL in average reward setting and result-wise, $\Delta_T$ dependency seems to be better than related algorithms based on Table 1.

* Introducing auxiliary Markov chain for non-stationary setting seems a relative novelty in the analysis.

**Weaknesses:**

* The regret upper bound seems to be worse in time horizon $T$ dependency.

* The synthetic empirical results only seem to suggest comparative results. In addition, the experimental setting seems relatively simple.

* The implementation of the algorithm depends on strong assumptions on knowing variation budgets $\Delta_{R,T}$ and $\Delta_{P,T}$. Please justify such an assumption.

**Questions:**

In addition to addressing the weakness items, the reviewer has the following questions:

1) In the implementation of algorithm1, it seems to suggest the use of variation budgets $\Delta_{R,T}$ and $\Delta_{P,T}$ and $\epsilon$. However, it's not clear in the algorithm, where these quantities have been used.

2) The last paragraph in section 4 doesn’t seem to provide much argument, as the same argument can be applied to stationary setting. Please provide clarification.

3) From Table 1, the theory seems to suggest NS-NAC should perform better when $\Delta_T$ is large, compared with other algorithms. But the simulation results in Figure 1(c) don’t seem to suggest such an advantage. Please elaborate.

4) What is the difference between Figure 1(a) and 1(b) besides the fact that (b) uses log scales?

---

### Official Review · Reviewer_VTBX · 2024-11-04

**Soundness:** 2
**Presentation:** 2
**Contribution:** 2
**Rating:** 3
**Confidence:** 4

**Summary:**

This paper studies the problem of non-stationary RL in the infinite horizon average reward setting. The reward function and transition kernel change with time. The change budget is bounded by $\Delta_T$. This paper develops a model-free policy-based algorithm, Non-Stationary Natural Actor-Critic (NS-NAC. A dynamic regret was proved.

**Strengths:**

This paper provides the first policy gradient-based method for average reward RL in non-stationary environments, and derive dynamic regret bounds.

**Weaknesses:**

1. The regret definition used in this paper is questionable. J_t^{\pi*_t} is the optimal reward in the steady state for reward r_t and transition kernel P_t. It may not represent the largest reward that can be achieved at time $t$. Therefore, measuring the difference from J_t^{\pi*_t} may not be a good option. In other words, the definition in (4) may be negative.
2. Analysis in this paper is limited to the tabular case. While existing analysis for AC and NAC with function approximation are quite mature, the results in this paper then seem rather limited.
3. The assumption of uniform ergodicity rules out the need for exploration in this paper. This makes the contribution of this paper weak comparing to value based approaches where exploration is explicitly addressed.
4. In the literature, there are studies on adversarial MDPs with corrupted transition kernels, see e.g.,
No-Regret Online Reinforcement Learning with Adversarial Losses and Transitions
The results in this paper has a linear regret in T if the variation budget is linear in T. Even if the variation budget is 0, i.e., it is a stationary problem, the regret has an order of T^8/9, which is strictly suboptimal.

**Questions:**

see weaknesses.

---

> ### Comment · Reviewer_VTBX · 2024-11-24
> **Thank you**
>
> I would like to thank the authors for the response. However my major concern about the regret , challenge of exploration and limited analysis to tabular setting remains. Therefore I keep my original score.

---

### Meta-Review · Area_Chair_9rf7 · 2024-12-17

**Metareview:**

The reviewers appreciate the extension of the analysis of NPG to non-stationary settings and the introduction of adaptive exploration schedule through the use of entropy regularization. However, without algorithmic understanding of whether changes occur, the core technical results appear to hinge upon quantities that are practically unknown, making the applicability of the proposed techniques of limited usefulness. For this reason, the paper does not meet the bar for acceptance at this time.

**Additional Comments On Reviewer Discussion:**

See above.

---

### Decision · Program_Chairs · 2025-01-22

Reject